# Modelling the water budget of the Upper Blue Nile basin using the JGrass-NewAge model system and satellite data

Wuletawu Abera[1,2], Giuseppe Formetta[3], Luca Brocca[4], and Riccardo Rigon[1]

[1]Department of Civil, Environmental and Mechanical Engineering, University of Trento, Italy
[2]Department of Geography and Environmental Studies, Mekelle University, Ethiopia
[3]Centre for Ecology & Hydrology, Crowmarsh Gifford, Wallingford, UK
[4]Research Institute for Geo-Hydrological Protection, National Research Council, Perugia, Italy

*Correspondence to:* Wuletawu Abera (wuletawu979@gmail.com); Riccardo Rigon( riccardo.rigon@unitn.it)

**Abstract.** The Upper Blue Nile basin is one of the most data-scarce regions in developing countries, hence, the hydrological information required for informed decision making in water resources management is limited. The hydrological complexity of the basin, tied with the lack of hydrometerological data, means that most hydrological studies in the region are either restricted to small subbasins where there are relatively better hydrometeorological data available, or at the whole basin scale but at
very coarse time scales and spatial resolutions. In this study we develop a methodology that can improve the state-of-art by using the available, but sparse, hydrometerological data and satellite products to obtain the estimates of all the components of the hydrological cycle (precipitation, evapotranspiration, discharge, and storage). To obtain the water budget closure, we use the JGrass-NewAge system and various remote sensing products. The satellite products SM2R-CCI is used for obtaining the rainfall inputs; SAF EUMETSAT for cloud cover fraction for proper net radiation estimation; GLEAM for comparison with
NewAge estimated ET and GRACE gravimetry data for comparison of the total water storage amounts available in the whole basin. Results are obtained at daily time-steps for the period 1994-2009 (16 years), and they can be used as a reference for any water resource development activities in the region. The overall mean budget analysis shows that precipitation of the basin is 1360 ±230 mm per year. Evapotranspiration accounts for 56% (per cent) of the annual water budget, runoff is 33%, storage varies from minus 10% to plus 17% of the water budget.
**Key Words:** Water budget, Upper Blue Nile, JGrass-NewAge system, Satellite data, evapotranspiration

## 1   Introduction

Freshwater is a scarce resource in many regions of the world: the problem continues to be aggravated by growing populations and significant increases in demand for agricultural and industrial purposes. The Nile River basin is one such region, with relatively arid climate because of high temperatures and solar radiation which foster rapid evapotranspiration. Most of the
countries within the basin, such as Egypt, Sudan, Kenya, and Tanzania, receive insufficient fresh water (Pimentel et al., 2004). Exceptions to this are the small areas at the equators and the Upper Blue Nile basin in the Ethiopian highlands, which receives up to 2000 mm of precipitation per year (Johnston and McCartney, 2010). Particularly, the Upper Blue Nile (hereafter UBN) basin is the main sources of water in the region.

In Ethiopia, UBN is inhabited by 20 million people whose main livelihood is subsistence agriculture (Population Census Commission 2008). The Ethiopian government, therefore, has started many water resource development projects, such as irrigation schemes and dams, among which the Grand Ethiopia Renaissance Dam (GERD), which, upon completion, will be one of the largest in Africa. However, as the principal contributor (i.e 51% of discharge) to the main Nile basin, UBN also supports hundreds of millions of people living downstream, and it is referred to as the "Water Tower" of northeast Africa. Therefore UBN is a part of trans-boundary river, and its development and management require obtaining agreements between many national governments and also non-governmental organizations, each involving different policies, legal regimes, and contrasting interests. Tackling all these complexities and developing better water resource development strategies is only possible by gathering quantitative information (Hall et al., 2014). Understanding the hydrological processes of UBN, therefore, is the basis for both the transboundary negotiations about sharing the water resources and for assessing the sustainability of farming systems in the region. In fact, because of the lack of hydrometeorological data and a proper modelling framework, the recent modelling efforts conducted within the basin have evident limitations in addressing these problems. Studies in the region are limited to small basins, particularly within the Lake Tana basin where there are relatively better hydrometeorological data (Rientjes et al., 2011; Uhlenbrook et al., 2010; Tekleab et al., 2011; Wale et al., 2009; Kebede et al., 2006; Bewket and Sterk, 2005; Steenhuis et al., 2009; Conway, 1997; Mishra et al., 2004; Mishra and Hata, 2006; Teferi et al., 2010), or at the whole basin scale, but in which case information on spatial variability is usually ignored (Kim et al., 2008; Kim and Kaluarachchi, 2009; Gebremicael et al., 2013; Tekleab et al., 2011). Other studies are limited to a specific hydrological process e.g. rainfall variability (Block and Rajagopalan, 2007; Abtew et al., 2009) and evapotranspiration (Allam et al., 2016), time series and statistical analysis of in situ discharge/rainfall data (Teferi et al., 2010; Taye and Willems, 2011) or perform modelling at very low temporal resolutions (e.g. monthly) (Kim and Kaluarachchi, 2008; Tekleab et al., 2011). Spatially distributed information on all the components of the water budget does not exist and basin modelling approaches that are tailored to a single component do not provide an effective picture of the dynamics of the water resources within the basin.

To overcome data scarcity, large scale hydrological modelling can be supported by remote sensing (RS) products, which fill the data gaps in water balance dynamics estimation (Sheffield et al., 2012). For instance, a considerable number of researches has been carried out in the last two decades in developing satellite rainfall estimations procedures (Hong et al., 2006; Bellerby, 2007; Huffman et al., 2007; Kummerow et al., 1998; Joyce et al., 2004; Sorooshian et al., 2000; Brocca et al., 2014). RS is also a viable option to fill the gaps for basin scale evapotranspiration estimation. Global satellite evapotranspiration products have been available by applying energy balance and empirical models to satellite derived surface radiation, meteorology and vegetation characteristics, and they are recognised to have a certain degree of reliability (e.g. Fisher et al., 2008; Mu et al., 2007; Sheffield et al., 2010). Basin scale storage estimation is the most difficult task. Fortunately, the Gravity Recovery and Climate Experiment (GRACE) (Landerer and Swenson, 2012) came to fill this gap (e.g. Han et al., 2009; Muskett and Romanovsky, 2009; Rodell et al., 2007; Syed et al., 2008; Rodell et al., 2004). Guntner (2008), Ramillien et al. (2008) and Jiang et al. (2014) reviewed the use of GRACE data and positively recommended it for large scale water budget modeling. At the moment, satellite based retrievals of discharge are not available as operational or research products, but, potentially it can be retrieved from satellite altimetry and multispectral sensors (e.g. Tarpanelli et al., 2015; Van Dijk et al., 2016). Moreover, the Surface

Water Ocean Topography (SWOT, Durand et al. (2010)) mission, which is expected to be launched in 2020, will provide river elevation (with an accuracy of 10 cm), slope (with an accuracy of 1 cm/1 km) and width that can be used in estimating river discharge (Paiva et al., 2015; Pavelsky et al., 2014). Notwithstanding the availability of these RS products at various (spatial and temporal) resolutions and accuracy, their use is clearly a new paradigm in water budget closure estimations (Sheffield et al.,

2009; Andrew et al., 2014; Sahoo et al., 2011; Gao et al., 2010; Wang et al., 2014).

This study is an effort contributing to answering the quantitative issues related to the aforementioned management problems by estimating the components of the water budget of the UBN basin using a new hydrological modelling framework (see section 3.1) and remote sensing data. It obtains, at relatively small spatial scales and at daily time step, groundwater storage, evapotranspiration, discharges in such a way to satisfy the water budget equation. It is also a methodological paper, in that it

delineates various methodologies to overcome the data scarcity. The paper is organized as follows: firstly, descriptions of the study area is given (section 2), then the methodologies for each water budget component and the model set-up are detailed in section 3. The results and discussions of each component and the water budget are presented in section 4. Finally, the conclusions of the study are given (section 5).

## 2   The Study Basin

The Upper Blue Nile (UBN) river originates at Lake Tana at Bahir Dar, flowing southeast through a series of cataracts. After about 150 km, the river enters to a deep canyon, and changes direction to the south. After flowing for another 120 km, the river again changes its direction to the west and northwest, towards the El Diem (Ethiopia-Sudan border). Many tributaries draining from many parts of the Ethiopian highlands join the main river along its course. The total length of the river within Ethiopia is about 1000 km.

The UBN basin represents up to 60% of the Ethiopian highlands contribution to the Nile river flows, which is itself 85% of the total (Abu-Zeid and Biswas, 1996; Conway, 2000). The area of the river basin enclosed by a section at the Ethiopia-Sudan border is about 175,315 km$^2$ (figure 1), covering about 17% of the total area of Ethiopia. The large scale hydrological behaviour of the basin is described in a series of studies (Conway, 1997, 2000, 2005; Conway and Hulme, 1993). Specifically, its hydrological behaviour is characterized by high spatio-temporal variability. Since the UBN basin gives the largest contribution

to the total Nile flow, it is the economic mainstay of downstream countries (i.e. Sudan and Egypt). Moreover, the Ethiopian highlands are highly populated and have high water demands for irrigation and domestic uses on their own.

The maps of elevation of the basin is shown in figure 1. The topography of UBN is very complex, with elevation ranging from 500 m in the lowlands at the Sudan border to 4160 m in the upper parts of the basin. Due to the topographic variations, the climate of the basin varies from cool (in the highlands) to hot (in the lowlands), with large variations in a limited elevation

range. The hot season is from March to May, the wet season, with lower temperatures, is from June to September, while the dry season runs from October to February. The mean annual rainfall and potential evapotranspiration of the UBN basin are estimated to be in the ranges of 1,200-1,600 mm and 1,000-1,800 mm, respectively (Conway, 1997, 2000), with high spatio-temporal variability. The annual temperature mean is 18.5$^o$C, with small seasonal variability.

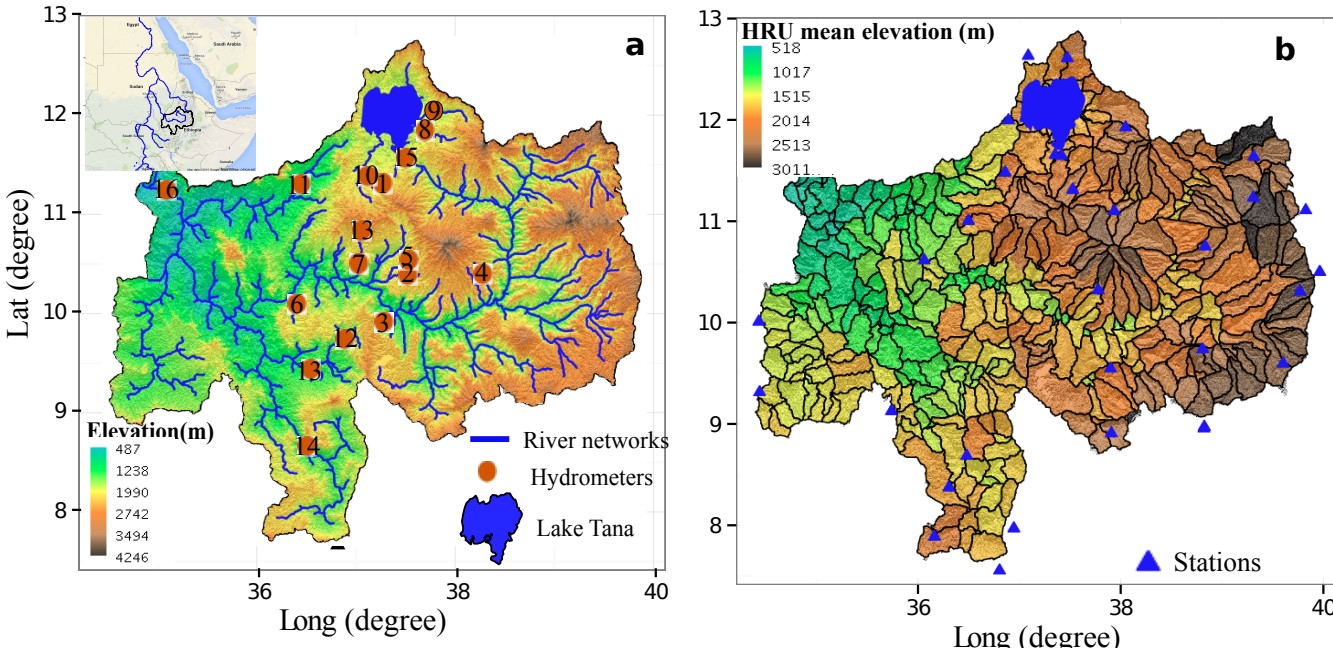

**Figure 1.** The Upper Blue Nile basin digital elevation map, along with the gauge stations present in the basin. Numbers inside the circles designate the river gauging stations whose names are provided in table (a); and subbasin partitions and meteorological stations used for simulation (b).

## 3  Methodology

Water budget simulation is essential to the estimation of both water storage and water fluxes (rate of flow) for given, appropriate, control volumes and time periods. It is given by:

$$\frac{\partial S_k(t)}{\partial t} = J_k(t) + \sum_{i}^{m(k)} Q_{ki}(t) - ET_k(t) - Q_k(t) \tag{1}$$

where $J(t)$ is rainfall, and $ET(t)$ is actual evapotranspiration, $Q(t)$ is discharge, $Q_{ki}(t)$ are inflows from upstream HRUs. The index $k = 1, 2, 3...$ is the control volume where the water budget is solved. In our case, the control volume is a portion of the basin (a subbasin) derived from topographic partitioning as described in section 3.1.

### 3.1  JGrass-NewAge system set-up

UBN water budget is estimated using the JGrass-NewAge hydrological system, which is, in turn, based on the Object Modelling System framework (David et al., 2013). It is a set of modelling components, reported in table 1, that can be connected at runtime to create various modelling solutions. Each component is presented in details and tested against measured data in the corresponding papers cited in the table 1. A similar study using JGrass-NewAge system, but utilizing mostly in-situ observa-

**Table 1.** JGrass-NewAge system components and respective references. The components in bold are the ones used in this study.

| Role | Component Name | Description |
|---|---|---|
| Basin partitioning | **GIS spatial toolbox and Horton Machine** | A GIS spatial toolbox that uses DEM to extract basin, hillslopes, and channel links for NewAge-JGrass set-up (Formetta et al., 2014a; Abera et al., 2014). |
| Data interpolation | Kriging and Inverse Distance Weighting | Interpolates meteorological data from meteorological stations to points of interest according to a variety of kriging algorithms (Goovaerts, 2000; Haberlandt, 2007; Goovaerts, 1999; Schiemann et al., 2011), and Inverse Distance Weighting (Goovaerts, 1997) |
| Energy balance | **Shortwave radiation**, Longwave radiation | Calculate shortwave and longwave radiation, respectively, from topographic and atmospheric data (Formetta et al., 2013, 2016). |
| Evapotranspiration | Penman-Monteith, **Priestly-Taylor**, Fao-Evapotranspiration | Estimates evapotranspiration using Penman-Monteith (Monteith et al., 1965), Priestly-Taylor (Priestley and Taylor, 1972), and Fao-Evapotranspiration (Allen et al., 1998) options |
| Runoff | ADIGE (**Hymod**) | Estimates runoff based on Hymod (Moore, 1985) algorithm (Formetta et al., 2011) described in Appendix A |
| Snow melting | Snow melt | Modelling snow melting using three types of temperature and radiation based algorithms (Formetta et al., 2014b) |
| Optimization | **Particle Swarm Optimization**, DREAM, **LUCA** | Calibrate model parameters according to Particle Swarm Optimization (Kennedy and Eberhart, 1995), DREAM (Vrugt et al., 2009), LUCA (Hay et al., 2006) algorithms respectively. |

tions, has been conducted in Posina river basin, northeast Italy (Abera et al., 2017). A brief descriptions of the components used in this study are provided in the following sections. In this study, the shortwave solar radiation budget component (section 3.3), the evapotranspiration component (Priestley and Taylor, section 3.3), the Adige rainfall-runoff model (section 3.4), and all the components illustrated in figure 2 are used to estimate the various hydrological flows.

5     A necessary step for spatial hydrological modelling is the partitioning of the topographic information into an appropriate spatial scale. The SRTM 90 m X 90 m elevation data is used to generate the basin Geographic Information System (GIS) representation. The basin topographic representation in GIS, as detailed in Abera et al. (2014); Formetta et al. (2011), is based on the Pfafstetter enumeration. The basin is subdivided in Hydrologic Response Units (HRUs), where the model inputs (i.e. meteorological forcing data), and hydrological processes and outputs (i.e. evapotranspiration, discharge, shortwave solar

10    radiation) are averaged (Formetta et al., 2014a). A routing scheme is applied to move the discharges from HRUs to the basin outlet through the channel network is included in the Adige component.

    In this study, the UBN basin is divided into 402 subbasins (HRUs of mean area of $430 \pm 339$ km$^2$) and channel links, as shown in figure 1b. This spatial partitioning may not be the finest scale possible, however it is consistent with input data

resolution, including satellite products, meaning that a finer subdivision would imply uniform inputs for adjacent HRUs, and a coarser one would average out inputs variability. In this paper, the term HRU actually identifies subbasin.

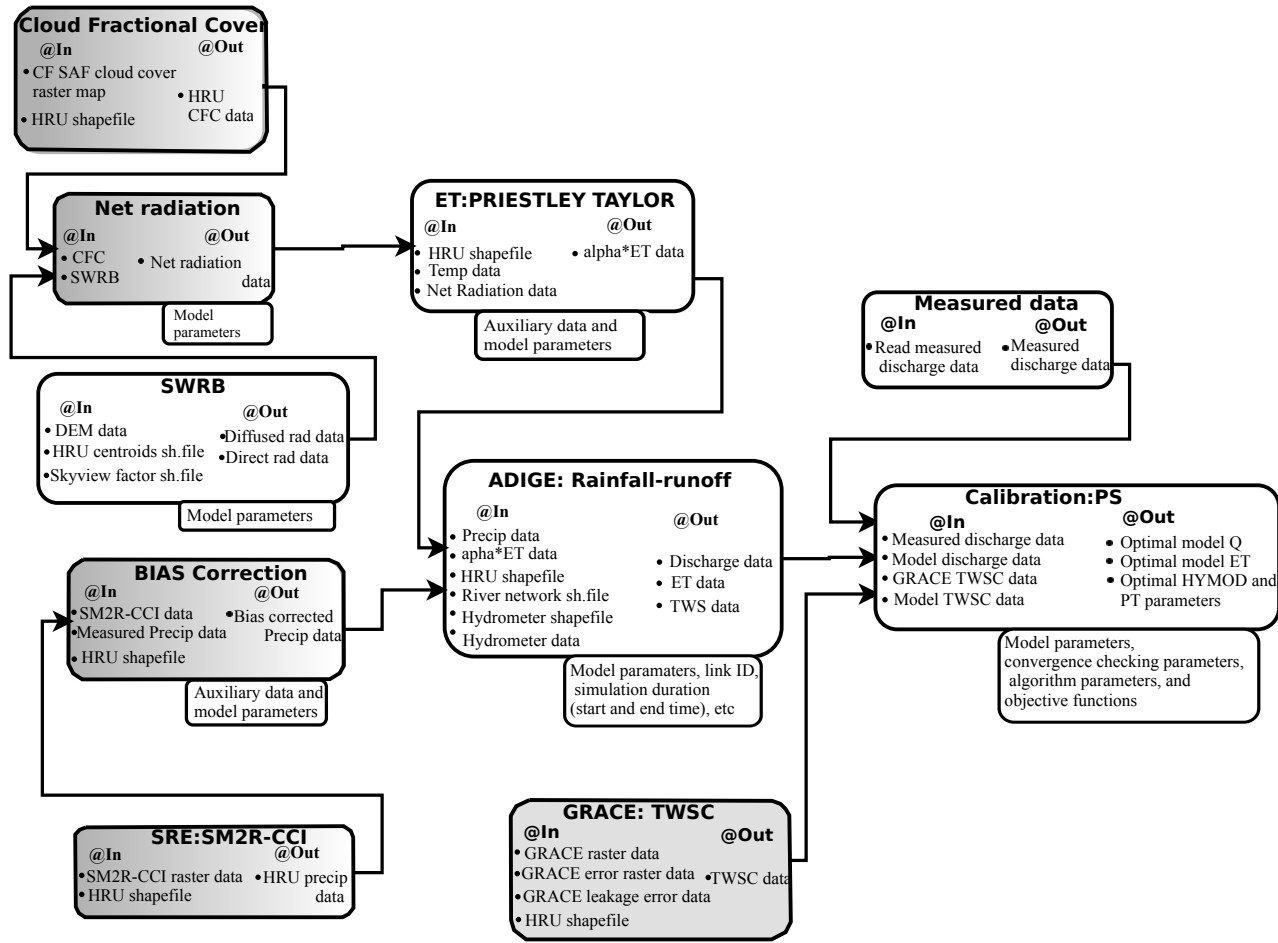

**Figure 2.** Workflow with a list of NewAge components (in white), and remote sensing data processing parts (shaded in grey, not yet included in JGrass-NewAGE and currently performed with R tools) used to derive the water budget of the UBN. It does not include the components used for the validation and verification processes.

## 3.2 Precipitation $J(t)$

The spatio-temporal precipitation input term of Eq. 1 (J(t)), is quantified with RS-based approaches. Currently, there are several satellite rainfall estimates (SREs) available for free and Abera et al. (2016) compared five of them with high spatial and temporal resolutions over the same basin. It was shown that SM2R-CCI (Brocca et al., 2013, 2014) is one of the best products, particularly in capturing the total rainfall volume. Regards to the quality of SM2RAIN-based products, recent studies positively assessed their accuracy on a regional (Brocca et al., 2016; Ciabatta et al., 2017) and a global (Koster et al., 2016)

scale. A comparative analysis of the effects of different SREs on basin water budget components is an interesting area of research, however, here only SM2R-CCI is used for obtaining the precipitation input. The systematic error (bias) of SM2R-CCI is removed according to the ecdf matching techniques by Michelangeli et al. (2009) and specialised for UBN byAbera et al. (2016) by using in-situ observations. The subbasin mean precipitation is estimated by averaging all the pixels RS corrected data within each subbasin. In accordance with the basin partition described in section 3.1, the 1994-2009 daily precipitation set is generated for each of the 402 subbasins.

## 3.3 Evapotranspiration $ET$

Evapotranspiration estimation is crucial for agricultural and water resources management as it is an important flux within a basin. The lack of in-situ data relating to $ET$ impedes modelling efforts and makes it probably the most difficult task in water budget assessment. Here, $ET$ is estimated according to the NewAge specific component. It provides estimates at any temporal and spatial resolution required by using the Priestley and Taylor (PT) Formula (Priestley and Taylor, 1972), which is one of the more common models used. PT is mainly based on net radiation estimation, $Rn$, grouping all the unknowns into the $\alpha_{PT}$ coefficient, as shown in Eq. 2.

$$PET = \alpha_{PT} \frac{\Delta}{\Delta + \gamma} Rn \tag{2}$$

$PET$ is Priestley and Taylor potential evapotranspiration, $\Delta$ is the slope of the Clausius-Clapeyron relation and $\gamma$ is the psychometric constant (Brutsaert, 2005). In this study, however, we need an estimate of the actual evapotranspiration ($ET$), which is constrained not only by the atmospheric demands as in (Eq. 2), but it uses storage information which can be obtained from the ADIGE rainfall-runoff component of JGrass-NewAge. Hence, the $ET$ equation is modified as (Abera et al., 2017):

$$ET(t) = \alpha_{PT} \frac{S(t)}{S_{max}} \frac{\Delta}{\Delta + \gamma} Rn \tag{3}$$

where $S(t)$ is the groundwater storage, and $S_{max}$ the maximum storage capacity for each HRU. The important unknown coefficient $\alpha_{PT}$ (Pejam et al., 2006; Assouline et al., 2016) and the $S_{max}$ are calibrated within the rainfall-runoff model component, as explained below. The ratio $S(t)/S_{max}$ represents a stress coefficient which became very popular since the work of Feddes and coworkers (Feddes et al., 2001).

In our procedure, given that $S(t)$ is not measured, the assumption that there is null water storage difference after a long time, named Budyko's time, $T_B$, (Budyko, 1978), is required. So, here, what is searched is a time duration ($T_B$) such that the water storage assumes again its initial value (Abera et al., 2017). Once $T_B$ is fixed, the tools for automatic calibration, provided by the Object Modelling System, produce the set of parameters in tab 4, including $\alpha_{PT}$ and $S_{max}$, for which discharge is well reproduced and is also $S(T_B) = S(0)$. In this study, $T_B = 6$ years.

In equation (3), $Rn$ is the main input modulating the atmospheric demand component of ET. To this scope, the NewAge shortwave radiation budget component, SWRB (Formetta et al., 2013), is used to return a value for each subbasin in clear sky

conditions. Irradiance in clear sky conditions, however, is unsuitable for all sky condition since surface shortwave radiation is strongly affected by cloud cover and cloud type (Arking, 1991; Kjærsgaard et al., 2009). Therefore, the clear sky SWRB estimated using NewAge-SWRB is cut by using the cloud fractional cover (CFC) satellite data set (Karlsson et al., 2013), processed and provided by EUMETSAT Climate Monitoring Satellite Application Facility (CM SAF) project (Schulz et al.,

2009). In this case net radiation is generated only from the shortwave radiation and the cloud cover data, as in the following formulation (Kim and Hogue, 2008):

$$Rn = (1 - CFC)R_S \tag{4}$$

Where $R_S$ is the net shortwave radiation and $Rn$ is the net radiation. The daily CFC data originates from polar orbiting satellites, version CDRV001, using a daily temporal resolution and a $0.25^o$ spatial resolution from 1994 to 2009 (16 years).

Satellite data are processed (Karlsson et al., 2013) to obtain the mean daily CFC for each subbasin. In comparison to CFC, the effects of surface albedo on $Rn$ is minimal, particularly in highland areas with vegetation and no snow cover such as the UBN basin.

Once $ET$ is estimated according to the methods described, it is useful to compare it with independently obtained $ET$ estimates or data. In situ $ET$ observations are not present for this basin, as is the case for most regions. Estimates of $ET$ based

on RS have been made available by different algorithms (Norman et al., 1995; Mu et al., 2007; Jarmain, 2009; Fisher et al., 2008). In this study, the Global Land Evaporation Amsterdam Methodology (GLEAM, version_v3_BETA) (Miralles et al., 2011a; Martens et al., 2016), a global, satellite-based, $ET$ data set is used. GLEAM, as well NewAge, uses the PT scheme for estimating ET. However, all inputs of the formula, in GLEAM and NewAge, are evaluated according to different strategies and RS tools. GLEAM sets $\alpha_{PT} = 0.98$ while in NewAge it has been calibrated. In GLEAM PET is additively increased, by

intercepted rainfall estimated according to a version of the Gash model (Gash, 1979), and multiplicatively decreased by a stress coefficient depending on five soil cover types (bare soil, snow, tall vegetation, two levels of low vegetation) and has a different expression for anyone of the storages. Moreover, according to the case, the stress coefficient is evaluated through various RS, according to procedures which are described in the paper by Martens et al. (2016). Differently from the NewAge approach, GLEAM also considers dynamic vegetation information to estimate the stress factor (Miralles et al., 2011a).

GLEAM is available at $0.25^o$ spatial resolution ( 28 km of side or  800 square kilometers of area) and daily temporal resolution, and assessed positively in different studies (McCabe et al., 2016; Miralles et al., 2011b). The most recent version of GLEAM was validated globally over sixty-four Fluxnet sites (Martens et al., 2016) with consistent results, letting us guess that it behaves properly also in Ethiopia. The differences between NewAGE estimation and GLEAM's one allow to assume that the our results and their results can be seen as largely independent . For comparison with NewAge ET, we averaged GLEAM ET for

each HRU polygon. Comparison of the NewAge $ET$ with MODIS standard $ET$ product is also available in the supplementary material of the paper.

### 3.4 Discharge $Q$

For discharge estimation, the ADIGE rainfall-runoff component is used. It is based on the well-known HYMOD model (Moore, 1985) as runoff production component which also include a routing component and artificial inflows-outflows management. Detailed descriptions of HYMOD implementations in the NewAge model system are given in Formetta et al. (2011); Abera
et al. (2017) and summarized in Appendix A. The main inputs for the ADIGE model are $J(t)$ and $PET(t)$, as estimated in the previous sections. The NewAge Hymod component is applied to any HRU, in which the basin is subdivided and the total watershed discharge is the sum of the contribution of each of the 402 HRU routed to the outlet. The ADIGE rainfall-runoff has five calibration parameters ($C_{max}$, $B_{exp}$, $\alpha_{Hymod}$, $R_s$, $R_q$, see the details in Appendix A), and the calibration is performed using the particle swarm (PS) optimization method. PS is a population-based stochastic optimization technique (Kennedy and
Eberhart, 1995). The objective function used to estimate the optimal value of the parameter is the Kling-Gupta efficiency (KGE, Kling et al. (2012)). The KGE is preferred to the commonly-used Nash-Sutcliffe efficiency (NSE, Nash and Sutcliffe (1970)) because the NSE has been criticized for its overestimation of model skill for highly seasonal variables by underestimating flow variability (Schaefli and Gupta, 2007; Gupta et al., 2009). For evaluation of the model performances, in addition to the KGE, the two other goodness-of-fit (GOF) methods (percentage bias (PBIAS) and correlation coefficient) used in this study
are described in Appendix B.

### 3.5 Total water storage change $ds/dt$

The $ds/dt$ in Eq. 1 is the water contained in the ground, soil, snow and ice, lakes and rivers, and biomass. It is the total water storage (TWS) change, calculated as the residuals of the water budget fluxes for each control volume. In this paper, the $ds/dt$ estimation at daily time steps is based on the interplay of all the other components as presented in Eq. 1. There is no way to
estimate areal TWS from in situ observations. The new Gravity Recovery and Climate Experiment (GRACE) data (Landerer and Swenson, 2012) has a potential to estimate this component, but at very low spatial and temporal resolutions. At large scale, however, it can still be used for constraining and validating data of the modelling solutions. Here, the performance of our modelling approach to close the water budget is assessed using the GRACE estimation at the basin scale. Monthly GRACE data is obtained from NASA's Jet Propulsion Laboratory (JPL) ftp:// podaac-ftp.jpl.nasa.gov/allData/tellus/L3/land mass/RL05.
The leakage errors and scaling factor (Landerer and Swenson, 2012) that are provided with the product are applied to improve the data before the comparison is made. The total error of GRACE estimation is a combination of GRACE measurement and leakage errors (Billah et al., 2015). Based on the data of these two error types, the mean monthly error of GRACE in estimating total water storage change (TWSC) in the basin is about 8.2 mm. Since the other fluxes, for instance Q and ET, are modelled as functions of basin water storage, the good estimation of water storage by a model affects the goodness of fit of all the other
fluxes as well (Döll et al., 2014).

## 3.6 Calibration and validation approach

The satellite precipitation data set (SM2R-CCI) is error corrected based on in situ observations. At the basin outlet (Ethiopia-Sudan Border), the ADIGE rainfall-runoff component (i.e. HYMOD model) is calibrated to fit the observed discharge during the six years of calibration period (1994-1999) at daily time steps. Based on the approach described in section 3.2, $\alpha_{PT}$ is calibrated by by imposing that $S(0) = S(T_B)$ after $T_B = 6$ years. The value of six years is arbitrary but it was found to give good agreement with GRACE data (see below), so no other values were used. The simulation for each hydrological component is then verified using available in-situ or remote sensing data (Table 2), and three goodness-of-fit (KGE, PBIAS, r) are used as comparative indices (for detail information please see Appendex B), as follows:

- Discharge validation: Discharge simulation is validated at the outlet close to the Ethiopian-Sudan border, where the model is calibrated. In addition, the simulation of NewAge at the internal links is validated in 15 discharge measurement stations, where in situ data are available. The evaluations of discharges at the internal links provide an assessment of model estimation capacity at ungauged locations.

- ET comparison: Once ET is estimated according to the procedures described above, GLEAM (Miralles et al., 2011a) is used as an independent data set to assess ET estimation. After GLEAM is aggregated for each subbasin, the GLEAM and the NewAge ET are compared and the goodness-of-fit (GOF) indexes are calculated, based on 16 years of data (1994-2009).

- $ds/dt$ validation: The water storage change, $ds/dt$, estimated as residual of the water budget, is validated against the GRACE based data-set. To harmonize and enable comparison between the model and the GRACE TWS data, it is necessary to do both time and spatial filtering. Following the GRACE TWSc temporal resolution, the model $ds/dt$ is aggregated at monthly time steps and at the whole basin scale.

**Table 2.** Short summary of the list of remote sensing products used in this study.

| Satellite products | Spatial resolution | Temporal resolution | Data used | Reference | used as |
|---|---|---|---|---|---|
| SM2R-CCI | 0.25 degree | daily | 1994-2009 | Brocca et al. (2014, 2013); Abera et al. (2016) | input for Precipitation |
| GLEAM | 0.25 degree | daily | 1994-2009 | Miralles et al. (2011a); McCabe et al. (2016) | verification for evapotranspiration |
| MODIS ET (MOD16) | 1-km | 8-days | 1994-2009 | Mu et al. (2007, 2011) | verification for evapotranspiration |
| GRACE TWS | 1 degree | 30-days | 2003-2009 | Landerer and Swenson (2012) | Verification for storage change |
| CM-SAF | 0.25 degree | daily | 1994-2009 | Schulz et al. (2009) | input for evapotranspiration component |

## 4 Results and Discussion

The results of the study are organized as follows: firstly, we present the results for 1) precipitation, 2) evapotranspiration, 3) discharge and 4) total water storage; secondly, the JGrass-NewAGE system is used to resolve the water budget closure at each subbasin, and the contribution of each term water budget term is further is analyzed.

### 4.1 Precipitation $J$

The spatial distribution of mean, long-term, annual precipitation is presented in figure 3a. Generally, precipitation increases from the east (about 1000 mm/year) to the south and southwest (1800 mm/year). This spatial pattern is consistent with the results of Mellander et al. (2013) and Abtew et al. (2009). SM2R-CCI shows that the south and southwest parts of the basin receive higher precipitation than the east and northeast parts of the highlands. The rainiest subbasins are in the southern part of the basin.For this location the precipitation data used correspond to a mean annual rainfall of about 1900 mm, while the mean annual precipitation reported for this region by Abtew et al. (2009) is about 2049 mm. The latter estimation, however, is from point gauge data, while this study is based on areal data. To understand the spatial distribution of the seasonal cycle, the quarterly percentage of total annual precipitation, calculated from 1994 to 2009 in daily estimations, is presented in figure 3 b. During the summer season (June, July and August), while the subbasins in the north and northeast receive about 65% of annual precipitation (figure 3 b), the subbasins in the south receive about 40% of total precipitation.

### 4.2 Evapotranspiration $ET$

ET is estimated for each subbasin at daily time steps. In this section We provide mainly discussion about the comparison of NewAge ET and RS estimates, but further comments on ET can be found in section 4.5 which show that ET is mostly water limited than energy limited. Figure 4 a shows the comparisons of the $ET$ time series from 1994-2002 (aggregated at daily, weekly, and monthly, from top to bottom) between NewAge and GLEAM. The figure specifically refers to three selected subbasins representing different ranges of elevations and spatial locations. NewAge estimates have higher temporal variability in comparison to GLEAM. In the represented locations, GLEAM therefore accumulates a systematic growing difference in evapotranspired water volume, which could be not consistent with the estimated storage (see below).

The agreement/disagreement between the two $ET$ estimations vary from subbasin to subbasin (figure 4). The spatial distribution correlation and PBIAS between the NewAge and GLEAM ET is presented in figure 4b. Spatially, the correlation between JGrass-NewAGE and GLEAM is higher in the eastern and central parts of the basin, while it tends to decrease systematically towards the west (i.e. to the lowlands, see figure 4b). The correlation between the two ET estimations increases when passing from daily to monthly time steps. The PBIAS between the two estimates ranges from -10% to 10%, with large numbers of subbasin being from -3% to 3%. Spatially, the comparison shows that GLEAM overestimates ET in the western parts of the basin (border to the Sudan) and underestimates ET in the northern parts of the basin (figure 4b). The overall basin correlation is $0.34 \pm 0.07$ (daily time step), $0.51 \pm 0.08$ (weekly time step), and $0.57 \pm 0.10$ (monthly time steps). Generally, except at daily time step, the two estimates have acceptable agreements (very low bias, and acceptable correlation). In comparison with

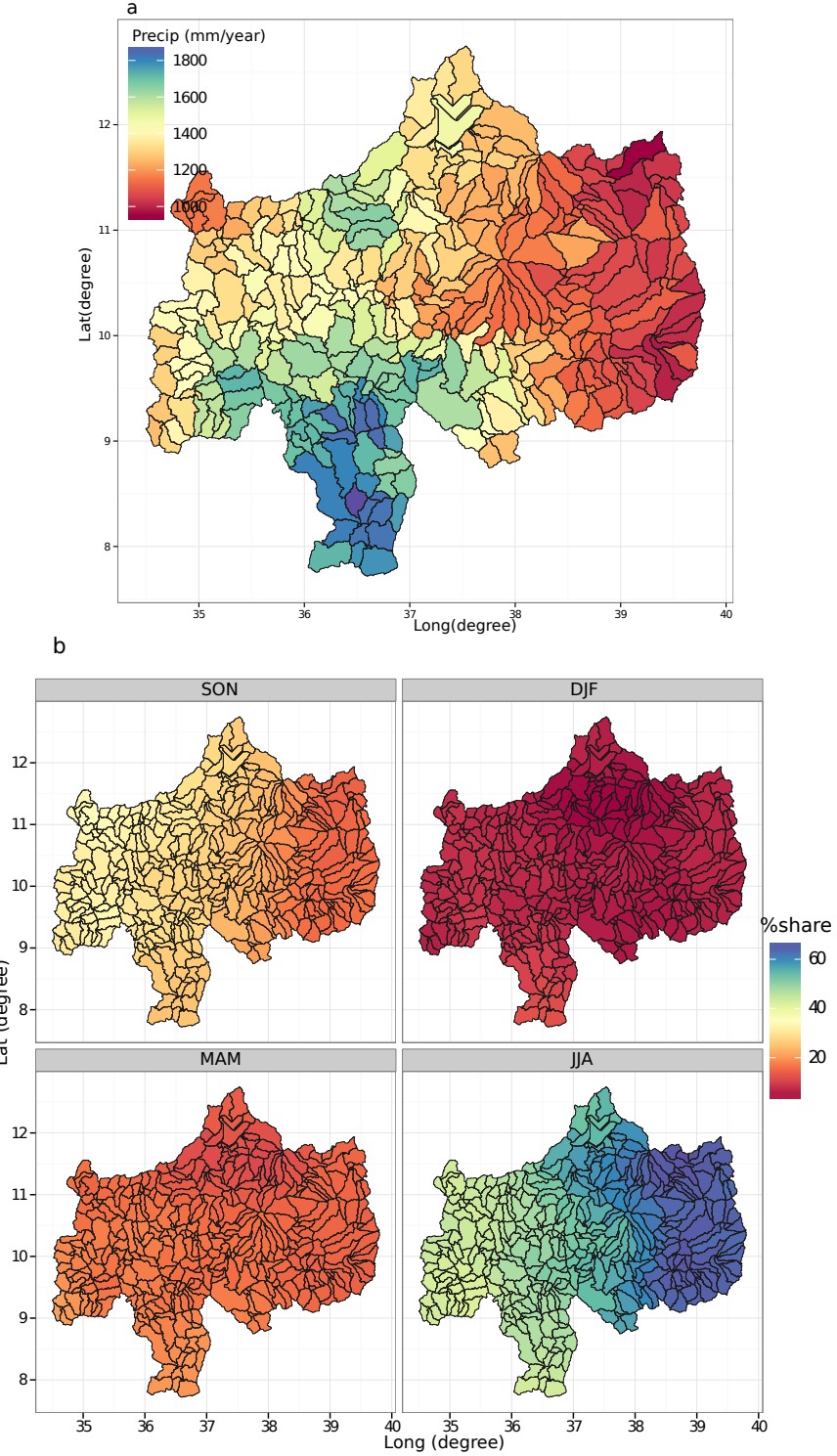

**Figure 3.** The spatial distribution of mean annual rainfall (a), and quarterly percentages share of the total rainfall (b) estimated from long term data (1994-2009): SON (September, October, and November), DJF (December, January, and February), MAM (March, April, and May), JJA (June, July and August). Note that high seasonality is observed in the eastern part of the basin.

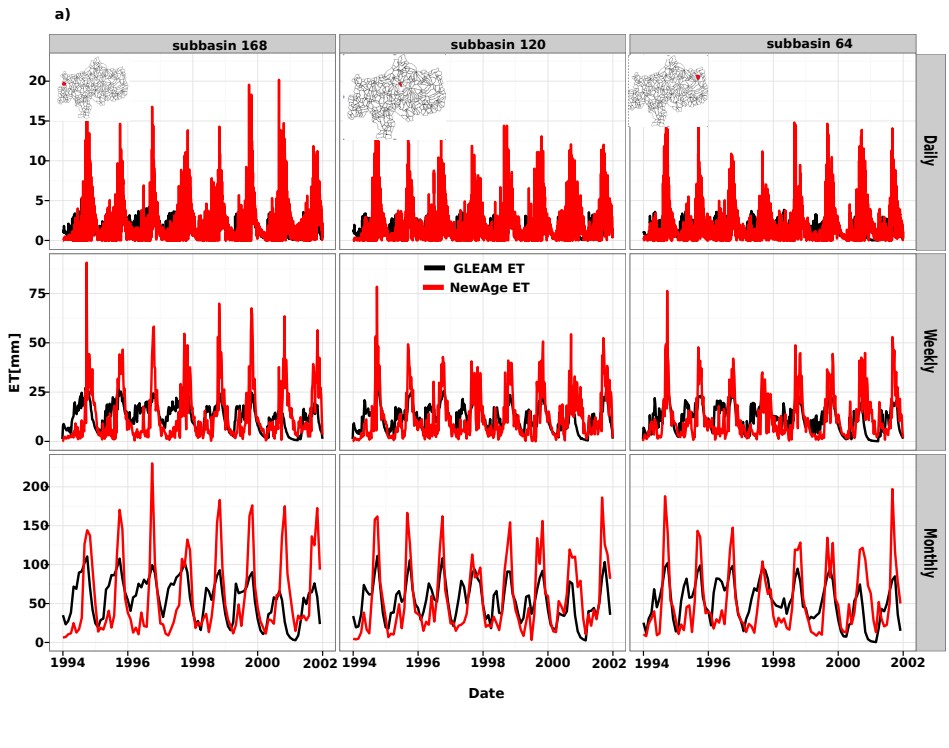

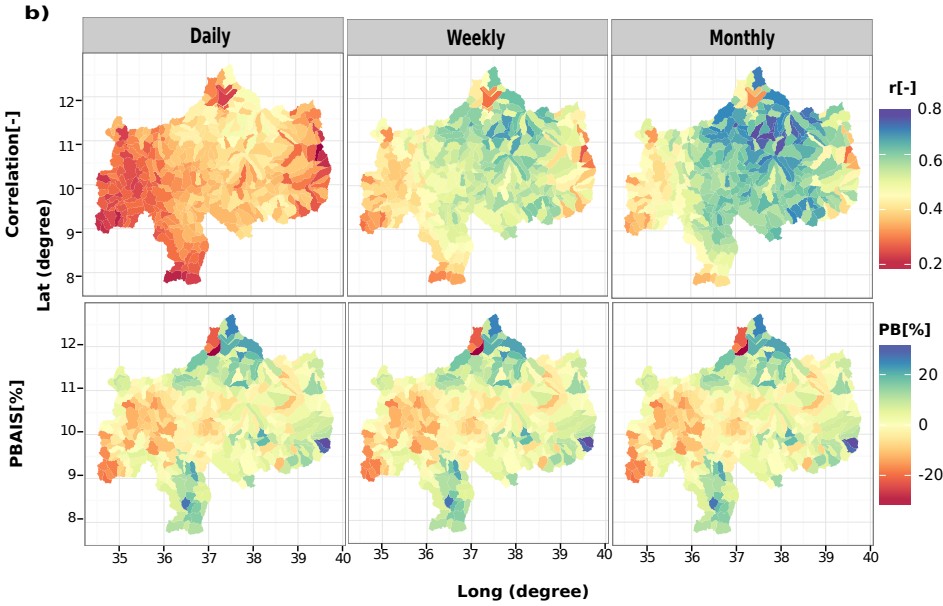

**Figure 4.** a:Time series ET estimation with NewAge and GLEAM for three subbasins: subbasin ID168 , subbasin ID120, and subbasin ID64 at daily, weekly and monthly time steps. The locations of the subbasins are indicated on the maps at the top of each column of plots. b: spatial distribution of correlation coefficient and PBIAS between NewAge and GLEAM estimations at daily, weekly and monthly time steps. A more detailed reading of the Figure is made in the main text.

the correlation (0.48 ± 0.15) and PBIAS (14.5 ± 18.9%) obtained between NewAge ET and MODIS ET Product (MOD16), as shown in the supplementary material, the correlation and PBIAS between NewAge ET and GLEAM ET is much better.

## 4.3 Discharge $Q$

The optimized parameters of the Adige model, obtained using automatic calibration procedure of NewAge, are given at table 3. At the basin outlet, the automatic calibration of the NewAge components provided very good values of the GOF indices (KGE=0.93, PBIAS = 2.2, r = 0.94). The performances, at the outlet remain high also during the validation period, having KGE=0.92, PBIAS = 2.4, and r = 0.93. Model performances are also evaluated within the basin at the internal catchments outlets (table 4) where stage measurements are available. Figure 5 shows simulated hydrographs along with the observed discharges for some locations. The results show that the performances of the NewAge simulation are a little better than the performances reported by Mengistu and Sorteberg (2012), with slightly lower PBIAS value (PBIAS=8.2, r=0.95). Generally, the model predicts both the high flows and low flows well, with slight underestimation of peak flows (figure 5 a). This is likely due to the underestimation of SM2R-CCI precipitation data for high rainfall intensities (Abera et al., 2016). Additional source of error can also be caused by model inconsistency due to averaging out input data over large areas or from some inadequacy in stage-discharge curves used to obtain discharges from water levels. The slight underestimation of runoff could result from the overestimation of evapotranspiration. However, in this case, GLEAM (or MODIS) would cause larger discrepancies.

**Table 3.** Optimized parameters obtained from daily ADIGE simulation during the calibration period (1994-1999). Parameters' physical meaning is explained in Appendix A. The last parameter is for the ET component.

| Parameters | value |
|---|---|
| $C_{max}[L]$ | 694.18 |
| $B_{exp}[-]$ | 0.64 |
| $\alpha_{Hymod}[-]$ | 0.61 |
| $Rs[T]$ | 0.086 |
| $Rq[T]$ | 0.394 |
| $\alpha_{PT}[-]$ | 2.9 |

Regarding the internal sites discharge simulation, we remark some representative results. The hydrograph comparison between the NewAge simulated discharge and the observed one of the Gelgel Beles river, enclosed at the bridge near to Mandura with an area of 675 km$^2$, is shown in figure 5 b. The performance of the uncalibrated NewAge at Gelegel Beles has a correlation coefficient of 0.70, PBIAS is 11.40% and the KGE value is 0.68 (table 4).

Simulation performances for the medium size basins, such as the Ribb river, enclosed at Addis Zemen (area=1592 km$^2$, KGE = 0.81, PBIAS = 12% and $r$ = 0.82, figure 5 c), and Gilgel Abay river, enclosed at Merawi (area = 1664 km$^2$, KGE=0.81, PBIAS=12%, r=0.93), are very good. For the Ribb river, the NewAge simulation performance can be compared with SWAT Model performances by Setegn et al. (2008) (r=0.74-0.76). Even though SWAT was calibrated for this specific subbasin, the results of our study are much better. Similarly, without calibration for the Gilgel Abay river, the NewAge simulation

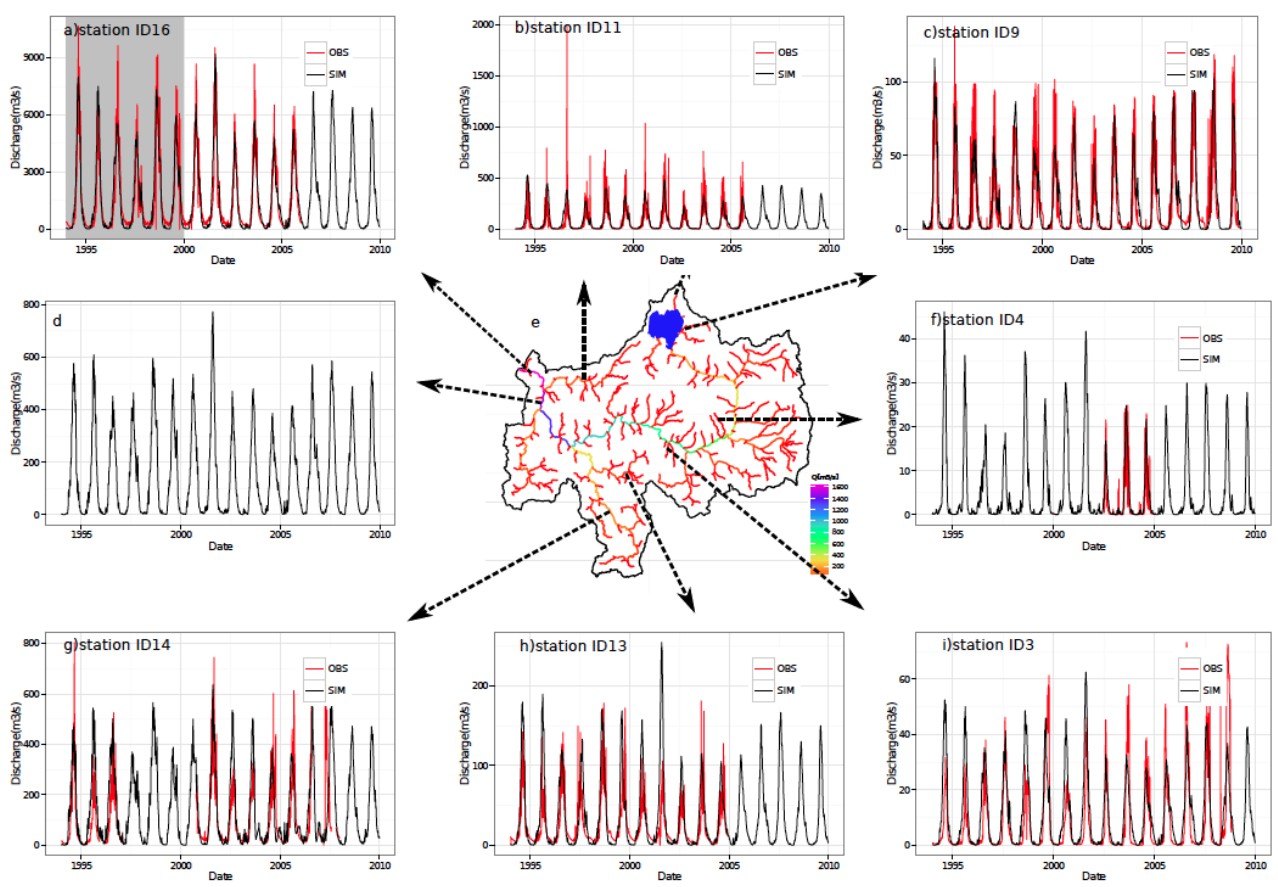

**Figure 5.** NewAge model simulation validation at internal subbasins. The model calibrated (shown by gray shaded period) and validated at El Diem (a) is used to estimate at each channel link and, where discharge measurements are available, they are verified: main Beles bridge (b), Ribb river enclosed at Addis Zemen (c), just simulation of the main Blue Nile before joining Beles river (d), Jedeb near Amanuel (f), Dedisa river basin enclosed near Arjo (g), Angar river basin enclosed near Nekemt (h), and Nesh near Shambu (i). Figure (e) shows the long term estimated daily discharge for all river links of the basin.

**Table 4.** The simulation capacity of the NewAge Adige rainfall-runoff component at the internal sites, based on the optimized parameters calibrated at the outlet. The performance at the outlet (El Diem) is the model performance during validation period.

| Hydrometer stations ID | River Name | Area (km$^2$) | KGE | PBIAS | r |
| --- | --- | --- | --- | --- | --- |
| 1 | Koga @ Merawi | 244.00 | 0.67 | -8.70 | 0.73 |
| 2 | Jedeb @ Amanuel | 305.00 | 0.38 | 40.80 | 0.53 |
| 3 | Neshi @ Shambu | 322.00 | 0.58 | 32.00 | 0.57 |
| 4 | Suha @ Bichena | 359.00 | 0.54 | 39.20 | 0.82 |
| 5 | Temcha @ Dembecha | 406.00 | 0.70 | 3.30 | 0.71 |
| 6 | Gilgel Beles @ Mandura | 675.00 | 0.68 | 11.40 | 0.70 |
| 7 | Lower Fettam @ Galibed | 757.00 | 0.67 | -7.7 | 0.78 |
| 8 | Gummera @ Bahir Dar | 1394.00 | 0.19 | -53.20 | 0.88 |
| 9 | Ribb @ Addis Zemen | 1592.00 | 0.81 | 12.00 | 0.86 |
| 10 | Gelgel Abay @ Merawi | 1664.00 | 0.81 | 12.00 | 0.93 |
| 11 | Main Beles @ Bridge | 3431.00 | 0.68 | -1.70 | 0.74 |
| 12 | Little Anger @ Gutin | 3742.00 | 0.65 | 24.30 | 0.81 |
| 13 | Great Anger @ Nekemt | 4674.00 | 0.72 | -14.10 | 0.82 |
| 14 | Didessa @ Arjo | 9981.00 | 0.55 | 19.60 | 0.81 |
| 15 | Upper Blue Nile @ Bahir Dar | 15321.00 | 0.26 | 5.10 | 0.60 |
| 16 | Upper Blue Nile @ El Diem | 174000.00 | 0.92 | 2.40 | 0.93 |

performance is better than the results of Wase-Tana (Wosenie et al., 2014, PBIAS=34)) and Flex$_B$ (Fenicia et al., 2008, PBIAS=77.6) or comparable to SWAT (PBIAS=5).

To analyze the simulation capacity of NewAge for the larger size basins, the performances at Angar river (area 4674 km$^2$), Lake Tana (area 15321 km$^2$), and Dedisa river basin (9981 km$^2$) are reported. The simulation analysis at the Angar river

5    enclosed near Nekemt (KGE = 0.72, PBIAS = -14.10%, and r = 0.82), Lake Tana (KGE = 0.26, PBIAS = 5.10, and r = 0.60), and Dedisa (KGE=0.55, PBIAS = 19.60, and r = 0.81) indicate that the performances are acceptable. The comparison of simulated and observed discharges, as well as the locations of the Angar (basin brief description (Easton et al., 2010)) and Dedisa rivers are shown in figure 5, in plots h and g respectively.

For most subbasins, because of the good model performances (i.e. KGE is higher than 0.5 and PBIAS is within 20%), the

10    estimated discharges are deemed adequate for estimating water resource at locations where gauges are unavailable. The model is also able to reproduce discharge across the range of scales. For instance, the model performances at the Ethiopia-Sudan border (175 315 km$^2$), Dedisa near Arjo (9981 km$^2$), main Beles (3431 km$^2$), and Temcha near Dembecha (406 km$^2$) are also acceptable. An exception is Lake Tana, where the discharge is regulated (figure 5 and table 4). Model performance varies with basins and a consistent behavior with respect to basin size, climate, vegetation density and topographic complexity is not

15    found. Indeed, there are many factors that affect the model performance, including uncertainties in input observations. Sample simulations at all the channel links of the study basin at daily time step are provided in the supplementary material.

## 4.4   Total water storage change

NewAge simulated $ds/dt$ for 16 years for each subbasin is calculated as a residual of the flux terms. The simulated $ds/dt$ is represented and compared with the GRACE-based TWSC in figure 6. The storage change shows high seasonality over the basin, with positive change in summer and negative change in winter. The change varies from -100 to +120 mm/month. The model $ds/dt$, aggregated at monthly time scale and for the whole basin, is in accordance with the GRACE TWSC both in temporal pattern and amplitude The good correlation coefficient of 0.84 and the general good performances of the $ds/dt$ component is certainly caused also by the ability of NewAge to well reproduce the other water fluxes. Due to the possible high leakage error introduced in GRACE TWSC at high spatial resolutions  (Swenson and Wahr, 2006), statistical comparison at subbasin level is not performed.However, the spatial distribution of NewAge and GRACE $ds/dt$ estimates can be found in the supplementary material.

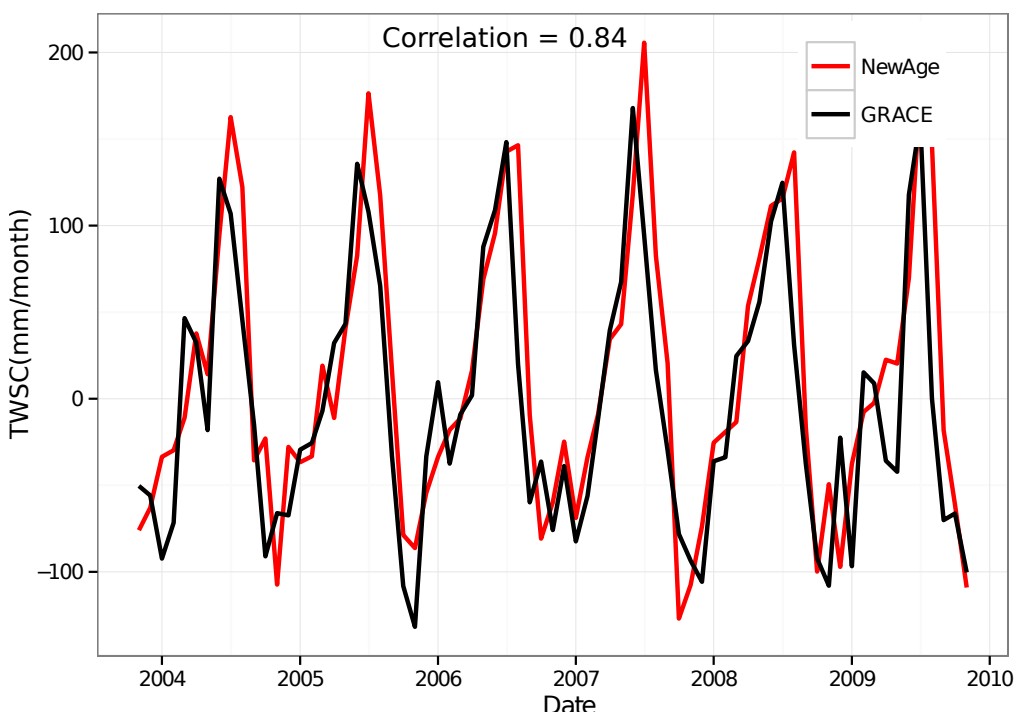

**Figure 6.** Comparison between basin-scale (whole UBN, 176,315 square kilometers) NewAge $ds/dt$ and GRACE TWSC from 2004-2009 at monthly time steps.

## 4.5   Water budget closure

The water budget components (J, ET, Q, $ds/dt$) of 402 subbasin of the UBN are simulated for the period 1994-2009 at daily time steps. Figure 7 shows the long-term, monthly-mean, water budget closure derived from 1994-2009. The four months (January, April, July, and October) are selected to show the four seasons (Winter, Spring, Summer and Autumn). For all

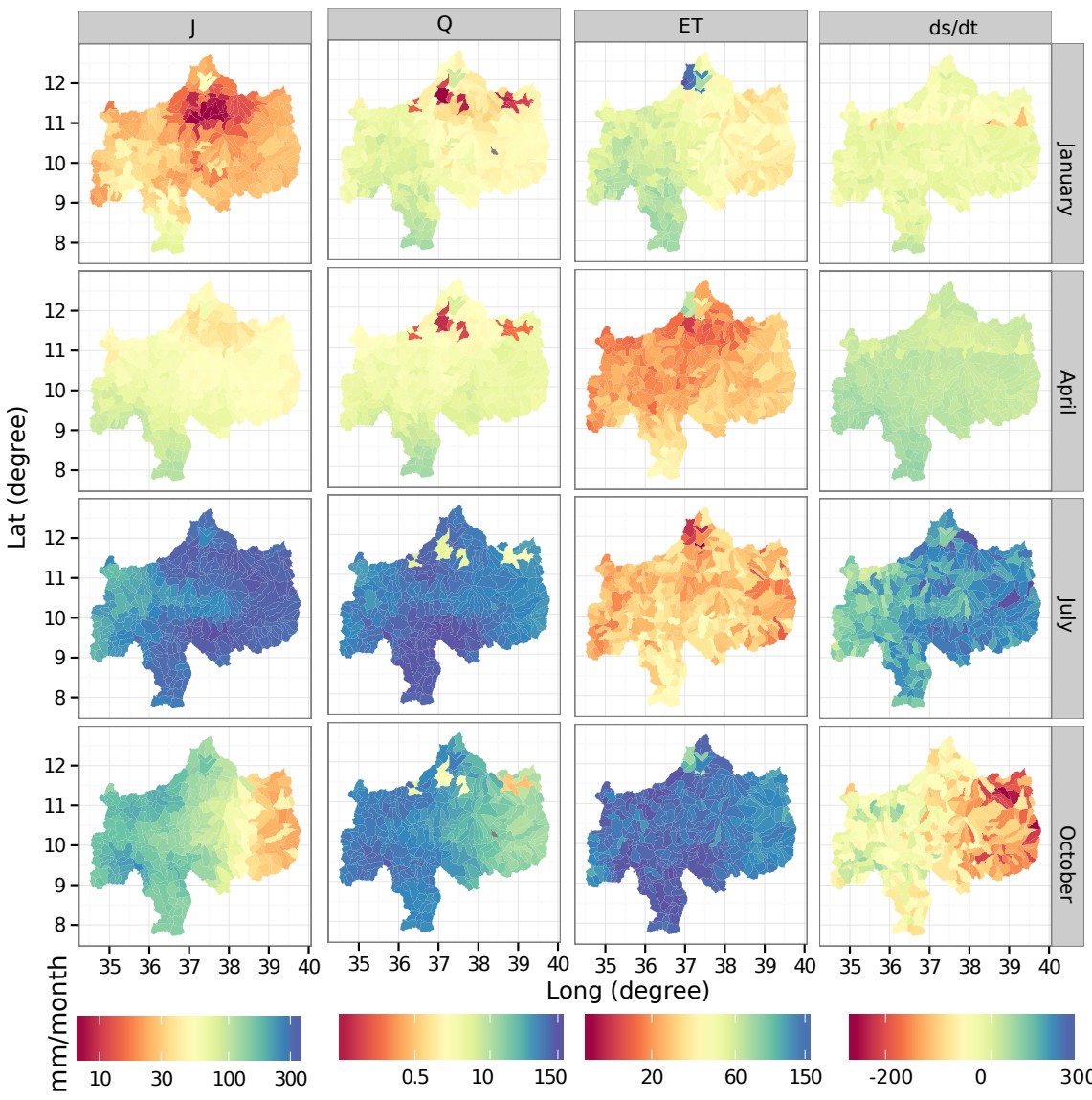

**Figure 7.** Spatial distribution of long term mean monthly water budget (January, April, July and October) in the UBN basin. For the sake of visibility, the legend is plotted separately and on logarithmic scale, except for the storage component.

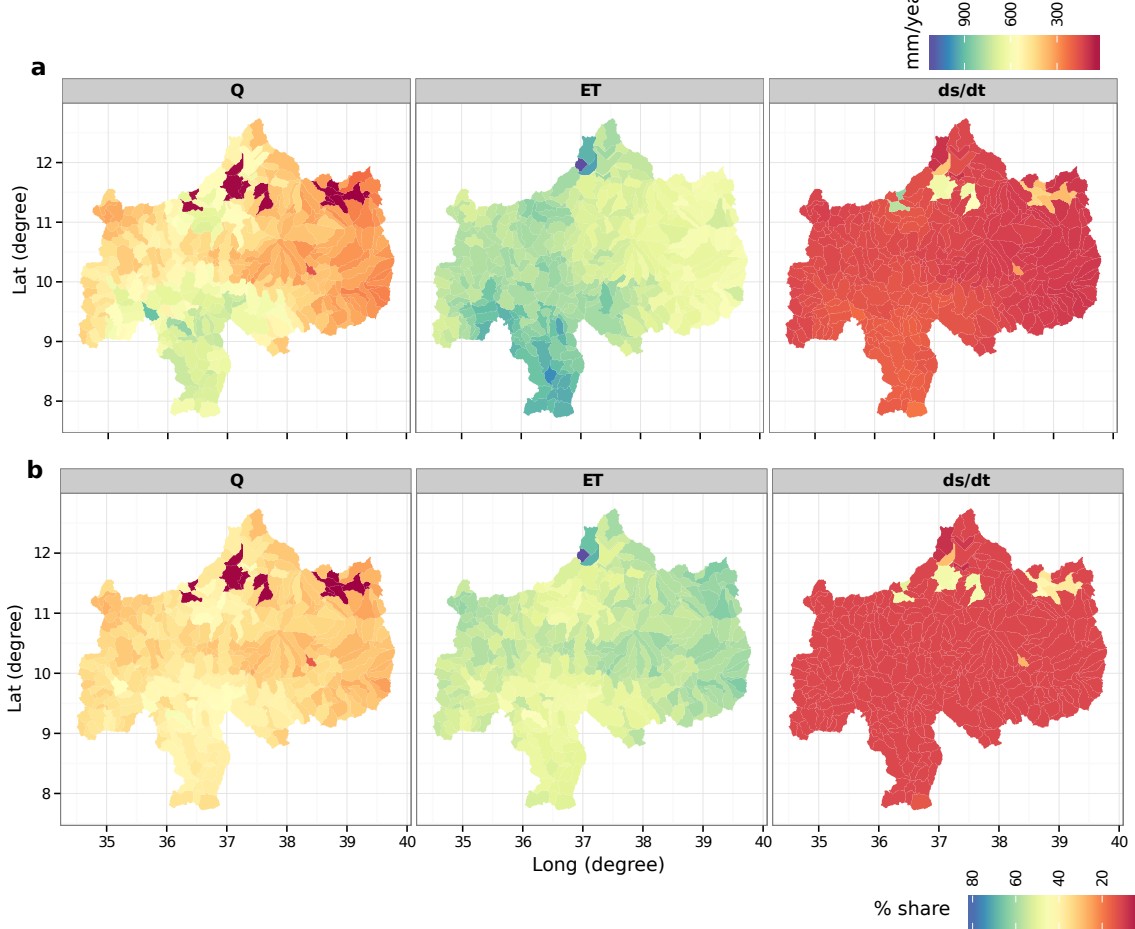

**Figure 8.** The spatial distributions of long term mean annual water budget closure: precipitation in mm (figure 3), the output terms (Q, ET, ds/dt) in mm (a), and the percentage share of the output term (Q, ET, ds/dt) of the total precipitation (b).

components, the mean seasonal variability is very high. Generally, the seasonal patterns of Q and $ds/dt$ follow the J, showing the highest values in summer (i.e. July) and the lowest in winter (i.e. January). However, simulated ET shows distinct seasonal patterns with respect to the other components, the highest being during autumn (October), followed by winter (January). During the summer it is low, most likely due to high cloud cover.

5      The variability between the subbasins is also appreciable. Generally, all water budget components tends to increase from the east to the southwest part of the basin, except for the summer season (July). During summer, on the other hand, the eastern part of the basin receives its highest rainfall, stores more water, and generates high runoff as well. In general the dominant budget component varies with months. For instance, in January ET is the dominant while in June and July $ds/dt$ is more dominant. After the summer season, Q and ET are the dominant fluxes. A regression analysis based on the results for all subbasins and all

years shows that, at short time scales such as at daily or monthly, the variability in ET is not due to variablity in J ($R^2$=0.01). Conversely, at the yearly time scale, 78% of ET variance is explained by variability in J.

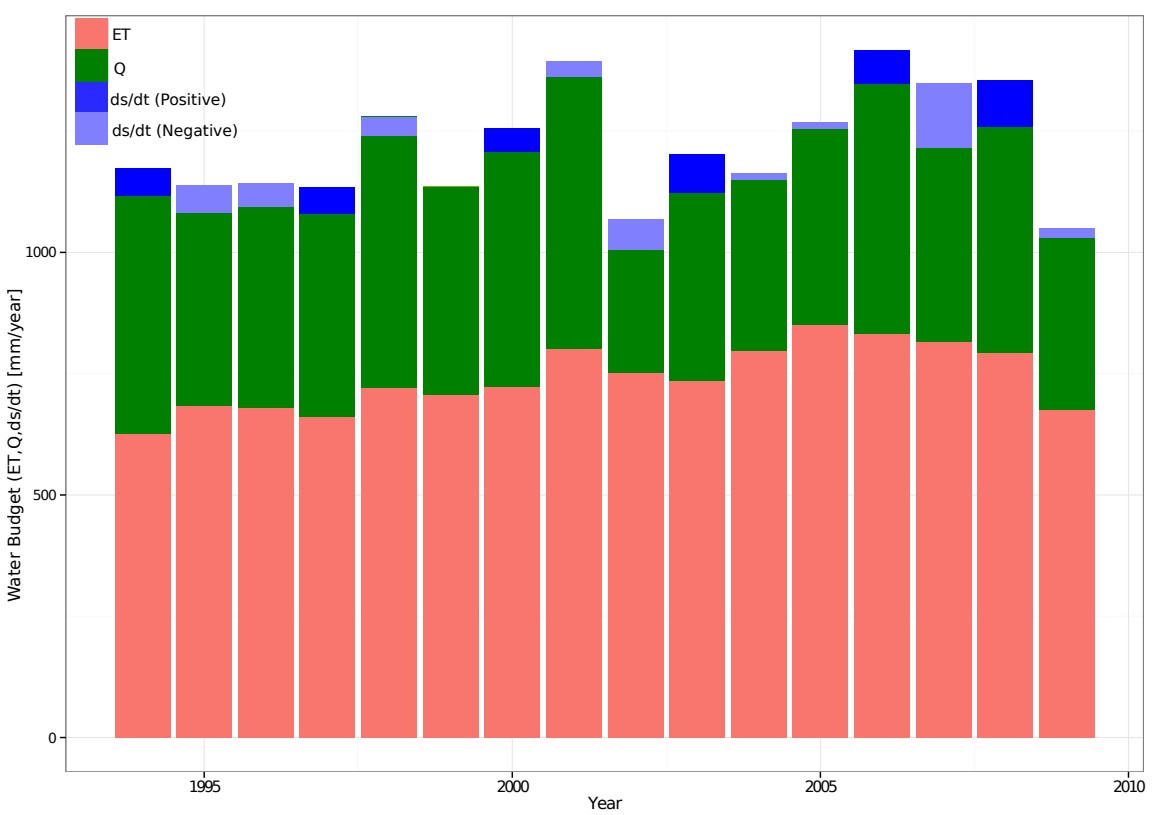

**Figure 9.** Water budget components of the basin and its annual variabilities from 1994 to 2009. The relative share of each of the three components (Q, ET and ds/dt) of the total available water J is represented by the length of the bars (N.B. the total length of the bar minus the negative storage is J). The positive and negative storage of the years are shown by dark blue and light blue respectively).

The spatial variability of the long term mean annual water budget closure is shown in figure 8. The spatial variability for J and Q is higher than $ds/dt$ and ET. The higher Q and ET in the southern and southwestern part of the basin are due to higher J. Similarly Q is lower in the eastern and northeastern part of the basin. Focusing on the percentage share of the output term (Q, ET, ds/dt) of total J (figure 8 c), ET dominates the water budget, followed by Q. It is noteworthy that the eastern subbasins with low ET still have percentage share of ET due to low amount of J received.

The long-term basin-average water budget components shows: 1360 ± 230 mm of J, followed by 740 ± 87 mm of ET, 454 ± 160 mm of Q and -4 ± 63 mm of $ds/dt$. While the spatial variability of the water budget is high, the annual variability is rather limited. Higher annual variability is observed for J, followed by Q. 2001 and 2006 are wet years, characterized by high J and Q. Conversely, 2002 and 2009 are dry years with 1167 mm and 1215 mm per year of precipitation. Details on the two dry years (2002, 2009) of the region can be read in Viste et al. (2013).

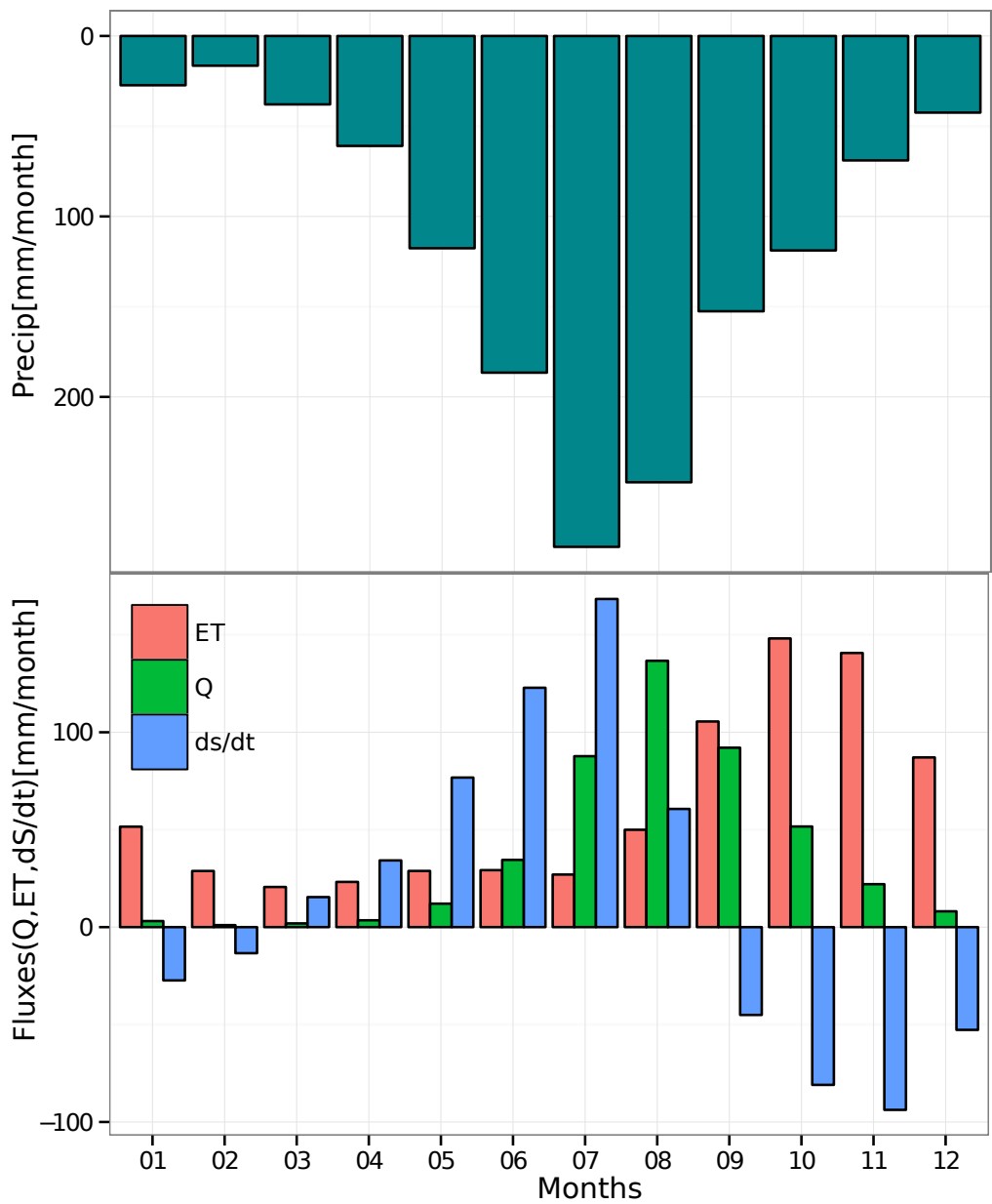

**Figure 10.** Monthly mean Water budget components at basin scale and long term, based on estimates from 1994 to 2009. The relative shares of the three components (Q, ET and dS/dt) of the total available water J are shown .

Figure 10 provides long term monthly mean estimates of water budget fluxes and storage. The average basin scale budget is highly variable. The highest variability is mainly in J and $ds/dt$. During summer months, J, Q, and $ds/dt$ shows high magnitude. ET is not high in June, July and August, but in October and December. The $S(t)$ accumulated in the summer season feeds high ET in autumn, and causes very high drops in $ds/dt$ (figure 10). The seasonal trend between J and ET is slightly out-of-phase, i.e., the highest energy to evaporate water occurs during low precipitation months (March, April, May). Due to this slight out-of-phase, ET is minimal and Q and $ds/dt$ is enhanced during wet months (figure 10) thus revealing that ET is water limited more than energy limited. The same Figure also shows the complex interplay between discharges, (variation of) storages and evapotranspiration. A first look at Figure 4 and 5 could bring to the conclusion that overestimation of ET brings in underestimation of Q. However, Figure 10 shows that the role of $ds/dt$ is not negligible at all.

## 5    Conclusions

The goal of this study is to estimate the whole water budget and its spatial and temporal variability of the upper Blue Nile basin using the JGrass-NewAge hydrological system and remote sensing data. The study covered 16 years from 1994-2009 at a finer spatial and temporal resolution than in previous studies. In order to achieve this result, we used various remote sensing products, rainfall from SM2R-CCI, cloud cover from SAF EUMETSAT CFC, evapotranspiration from GLEAM and MODIS (used for comparison), and storage change from GRACE (also used for comparison). We also used all the ground data currently available, i.e. sixteen discharge time series, and thirty-five ground based meteorological stations. The results can be summarized as follows:

– The basin scale annual precipitation over the basin is 1360 ± 230 mm, and highly variable spatially. The southern and southwestern parts of the basin receive the highest precipitation, which tend to decrease towards the eastern parts of the basin (figure 3).

– Generally, the interannual variability of ET is high, and tends to be higher in autumn and lower in summer. The average basin scale ET is about 740 ± 87 mm, and is the larger flux in water budget in the basin.

– The comparison of simulated ET with the satellite product GLEAM shows that GLEAM has low temporal variability than our estimates. The correlation between GLEAM ET and NewAge ET increases from daily time steps to monthly time steps, and spatially it is higher in the east and central parts of the basin. Comparison with MODIS products was also performed (reported in supplementary material). MODIS actually shows even more large departure from JGrass-NewAge results. Both satellite products, however, seem to introduce a systematic bias which would not allow to close the water budget according both simulated and measured discharges.

– The NewAge ADIGE rainfall-runoff component is able to reproduce discharge very well at the outlet (KGE = 0.92). The long term annual runoff of the UBN basin is about 454 ± 160 mm. The verification results at the internal sites where measurements are available reveal that the model can be used for forecasting at ungauged locations with some success.

- The performances obtained are promising (figures 5 and 6, and table 4) and often greatly improve previous results.

  The NewAge storage estimations and their space-time variability are effectively verified by the basin scale GRACE TWSC data which show high correlation and similar amplitude.

Despite the good results obtained, it is important to note that this study is limited by the lack of in-situ ET observation and low resolution GRACE data for confirmation of storage. To these regards, the results of this study would benefit from basin specific assessments of ET and $ds/dt$ RS products based on ground measurements, as done in Abera et al. (2016) for precipitation. We claim that the procedure we followed can be easily transported in any other poorly gauged basin with benefits for the hydrological knowledge of any region on Earth.

### Reproducibility

The forcing data used for NewAge simulation: SM2R-CCI is obtained from http://hydrology.irpi.cnr.it/people/l.brocca; the rain gauge precipitation and hydrometer discharge data were obtained from the National meteorological Agency and Ministry of Water and Energy of Ethiopia respectively, and it can be requested for research. The remote sensing data used for comparison: GLEAMS ET, MODIS ET and GRACE TWSC are freely available and can be downloaded at http://www.gleam.eu,http://www.ntsg.umt.edu/project/mod16, and ftp://podaac-ftp.jpl.nasa.gov/allData/tellus/L3/landmass/RL05 respectively. Modelling components used for the simulations are available and documented through the Geoframe blog http://geoframe.blogspot.com. Additional data (i.e. GIS database, topographic information, input data and additional results) and other notes regarding the paper can be found at Zenodo DOI:10.5281/zenodo.264004

*Acknowledgements.* This research has been partially financed by the CLIMAWARE projects of University of Trento (/http://abouthydrology.blogspot.it/search/label/CLIMAWARE) and by European Union FP7 Collaborative Project GLOBAQUA (Managing the effects of multiple stressors on aquatic ecosystems under water scarcity, grant no. 603629-ENV-2013.6.2.1). We would like to acknowledge the National meteorological Agency and Ministry of Water and Energy of Ethiopia for providing us the gauge rainfall and discharge data. We also thank the two anonymous reviewer for their work that helped to enhance the initial manuscript with their comments.

### Appendix A: Hymod model in NewAge-JGrass system

The NewAge system executes one Hymod model at each HRU, and routes water downslope. Detailed description of Hymod model is provided in many researches (Moore, 1985; Van Delft et al., 2009; Boyle et al., 2001; Formetta et al., 2011). In Hymod, each HRU, is supposed to be a composition of storages of capability $C$ [L] according to distribution (Moore, 1985):

$$F(C < c) = 1 - (1 - \frac{c}{C_{max}})^{B_{exp}} \tag{A1}$$

where $F(C)$ represents the cumulative probability of a certain water storage capacity ($C$); $C_{max}$ is the largest water storage capacity within each hillslope and $B_{exp}$ is the degree of variability in the storage capacity. As shown in the schematic diagram (figure 11), the precipitation exceeding $C_{max}$ is send directly to the volume available for surface runoff. If we call the precipitation volume in a time interval $\Delta t$, $J(t) := P(t)\Delta t$, then this "direct" runoff can be estimated according to:

$$R_H(t) = \max(0, J(t) + C(t) - C_{max}) \tag{A2}$$

where $C(t)$ defines the fraction of storages already filled at time $t$. The latter equation is true for any precipitation and storage level, even when the maximum storage $C_{max}$ is not exceeded. When precipitation does not exceeds $C_{max}$ runoff volume can be produced by filling some of the smaller storages. To which extent this happens, can be derived by the knowledge of the storage distribution, eq. (A1), the initial storage $C(t)$ and the precipitation $J(t)$. This residual runoff is, in fact, given by:

$$R(t) = \int\limits_{C(t)}^{\min(C(t)+J(t),c_{max})} F(c)\, dc \tag{A3}$$

An analytic expression for the integral in eq. (A3) is available, which makes the computation easier. Water in storage is made available to evapotranspiration. Water going into runoff the runoff volume, i.e. $R(t)$ and $R_H(t)$, is further subdivided into a surface runoff volume and subsurface storm runoff. Surface runoff, in turn, is composed by the whole of $R_H(t)$ and part of $R(t)$, and $R(t)$ is split according to a partition coefficient $\alpha$ such that the part $\alpha R(t)$ goes into surface runoff volume and $(1-\alpha)$ into the subsurface storm runoff volume. In Hymod, $\alpha$ is a calibration coefficient.

Finally, surface runoff volumes are routed through three linear reservoirs, and, subsurface storm runoff volume is routed through a single linear reservoir. A summary of equations for the surface runoff is therefore:

$$\frac{dS_1(t)}{dt} = \alpha R(t) + R_H(t) - kS_1(t) \qquad\qquad Q_1(t) = \frac{S_1(t)}{k} \tag{A4}$$

where $S_1$ [L$^3$] is the storage in the first of the linear reservoirs, and $k$ [T] is the mean residence time in each of the reservoirs. Then:

$$\frac{dS_i(t)}{dt} = Q_{i-1}(t) - kS_i(t) \quad Q_i(t) = \frac{S_i(t)}{k} \tag{A5}$$

for the other two reservoirs, where $S_i$ [L] with $i = 2,3$ is the storage in the two remaining surface reservoirs. Subsurface storm runoff is then modeled by:

$$\frac{dS_{sub}}{dt} = (1-\alpha)R(t) - k_{sub}S_{sub}(t) \tag{A6}$$

where $S_{sub}$ [L$^3$] is the storage in the subsurface storm-flow system and $k_{sub}$ [T] is its mean residence time. A budget equation can be written for the groundwater system as:

$$\frac{dS_g(t)}{dt} = (J(t) - R(t) - R_H(t)) - ET(t) - Q_g(t) \tag{A7}$$

where $S_g(t)$ [L$^3$] is the groundwater storage, and $Q_g(t)$ the groundwater flow which becomes surface flow at the closure of the HRU.

Summarizing, Hymod subdivides each HRU into three reservoirs: a groundwater reservoir, from where evapotranspiration and groundwater flow is allowed, a subsurface storm-water reservoir, and a surface runoff reservoirs set. Partition of precipitation into the three reservoirs is obtained by a calibration coefficient, $\alpha$, and the use of a probability distribution function of storages' capacity, $F(c)$.

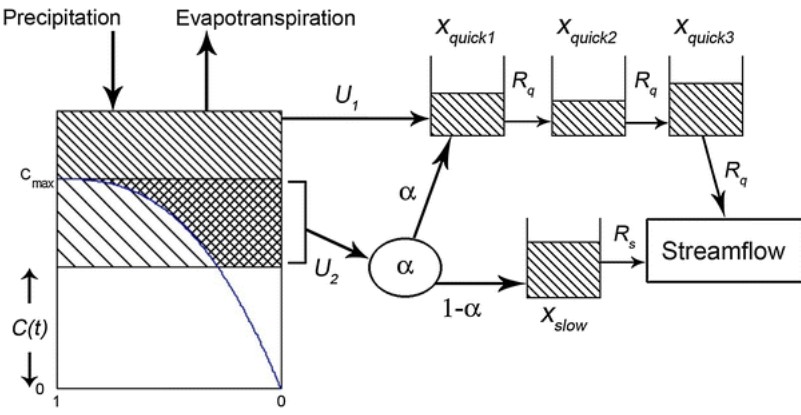

**Figure 11.** Schematic diagram of hymod model (adapted from Van Delft et al. (2009))

## Appendix B: Model performance criteria

The model evaluation statistics used in the paper are the goodness-of-fit (GOF) indices. The following indexes are used as objective function and comparison of estimations.

1. PBIAS: is the measure of average tendency of estimated values to be large or smaller that their measured values. The value near to zero indicates high estimation, whereas the positive value indicates the overestimation and negative values indicate model underestimation (Moriasi et al., 2007; Gupta et al., 1999).

$$PBIAS = \frac{\sum_{i=1}^{n}(P_i - O_i)}{\sum_{i=1}^{n}O_i} 100 \tag{B1}$$

The PBIAS value ranges from -20 to 20% is considered good, and values between $\pm20\%$ and $\pm40\%$ and those greater than $\pm40\%$ are considered satisfactory and unsatisfactory respectively (Stehr et al., 2008).

2. Kling-Gupta efficiency (KGE) is developed by Gupta et al. (2009) to provide a diagnostically interesting decomposition of the Nash-Sutcliffe efficiency (and hence MSE), which facilitates the analysis of the relative importance of its different

components (correlation, bias and variability) in the context of hydrological modelling. Kling et al. (2012) proposed a revised version of this index. It is given by

$$KGE = 1 - ED \tag{B2}$$

$$ED = \sqrt{(r-1)^2 + (vr-1)^2 + (\beta-1)^2} \tag{B3}$$

5    where ED is the Euclidian distance from the ideal point, $\beta$ is the ratio between the mean simulated and mean observed flows, $r$ is Pearson product-moment correlation coefficient, and $v$ is the ratio between the observed ($\sigma_o$) and modelled ($\sigma_s$) standard deviations of the time series and takes account of the relative variability (Zambrano-Bigiarini, 2013). The KGE ranges from infinity to a perfect estimation of 1, but a performance above 0.75 and 0.5 is considered as good and intermediate respectively (Thiemig et al., 2013).

10    3. Pearson correlation coefficient (r): please refer Moriasi et al. (2007). The correlation coefficient is best as much as it is close to 1.

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
