# Peer review of "Modelling the water budget of the Upper Blue Nile basin using the JGrass-NewAge model system and satellite data"

_Hydrology and Earth System Sciences, 2016_

## Referee Comment (RC1) · Anonymous Referee #1 · 20 Sep 2016

This manuscript proposes a method to improve water budget modelling by using the available, but sparse, hydrometerological data and satellite products. The current manuscript provides a good try to predict hydrological process in data-scarce regions or ungauged basins. Although there are publications related to such topic in ungauged basins, the intent of the manuscript is worthy and significant, and is of interest to readers of HESS. Seeing the potential of this study, I am in general supportive of publication if the following comments are addressed in the resubmission.

My major concerns are:

1. I would encourage the authors to rewrite the methodology section. Give a clear message to the reader what you did and how you did. For example, the manuscript entitled as 'JGrass-NewAge model system'. However, I could not find detail or key information about the method. What's the theory of the method based on? What's the advantage of the method? The headings in method section are the same as those in section 5. Some parts in the results analysis and discussion section are more suitable to be in the methodology section. For instance, it would be better to introduce the indices (i.e., KGE, PBIAS, r) in section 4. In addition, what's the spatial resolution of the HRU? When performing simulation, what are the time step and the spatial resolution of output? There are different hydrometerological data and satellite products, but it is difficult to readers to obtain their information (e.g., what kind of satellite products). I would suggest the authors providing a table to show all the data and their spatiotemporal resolutions. How did you deal with the different resolutions (especially spatial resolution) of input parameters?

2. Discussion should be enhanced. What's the disadvantage of the method when applying in data-scarce regions with large area? For example, results of figure 5 indicated that the simulated runoffs were underestimated. What's the reason? Was it caused by uncertainties/errors in precipitation products? I could not find any quantitative information about errors of SM2R-CCI. Meteorological stations should observe precipitation, radiation, and etc. Why didn't you use them for

validation and discussion?

3. The authors claimed that the JGrass-NewAGE system are described in a series of papers and not re-discussed in this manuscript. What's the difference between this study and the previous papers? What's the main contribution of this work?

My specific comments are given below.

The numbers in front of the comments indicate page and line number.

1. 1-21. 'up to 2000 mm per year'. It would be much clearer by adding precipitation.

2. 3-1. It should have space between 'given' and '('. The authors should proof read the manuscript to avoid such mistakes.

3. 3-6. 'the river enters a deep a canyon' contains grammatical errors.

4. 3-18. The elevation values show certain difference compared to those in page 2 line 3.

5. 3-30. It may mislead to conclude 'the seasonal variability of the basin is very high' because the authors claimed that the temperature has small seasonal variability.

6. 4-1. Figure 1. I suggest adding units for axes (also other figures) as well as enlarging the schematic map (at least the text). What does the color represent in figure 1b?

7. 4-15. It seems that the citation appeared in the first time, and 2014b should change to 2014a. The authors should proof read the manuscript to avoid such mistakes.

8. 5-4. What does GIS mean? Please consider defining the abbreviation.

9. 5-9. How did you divide the basin into 402 subbasins? According to what kind of rules? I'm not sure whether figure 1b is your results or not.

10. 5-13. Figure 2 is difficult to read. The texts were small and difficult to guess their meaning. I suggest the authors redraw it.

11. 6-23. Works cited in a manuscript should be accepted for publication or published already. There are many publications describing psychometric constant.

12. 6-27. What's the relation between S(t) and TB in equation 3? Can you explain

more?

13. 7-26. Semicolon should be replaced with 'and'.

14. 8-4. What does KGE mean? Please consider defining the abbreviation.

15. 8-8. What does 'described in A' mean? Does 'A' represent 'Appendix'?

16. 9-18. It is curious to use J representing precipitation. In addition, precipitation, evapotranspiration, and discharge are components of water budget. Why did you use different section headings (i.e., 5.1, 5.1.1, 5.1.2, …)?

17. 9-21. I would suggest the authors adding 'the Oromia region (or other mentioned places)' into Fig.1.

18. 10-1. Figure 3a indicates precipitation is highest in southern region. However, figure 3b showed a different pattern (i.e., east shared highest precipitation), especially in JJA.

19. 11-4. How and why did you select only some subbasins? Did you consider r and PBAIS (figure 4, e.g., high r and low PBAIS, and low r but high PBAIS)?

20. 11-10. 'while the it tends to' contains grammatical errors.

21. 11-23. 'within the basin at the internal channels (2)'. What does '(2)' mean?

22. 11-27. I do not think $r^2$=0.92 is lower than r=0.93 or r=0.94. I suggest the authors to unify the index.

23. 13-1. Are all the parameters unitless? Why are two $\alpha[-]$? Furthermore, I could not find table 1 in the context.

24. 13-2. Can you number the hydrometer stations and then add these IDs into figures 1b and 5?

25. 14-8. Are Wase-Tana and FlexB commonly used models? Please consider defining the abbreviation.

26. 18-5. Can you provide some radiation, cloud, and wind observations? This may be better to draw the conclusion.

27. 19-9. What does S mean?

28. 19-11. The number of decimal places was set to 3 for precipitation. Is it necessary? I suggest the authors unify the number of decimal places.

29. 21-12. 'figure' should be 'figures'.

30. 26-6. 'et al.'. The authors should list all the authors of a citation and unify the citation style. The authors should proof read the manuscript to avoid such mistakes.

31. Texts of most of the figures are unclear. I would suggest the authors redraw the figures.

---

## Referee Comment (RC2) · Anonymous Referee #2 · 27 Sep 2016

GENERAL AND IMPORTANT COMMENTS ABOUT THE MANUSCRIPT

The Manuscript (MS) is an attempt to integrate various sources of satellite remote sensing data towards macro-scale hydrologic modeling in a region in Africa. Such a concept is novel considering the eminent data limitations pertaining to lack or limited observed in-situ hydro-meteorological data important for model calibration and validation purposes. In this study, the authors seem to be interested in historical cases of the water budget, and hence may elect to put this is the title, or justify why they are not interest in forecasting. From the present standpoint, however, the paper can be considered for publication in the near future, but only after addressing some serious technical issues that degrade the novel concept proposed and applied by the authors. In this

respect, and to improve and make the MS much better, I wish to recommend major revisions before further consideration. The following are some of the major comments that need redress:

(a). Language Limitation: the MS is poorly written and generally very difficult to read right from the abstract to the conclusions. This may be due to language limitation/culture of the authors, but considering that the MS will have a bigger readership; it would be nice to English edit the MS so that the actual intentions - technical and linguistic- can come out clear. The way the results, especially the statistics and maps, are presented makes one question the objective of the work. In some cases, it is difficult to understand it the authors intend a comparative assessment at various spatial scales of the regions in the basin? There is also the random use of difficult expressions appearing from nowhere without prior definition, i.e. in defining the table in page 15, he used Figure 5, Table 2 which is difficult to understand.

(b). the author claim that his research is motivated by data limitation. However, he seems to have some stations with streamflow data as by the hydromet stations in the study area map or otherwise, the hydrographs used in the validation exercise. This begs the question: So where is the boundary of this data limitation he is claiming? Could it be possible to use the available data to parameterize the model and later regionalize the model? Or is it possible to develop criteria to extrapolate the results after calibration and validation of the satellite estimates with the limited but available observed data-sets? The authors may also need to justify why 402 sub watershed were delineated considering the limited river gauging stations shown in the study area map. If he wants to retains them, then he should define use a criteria to choose at least 10-15 sub-catchments and provide their morphometry together with the simulated values of the water balance components in the results section, for consistency and clarity. A table (and not maps) in this respect would quickly help things out here.

(c). considering data uncertainties, would it be wise to believe the higher model reliability and hence results? The authors need a good and elaborate justification of how the
errors cancelled out during the simulation. Furthermore, the author seems to be using some part of the available data for calibration, and the same half plus the rest within the time frame for validation. In my opinion, the conventional way would be to divide the data-sets into two, one for calibration and the other for calibration. Could this be the reason for the good efficiency realized? The authors need to justify this methodology very strongly.

SPECIFIC COMMENTS ABOUT THE MANUSCRIPT

(1) TITLE

The title is okay and acceptable, but may sound better if the authors consider the conventional way of staring a sentence with a verb i.e. Modeling/Estimation/Assessing of the Water Balance etc. This is however trivial at this moment.

(2) ABSTRACT In my opinion, the first sentence can be made simple and realistic i.e. . . .by saying the region is one of the data scarce regions is the developing regions (but not in the world as this raise a lot of questions and may temp one to ask for proof of review in the introduction.

Are there basins in the UNRB that have data? Is the justification of one of the data scarce regions in the world thus still valid?

In my opinion, the water budget components of study can be explicitly mentioned in the sentence without the brackets, and the tools used well captured and summarized. This makes the section clear and easy to read.

Considering that modeling procedure employed, and the possible uncertainties involved, the results need to be rounded off i.e. by saying that precipitation values between 1000-1600mm were estimated depending on seasonality etc

Generally, the abstract can be well written and summarized in good English language, and only important content.

[Figure]

(3) INTRODUCTION:

This section can be language edited and the phrases backed with the latest references. The references also need to be ordered either from the latest to the oldest or vice versa as required by the journal.

In my opinion, the text in lines # 4-10 can be summarized and well captured within the text without using bullets or points.

Lines # 27-28: the sentence beginning with [The use of RS precipitation products. . .] can be well written, more content added and justified. Here the authors can introduce and justify the use of other products such as GLEAM, MODIS data products etc for simulation. The author seems to neglect this section/paragraph and YET it forms the basis of their novel idea of using RS for data scarce regions. In my opinion, 'at least two paragraphs' on this section should be added to improve and justify his methodology where he has introduced a lot of RS products from nowhere. For instance, how have these RS tools and methods been applied in other regions of data scarcity? What were the results achieved? Can the methods be replicated in the current study basin? Has the JGrass Newage (JGNA) model been applied elsewhere and what were results and strengths etc? This section should a major part of the MS and if not well captured then it can be concluded that the MS contributes very little value to hydrological science.

(4) THE STUDY AREA:

There are loose statements here and there that can be tightened and generalized. For instance, in line 5, one would ask: where is Bahir Dar where the river originates? Such loose statements assume and make the MS only fit for regional publication.

In my opinion, one elaborate map of topography (DEM), river network and stream gauges can be sufficient here. I am also sure with good GIS skill, and added topo-logical data, the rainfall stations can still be added without making the map look untidy and congested. Or else, he may also elect to take a map of the catchment delineations

and the rainfall stations in the methodology, and use that chance to highlight the sub-catchments (better more than 10) where he wants to focus his results using a table as mentioned above already.

(5) METHODOLOGY:

On page 4 lines #12-15, the authors may want to choose one or two more applicable references of the co-author.

In page 5, Figure 2 needs simplifications and better explanations. The color coding shades used will not appear if the paper is printed in black and white.

Some parts in section 3.2.1 ideally belong to the introduction. Let the authors focus on the data-sets used and why they were used.

The reference Abara et al., submitted is completely out placed and may not be necessary at this stage of the journal.

There are many good ways of structuring this section in hydrology. Let the authors develop a simple and flowing structure from section 3.1. For example, section 3.1 can be titled 'Data and Methods'. Section 3.1.1 can be on 'Water Balance Modeling'. Section 3.1.2 can be on 'The Modeling System'. Section 3.1.3 can be on 'Data and Modeling Procedure' etc. The authors are free to choose what structure they want to adopt. As it is at the moment, there is too much information everywhere, a majority of which is not well captured and explained. Some content in section 3.2.3 on page 7 are not necessary and can be avoided generally.

Section 4 on calibration and validation can be renamed as section 3.2 and well elaborated as explained before. In this section, the authors need to JUSTIFY WHY the same data period used for calibration is also available for Validation? This may infer a technical limitation that can affect the model results purported by the authors.

(6a) RESULTS AND DISCUSSION:

Generally, the results are not balanced and well presented. The spatial maps dominate all the results. Well structured tables may provide more information considering the many catchments of study.

The first paragraph in the results section may not be necessary, or better be summarized.

The authors should find a way of presenting the maps in a nice, simple and clear manner. As they are at the moment, the polygons dominate the results. An elaborated table with selected catchment justified in the methodology can be good enough. Only one or two maps can be used here for visualization and overall balance of presentation of the results.

In line 23-24 of page 9, is the discrepancy small as mentioned? Could it be that the SM2R-CCI was not properly corrected? Please explain into details.

The legend for Fig 3 needs to be well placed and elaborated.

In section 5.1.1 of page 11, there is need for technical justification by the authors as this a very strong section of hydrology. (i) If GLEAM has had validation in other areas, with a good match with observations, then I it would be ok to use it for plausibility checks. However, as it stands, the New Age simulation of ET highly over- or under-simulate the ET fluxes. Should the results thus be fully trusted with these graphs?

The author can elect to present one or two of the Graphs/Figures but well elaborated and discussed into details. As it is, figure 4(b) is of limited value and would rather be discussed in the text or annexed.

Considering the model/data uncertainties, a KGE of 93% may be theoretically high if not good enough. There is hence a need for a strong justification of how the errors cancelled out during calibration and validation.

Fig 5 is not well represented. This can be avoided or the authors can choose the sub-catchments to illustrate 'a prior in the methodology section' as mentioned already. The

challenge here is that with the many sub catchments, the author does not seem to know how to cluster them in a consistent manner throughout the paper.

The results on page 14 can be summarized and well written. On table 2, is the final outlet of Upper Blue Nile located at El Diem with an area of 174 000km2? No idea!

Fig 6 on page 15 needs to be elaborated and well explained or else moved to the annex.

On page 16, it would be good to justify how the discharge in the entire basin was modeled. I.e. did you add/route all the upstream discharges and accumulated downwards? This as a technical consideration for the paper.

All the results needs to be discussed from a hydrological standpoint. This section is important for the authors to justify the publication, and provide key element of study that improves the knowledge in hydrology in such areas generally.

(7) CONCLUSIONS:

The paper needs to be summarized in the context of the study. Considering the uncertainties, the results need to be reported with this recognition i.e. ET values between 650-750mm were estimated for various sections of the basin etc

There is need for more conclusions about the challenges of the study and the methods generally. This will form a basis for recommending future studies in areas with similar data limitation.

As it is, the section is completely lacking and does not provide future research directions in hydrology.

(8) REFERENCE:

The references are not formatted to the Journal requirements as required by HESS. Check and realign all of them.

---

## Author Comment (AC1) · 27 Sep 2016

We would like to thank reviewer 1 for the constructive comments. We welcome all the issues raised. Below, you will find a point by point description of how each comment was addressed. The reviewer comments in bold font, and our response in normal font.

**This manuscript proposes a method to improve water budget modelling by using the available, but sparse, hydrometerological data and satellite products. The current manuscript provides a good try to predict hydrological process in data-scarce regions or ungauged basins. Although there are publications related to such topic in ungauged basins, the intent of the manuscript is worthy and significant, and is of interest to readers of HESS. Seeing the potential of this study, I**

**am in general supportive of publication if the following comments are addressed in the resubmission.**

Dear reviewer 1, we thank you for the general appreciation of our work, the comments and suggestions you give that helps to further improve our manuscript. In the followings, your comments are answered one by one:

**Major concerns:**

**1. I would encourage the authors to rewrite the methodology section. Give a clear message to the reader what you did and how you did. For example, the manuscript entitled as 'JGrass-NewAge model system'. However, I could not find detail or key information about the method. What's the theory of the method based on? What's the advantage of the method? The headings in method section are the same as those in section 5.**

Regards to the JGrass-NewAge system, it is a model system built on the object modeling system v3 (OMS3) informatics, which aims to deploy modern modeling solutions, with the philosophy of promoting reproducible research system. The best way to have general information about it is the paper Formetta et al., 2014. JGrass-NewAGE is a collection of various modeling solutions for all hydrological compartments or fluxes. The detail of each component are presented and validated in various papers: rainfall-runoff modeling (Formetta et al. 2011), shortwave solar radiation modeling (Formetta et al. 2013), longwave solar radiation modeling (Formetta et al. 2016), and digital watershed modeling (Formetta et al. 2014b; Abera et al. 2014). We believe the level of details about JGrass-NewAge in page 4 and 5 are enough, but we will revise the section for clarity.

Regarding to the titles of the subsections in methodology and in results section, the components of the water budget (precipitation, evaporation, discharge, and storage, sequentially) is given in both sections. It seems clear for us that the methodology details how we estimate each flux/storage and the results section presents results of

the work.

**1.a. Some parts in the results analysis and discussion section are more suitable to be in the methodology section. For instance, it would be better to introduce the indices (i.e., KGE, PBIAS, r) in section 4. In addition, what's the spatial resolution of the HRU? When performing simulation, what are the time step and the spatial resolution of output?**

It is true that goodness-of-fitness (GOF) indices can be in section 4. However, since those indices are common in literature, maintaining their detailed in the main text is distractive. That is the reason we decided to move description of the indices in the appendix section. However, we add a phrase that refers to the appendix also in the methodology section.

The mean (standard deviation) spatial resolution of the HRU is about 430 ($\pm$ 339) km2 and we use daily time steps. The simulation results are therefore one for each HRU and at each time step. The HRU estimates should be considered as an average. Discharges however, are simulated at the nodes of the river networks. We will describe better both the spatial and temporal resolution of the simulation in their respective sections.

**1.b. There are different hydrometerological data and satellite products, but it is difficult to readers to obtain their information (e.g., what kind of satellite products). I would suggest the authors providing a table to show all the data and their spatiotemporal resolutions. How did you deal with the different resolutions (especially spatial resolution) of input parameters?**

The approach we followed on the description of the satellite products is to use a single 'best' satellite product, based on our review, already discussed with detail in another paper for rainfall, e.g. Abera et al, 2016. Then, the product is described in the methodological section along with the description of the methods used to estimate the component. For instance, SM2R-CCI for precipitation, GLEAM for ET (but we will provide

appropriate comparison with MODIS in complimentary material), in-situ hydrometer data for discharge (no other choice possible), and GRACE for storage change (no other choice possible). The methods for processing and estimating the data at each HRU level are described in section 3 and 4. However, we will revise the section for adding clarity. In addition, a separate table describing all the satellite products used in the paper and its spatial and temporal resolutions will be added at the end of the methodology section in the revised paper.

The reference spatial resolution for model inputs and validation is the area of each HRU. So, for each HRU, we estimate the weighted average of the quantity weighted by how much of the pixel area overlaps with the HRU polygon.

**2. Discussion should be enhanced. What's the disadvantage of the method when applying in data-scarce regions with large area? For example, results of figure 5 indicated that the simulated runoffs were underestimated. What's the reason? Was it caused by uncertainties/errors in precipitation products? I could not find any quantitative information about errors of SM2R-CCI. Meteorological stations should observe precipitation, radiation, and etc. Why didn't you use them for validation and discussion?**

It is true that the model underestimation is most likely due to the underestimation of SM2R-CCI, as described on the page 11 line 29. Abera et al., 2016 (cited in the manuscript), by comparing with in-situ observations, shows that SM2R-CCI slightly underestimates the total cumulative rainfall in the study area. However, this resulted the best among the products we analysed. Obviously the error estimation can also be caused by models' inconsistencies, and the necessity to work by averaging out inputs over large areas. The fact to be remarked is, however, that our simulations improve previous results.

**3. The authors claimed that the JGrass-NewAGE system are described in a series of papers and not re-discussed in this manuscript. What's the difference**

**between this study and the previous papers? What's the main contribution of this work?**

The previous papers contain description of the single components that were validating separately on other catchments of different size. They cover the informatics, DEM treatment and river network schematisation, radiation, runoff, snow modeling. In this paper those components are united in a modelling solution and work all together cooperatively to solve the water budget closure. In addition, the application in a large basin using various data (satellite and in-situ), which NewAge was originally developed for, is an important contribution of this paper.

**Specific comments:**

**1. 1-21. 'up to 2000 mm per year'. It would be much clearer by adding precipitation.**

In the revised manuscript, we will add the exact precipitation value.

**2. 3-1. It should have space between 'given' and '('. The authors should proof read the manuscript to avoid such mistakes.**

Space will be added; we will remove such errors in the revised manuscript.

**3. 3-6. 'the river enters a deep a canyon' contains grammatical errors.**

Thank you for this, we will correct it

**4. 3-18. The elevation values show certain difference compared to those in page 2 line 3.**

Thanks you for spotting this. The one in page 2 line 3 was takes from literature value, and the page 3 line 18 was taken from SRTM digital elevation data. In any ways, we will revise and make it consistent.

**5. 3-30. It may mislead to conclude 'the seasonal variability of the basin is very**

high' because the authors claimed that the temperature has small seasonal variability.

We will improve this by explicitly mentioning the precipitation and evapotranspiration in the sentence.

**6. 4-1. Figure 1. I suggest adding units for axes (also other figures) as well as enlarging the schematic map (at least the text). What does the colour represent in figure 1b?**

We will enlarge the figure (text). The color in figure 1b is the mean elevation of HRU in the basin.

**7. 4-15. It seems that the citation appeared in the first time, and 2014b should change to 2014a. The authors should proof read the manuscript to avoid such mistakes.**

We will correct the citation.

**8. 5-4. What does GIS mean? Please consider defining the abbreviation.**

Thank you for this. GIS refers to geographic information system. We will add the list of abbreviations in the revised paper.

**9. 5-9. How did you divide the basin into 402 subbasins? According to what kind of rules? I'm not sure whether figure 1b is your results or not.**

The partition of the basin into 402 subbasins is based on the standard watershed partition approach, and the specific procedure for JGrass-NewAge is described in detail in Formetta et al., 2014 and Abera et al 2014. Yes, figure 1b is the subbasin partition results as mentioned in the caption.

**10. 5-13. Figure 2 is difficult to read. The texts were small and difficult to guess their meaning. I suggest the authors redraw it.**

We will improve the readability of the figure.

**11. 6-23. Works cited in a manuscript should be accepted for publication or published already. There are many publications describing psychometric constant.**

Appropriate citation will replace the submitted manuscript.

**12. 6-27. What's the relation between S(t) and TB in equation 3? Can you explain more?**

There is no relation between S(t) and TB, at least for what related to equation (3). S(t) is the water (storage) present in a HRU. Instead, TB, the Budyko time, affects the alpha in equation (3), because the value of alpha is obtained for balancing the water budget (i.e equation (1)) in such a way that after TB years the storage equals the initial one, i.e. S(TB) = S(0). This implies the use of an optimisation procedure, and such alpha is obtained together with the other parameters of the overall modelling solution (including runoff production, evapotranspiration, etc.) within the calibration procedure. We will try to explain it better in the revised text.

**13. 7-26. Semicolon should be replaced with 'and'.**

We will replace the semicolon with 'and'.

**14. 8-4. What does KGE mean? Please consider defining the abbreviation.**

Thank you for spotting this. We will add the definition in the first instance. In addition we will add the list of abbreviations in the revised paper.

**15. 8-8. What does 'described in A' mean? Does 'A' represent 'Appendix'?**

Thank you, we will add Appendix before 'A'.

**16. 9-18. It is curious to use J representing precipitation. In addition, precipitation, evapotranspiration, and discharge are components of water budget. Why did you use different section headings (i.e., 5.1, 5.1.1, 5.1.2, . . .)?**

[Figure]

We usually adopted J for precipitation, as for instance, in Rigon et al. 2016, but we can adopt any other symbol. Yes, there is error in the heading sections. We will use the same level of heading for all the components.

**17. 9-21. I would suggest the authors adding 'the Oromia region (or other mentioned places)' into Fig.1.**

Thank you for this. However, we argue that the important idea here is to show the spatial pattern within the natural basin. We already verified that adding regional boundaries (information) makes figure 1 very crowded. It seems better to delete the region name from the text, as it is the only one mentioned.

**18. 10-1. Figure 3a indicates precipitation is highest in southern region. However, figure 3b showed a different pattern (i.e., east shared highest precipitation), especially in JJA.**

The two figures are different. Figure 3a shows the long-term mean precipitation as perceived by reviewer 1. Figure 3b, however, shows the level of percentage share of precipitation falls by seasons. In the east part of the basin, the highest percentage share (of its lower annual precipitation) falls in summer (JJA) in comparison to the other parts.

**19. 11-4. How and why did you select only some subbasins? Did you consider r and PBIAS (figure 4, e.g., high r and low PBIAS, and low r but high PBIAS)?**

We didn't consider r or PBIAS to select the subbasins. We select the three sub basins systematically to cover the basin spatial distribution: one from eastern, center, and western part of the basin.

**20. 11-10. 'while the it tends to' contains grammatical errors.**

We will remove 'the' from this sentence.

**21. 11-23. 'within the basin at the internal channels (2)'. What does '(2)' mean?**

It is changed to "(Table 2)" in the new manuscript.

**22. 11-27. I do not think r2=0.92 is lower than r=0.93 or r=0.94. I suggest the authors to unify the index.**

It is very difficult to find similar index across all the papers. But, having PBIAS and r are relatively common, we decided to use r and PBIAS for comparison, in addition to KGE which is our primary index of model evaluation. We are also prudent to do comparison with other studies. So in this section, we just indicate the comparative performances "......(KGE=0.92, PBIAS = 2.4, r = 0.93). The results show that the performances of the NewAge simulation are similar to the performances reported by Mengistu and Sorteberg (2012), with slightly lower PBIAS value (PB=8.2, r2 =0.92)".

**23. 13-1. Are all the parameters unitless? Why are two [−]? Furthermore, I could not find table 1 in the context.**

The three parameters (with [-]) are unit less and for others it is length and time, which is given by [L] and [T] respectively in the table. Thanks for indicating the confusion between the two $\alpha$[-]. In the revised manuscript the first and second $\alpha$[-] will be changed into $\alpha_{hymod}$[-] and $\alpha_{ET}$[-] respectively.

**24. 13-2. Can you number the hydrometer stations and then add these IDs into figures 1b and 5?**

Yes, we will do that

**25. 14-8. Are Wase-Tana and FlexB commonly used models? Please consider defining the abbreviation.**

It is true the two models are not common. We will define them. In addition we will add these in the list of abbreviations.

**26. 18-5. Can you provide some radiation, cloud, and wind observations? This may be better to draw the conclusion.**

We don't have observations of radiation, cloud and wind. We used JGrass-NewAge shortwave component to estimate the radiation data, together with the information of cloud fractional cover (CFC) from EUMETSAT Climate Monitoring Satellite Application Facility (CM SAF) project (Schulz et al., 2009). Wind data is not used at all in this study. It is true that including the radiation estimates and cloud data provides more insight to understand the conclusion given at this particular line. Providing spatial maps of these data in the manuscript, however, reduce its fleetness. But, we will add some of these data in complimentary materials. Here are some samples of the cloud cover map for the basin: http://ecohydrogeomorpho-metry.blogspot.it/2016/04/cloud-cover-on-surface-net-radiation.html

**27. 19-9. What does S mean?**

We will change this into ds/dt.

**28. 19-11. The number of decimal places was set to 3 for precipitation. Is it necessary? I suggest the authors unify the number of decimal places.**

Of course it is not important. We will unify the decimal number throughout the paper.

**29. 21-12. 'figure' should be 'figures'.**

We will change to 'figures'.

**30. 26-6. 'et al.'. The authors should list all the authors of a citation and unify the citation style. The authors should proof read the manuscript to avoid such mistakes.**

We will correct all citation errors.

**31. Texts of most of the figures are unclear. I would suggest the authors redraw the figures.**

In the new manuscript, we will improve the figure clarity.

**References:**

Abera, W., Antonello, A., Franceschi, S., Formetta, G., and Rigon, R.: The uDig Spatial Toolbox for hydro-geomorphic analysis, British Society for Geomorphology, London, UK, in: clarke nield (eds.) geomorphological techniques (online edition) edn., 2014.

Abera, W., Brocca, L., and Rigon, R.: Comparative evaluation of different satellite rainfall estimation products and bias correction in the Upper Blue Nile (UBN) basin, Atmospheric Research, 178-179, 471-483, doi:1 2016.

Formetta, G., Mantilla, R., Franceschi, S., Antonello, A., and Rigon, R.: The JGrass-NewAge system for forecasting and managing the hydrological budgets at the basin scale: models of flow generation and propagation/routing, Geoscientific Model Development, 4, 943–10 955, 2011.

Formetta, G., Rigon, R., Chávez, J., and David, O.: Modeling shortwave solar radiation using the JGrass-NewAge system, Geoscientific Model Development, 6, 915–928, 2013.

Formetta, G., Antonello, A., Franceschi, S., David, O., and R., R.: The basin delineation and the built of a digital watershed model within the JGrass-NewAGE system, Bolet?n Geol?gico y Minero: Special Issue "Advanced GIS terrain analysis for geophysical applications, 2014a.

Formetta, G., Antonello, A., Franceschi, S., David, O., and Rigon, R.: Hydrological modelling with components: A GIS-based open-source framework, Environmental Modelling Software, 55, 190–200, 2014b.

Formetta, G., Kampf, S. K., David, O., and Rigon, R.: Snow water equivalent modeling components in NewAge-JGrass, Geoscientific Model Development, 7, 725–736, 2014c.

Mengistu, D. and Sorteberg, A.: Sensitivity of SWAT simulated streamflow to climatic changes within the Eastern Nile River basin, Hydrology and Earth System Sciences,

16, 391–407, 2012.

Rigon, R., Bancheri, M., Formetta, G., de Lavenne, A. (2016). The geomorphological unit hydrograph from a historical‐critical perspective. Earth Surface Processes and Landforms, 41(1), 27-37.
* * *

---

## Author Comment (AC2) · 3 Oct 2016

We would like to thank reviewer 2 for the constructive comments. Below, you will find a point by point description of how each comment was addressed. The reviewer comments in bold font, and our response in normal font.

**GENERAL AND IMPORTANT COMMENTS ABOUT THE MANUSCRIPT**

**The Manuscript (MS) is an attempt to integrate various sources of satellite remote sensing data towards macro-scale hydrologic modelling in a region in Africa. Such a concept is novel considering the eminent data limitations pertaining to lack or limited observed in-situ hydro-meteorological data important for**

[Figure]

**model calibration and validation purposes. In this study, the authors seem to be interested in historical cases of the water budget, and hence may elect to put this is the title, or justify why they are not interest in forecasting. From the present standpoint, however, the paper can be considered for publication in the near future, but only after addressing some serious technical issues that degrade the novel concept proposed and applied by the authors. In this respect, and to improve and make the MS much better, I wish to recommend major revisions before further consideration. The following are some of the major comments that need redress:**

We thank reviewer 2 for the appreciation of our work. When performing our studies we analyzed historical data, as any other hydrological study. We are, obviously, interested in forecasting the hydrological cycle components, but this necessarily relies on the availability of the meteorological forcings. It is possible to forecast (in the sense of meteorology) discharges (for instance) if we have had rainfall (and other meteorological) data. This assumes that we have access to real time data in the basin, which we do not have. More relaxed forecast, or better, projection, could be made after acquiring appropriate climate projections. But for this, to have a model system, which is validated for a given basin, is the first step. This is actually one of the goals of the present paper.

We will use as much as possible the suggestions given by the reviewer to improve our new manuscript.

**Major concerns**

**(a). Language Limitation: the MS is poorly written and generally very difficult to read right from the abstract to the conclusions. This may be due to language limitation/culture of the authors, but considering that the MS will have a bigger readership; it would be nice to English edit the MS so that the actual intentions - technical and linguistic- can come out clear. The way the results, especially the statistics and maps, are presented makes one question the objective of the work.**

**In some cases, it is difficult to understand it the authors intend a comparative assessment at various spatial scales of the regions in the basin? There is also the random use of difficult expressions appearing from nowhere without prior definition, i.e. in defining the table in page 15, he used Figure 5, Table 2 which is difficult to understand.**

We will improve as much as possible the layout (see detailed comments) and the writing of our manuscript. In page 15 there is not Tables. There are Tables in page 13, and we assume the reviewer refers to them. We will try to improve the quality of their caption to present the results in the most clear way we can.

**(b). the author claim that his research is motivated by data limitation. However, he seems to have some stations with streamflow data as by the hydromet stations in the study area map or otherwise, the hydrographs used in the validation exercise. This begs the question: So where is the boundary of this data limitation he is claiming?**

Data limitation does not mean total absence of data. Certainly we have some precipitations and discharge data. However these data are in 35 locations for precipitation data in an area of 175 thousand square kilometers. Meaning, just a station every 5000 square kilometers or squares of around seventy by seventy square kilometers of side (on average). Convective processes generating precipitation can be as small as 10 kilometers square, so the optimal gauge network distribution should be as small as that, to capture all the relevant phenomena. Considering this fact, almost any region in the world is data-scarce, but some regions such as the Upper Blue Nile basin are even more hydrometeorological data-scarce regions than others. For discharge analysis, the numbers of hydrometer stations are very few (16 hydrometers) with a data set having lots of missing data and gaps. So for the objective outlined in the study, estimations of spatially and temporally hydrological information of the basin, UBN surely can be characterized as data limited basin.

**Could it be possible to use the available data to parameterize the model and later regionalize the model? Or is it possible to develop criteria to extrapolate the results after calibration and validation of the satellite estimates with the limited but available observed data-sets?**

Yes, this is actually what it was done. We use all the data available to calibrate the model and we "interpolate" (propagate) all the data (hydrological information) by means of the model in the inner points. Actually, if with regionalisation the reviewer means statistical techniques, we did not use any of them. If the reviewer asks for the transferability of our approach, we can confirm that it can be extrapolated to any basin with similar or larger size.

**The authors may also need to justify why 402 sub watershed were delineated considering the limited river gauging stations shown in the study area map.**

Even if hydrometeo data are available in fewer stations, satellites allow us to have rainfall forcing at a much finer scale. Partition of the basins in 402 parts is functional to use all the rainfall spatial information we have, in a trade-off with a reasonable computational demand. It also serves to accounts for the morphological structure of the river network, which, obviously counts very much in forming the hydrologic response. On the latter topic, the last author co-authored some papers that can support this fact. We will add a clarification on this in the revised manuscript.

**If he wants to retains them, then he should define use a criteria to choose at least 10- 15 sub-catchments and provide their morphometry together with the simulated values of the water balance components in the results section, for consistency and clarity. A table (and not maps) in this respect would quickly help things out here.**

If we did not clearly communicate the objective of the paper, obviously, it is our fault. However, the objective of the paper is to estimate spatio-temporally distributed water budget of the UBN basin. Hence, the methodology followed and the results presented

all are for the whole basin, not for specific sub-catchments. When in-situ data is available, that specific sub-catchment is used to verify the performance of the model estimations. If the reviewer wants to select some catchments, we can provide part of this information in the complementary material of the revised manuscript.

**c. Considering data uncertainties, would it be wise to believe the higher model reliability and hence results?**

We considered ground measure as true. Untrusting them would lead us to absolute ignorance. However, the data provided by the model solution we used show that there is consistency between discharge gauges and rainfall estimates, give parameters that work decently also for the validation periods. Model and data are consistent (once the model is calibrated). That is all we can say. But what can we say else?

**The authors need a good and elaborate justification of how the errors cancelled out during the simulation.**

Errors do not cancel. When possible, any of the modelling components used was validated separately. We have checked the functioning of each of them in many other cases, as testify by our own literature, even if in those cases data were less scarce. In this specific case, precipitation from satellites is verified and corrected using the available few in-situ observations, storage (at least at the whole basin scale) is verified using GRACE data, discharge is verified at about 16 hydrometer stations. So we know that each component, besides implementing sound science, works fine with the appropriate data. That is what we can trust. When we calibrate hydrological model just on discharge data, parameters' values become a collector of uncertainties (a garbage collector, as some colleague calls it), but we assume that this is well understood and does not require a further disclaimer.

**Furthermore, the author seems to be using some part of the available data for calibration, and the same half plus the rest within the time frame for validation.**
We don't. We will clarify it. We used some part of the available data to calibrate the model at the main outlet, and used the other part for validation. In addition, the other data sets available in the interior hydrometer stations are used for validation the model capability to estimate discharge at each links of the river network of the basin.

**In my opinion, the conventional way would be to divide the data-sets into two, one for calibration and the other for calibration.**

Correct!

**Could this be the reason for the good efficiency realised? The authors need to justify this methodology very strongly.**

As we said, we did not use the same data for both validation and calibration. Hence, we believe that the reason for good model performance is due to the explicit character-isation of inputs strategies and the goodness of the modeling solutions adopted.

**(1) TITLE**

**1 - The title is okay and acceptable, but may sound better if the authors consider the conventional way of staring a sentence with a verb i.e. Model-ing/Estimation/Assessing of the Water Balance etc. This is however trivial at this moment.**

That's OK for us. We changed the title to: "Modelling the water budget of the Upper Blue Nile basin using the JGrass-NewAge model system and satellite data"

**(2) ABSTRACT**

**2 - In my opinion, the first sentence can be made simple and realistic i.e. . . .by saying the region is one of the data scarce regions is the developing regions (but not in the world as this raise a lot of questions and may temp one to ask for proof of review in the introduction. Are there basins in the UNRB that have data? Is the justification of one of the data scarce regions in the world thus still valid? In my**

**opinion, the water budget components of study can be explicitly mentioned in the sentence without the brackets, and the tools used well captured and summarized. This makes the section clear and easy to read. Considering that modeling procedure employed, and the possible uncertainties involved, the results need to be rounded off i.e. by saying that precipitation values between 1000-1600mm were estimated depending on seasonality etc Generally, the abstract can be well written and summarized in good English language, and only important content.**

We accept the corrections, and we revised the abstract following the reviewers' guidelines.

**(3) INTRODUCTION**

**3 -This section can be language edited and the phrases backed with the latest references. The references also need to be ordered either from the latest to the oldest or vice versa as required by the journal.**

We will do it in the revised manuscript.

**4 - In my opinion, the text in lines 4-10 can be summarised and well captured within the text without using bullets or points.**

In the revised manuscript, we will try to synchronize them in shorter sentences.

**5 - Lines 27-28: the sentence beginning with [The use of RS precipitation products. . .] can be well written, more content added and justified. Here the authors can introduce and justify the use of other products such as GLEAM, MODIS data products etc for simulation. The author seems to neglect this section/paragraph and YET it forms the basis of their novel idea of using RS for data scarce regions. In my opinion, 'at least two paragraphs' on this section should be added to improve and justify his methodology where he has introduced a lot of RS products from nowhere. For instance, how have these RS tools and methods been applied in other regions of data scarcity? What were the results achieved? Can**

the methods be replicated in the current study basin? Has the JGrass NewAge (JGNA) model been applied elsewhere and what were results and strengths etc? This section should a major part of the MS and if not well captured then it can be concluded that the MS contributes very little value to hydrological science.

The intention of these two sentences was to avoid the description of various remote sensing (RS) products, and instead suggest that the readers should look for this information in milestone papers in the use of RS for hydrology. We will add further information of this in the revised manuscript. In the same mood, we do not want to add much information about JGrass-NewAGE that can be better accessed in previous papers by the same authors. . We will try to improve this section.

**(4) THE STUDY AREA**

**6 - There are loose statements here and there that can be tightened and generalized. For instance, in line 5, one would ask: where is Bahir Dar where the river originates? Such loose statements assume and make the MS only fit for regional publication. In my opinion, one elaborate map of topography (DEM), river network and stream gauges can be sufficient here. I am also sure with good GIS skill, and added topological data, the rainfall stations can still be added without making the map look untidy and congested. Or else, he may also elect to take a map of the catchment delineations and the rainfall stations in the methodology, and use that chance to highlight the subcatchments** . . .

Thank you, we will improve our mapping and make a larger figure. As suggested by the reviewer, we will dedicate one map describing the DEM, river network, and stream gauges, with some places such as Bahir dar marked on it. Since the sub basins are the scale at which the water budget is estimated, we will also be maintain this map along the former .

**7 - (better more than 10) where he wants to focus his results using a table as mentioned above already.**

We do not think that adding more catchments' details will be useful for the readability of the paper. However, DEM, important shape files to be used in GIS, and the list of catchments details will be provided as complementary material.

**(5) METHODOLOGY**

**8 - On page 4 lines 12-15, the authors may want to choose one or two more applicable references of the co-author.**

The lists of papers cited are describing different modeling solutions, each for one component of the JGrass-NewAge system. Since all components are used, it is important that we cited all of them. However, we will revise the sentences for making it easier to read.

**9 - In page 5, Figure 2 needs simplifications and better explanations. The color coding shades used will not appear if the paper is printed in black and white.**

Thank you, we will improve the text and change the color shades.

**10 - Some parts in section 3.2.1 ideally belong to the introduction. Let the authors focus on the data-sets used and why they were used.**

Actually what has been written in the first and second paragraph was the explanation why and how we used SM2R-CCI precipitation data. In any case, we will revise it.

**11 - The reference Abera et al., submitted is completely out placed and may not be necessary at this stage of the journal.**

We think it is not a bad thing, and let the citation.

**12 - There are many good ways of structuring this section in hydrology. Let the authors develop a simple and flowing structure from section 3.1. For example, section 3.1 can be titled 'Data and Methods'. Section 3.1.1 can be on 'Water Balance Modeling'. Section 3.1.2 can be on 'The Modeling System'. Section 3.1.3 can be on 'Data and Modeling Procedure' etc. The authors are free to choose**

**what structure they want to adopt. As it is at the moment, there is too much information everywhere, a majority of which is not well captured and explained.**

We realized that sub-sectioning of section 3 and 4 went wrong. New subsections will be:

3 Methodology
3.1 JGrass-NewAGE System setup
3.2 Precipitation
3.3 Evapotranspiration
3.4 Discharge
3.5 Water storage
3.6 Calibration

4. Results and discussion
4.1 Precipitation
4.2 Evapotranspiration
4.3 Discharge
4.4 Water storage
4.5 Water budget closure

5. Conclusions

We think that in this way there will be a clear relation between the topics of the two sections (section 3 and 4). So we argue that the approach followed in the paper is better than the one suggested by the reviewer. Obviously we will try to improve the description and the discussion. However, the reviewer should understand that dealing with the whole hydrological cycle is a complex task that requires attention. A detailed understanding of all its parts cannot be obtained without reading the other papers on JGrass-NewAGE where we cover a huge amount of work and testing.

**13 - Some content in section 3.2.3 on page 7 are not necessary and can be avoided generally.**

Section 3.2.3 contains totally twelve lines. It is very difficult for us to understand what we can avoid to say. We give information about the algorithm we use for reproducing discharges, and the validation method. We believe that this information is necessary.

**14 - Section 4 on calibration and validation can be renamed as section 3.2 and well elaborated as explained before. In this section, the authors need to JUSTIFY WHY the same data period used for calibration is also available for Validation? This may infer a technical limitation that can affect the model results purported by the authors.**

Regarding about section renaming, please see specific comment 12. We did not use the same data for calibration and validation, as described in major comment C.

**6. RESULTS AND DISCUSSION**

**15 - Generally, the results are not balanced and well presented. The spatial maps dominate all the results. Well structured tables may provide more information considering the many catchments of study.**

We think that one figures convey more than thousands words if well understood. Evidently we were not able to convey clearly their meaning. We will work to improve figure captions and comments. Most of the data are (and will) be provided as complimentary material with some table of summary for what it is feasible to do. Finally all of our procedure are based on open software and can be repeated step by step by any researchers.

**16 - The first paragraph in the results section may not be necessary, or better be summarised.**

Thank you, we will summarise it.

**17 - The authors should find a way of presenting the maps in a nice, simple and clear manner. As they are at the moment, the polygons dominate the results. An elaborated table with selected catchment justified in the methodology can be good enough. Only one or two maps can be used here for visualisation and overall balance of presentation of the results.**

Given our objective, the presentation of our results without maps is impossible. We limited one, if not two, figure (plot) for each component. Data are averaged over a subbasin and there is not internal spatial variability in the output. So it is clear that "polygons" stand out.

**18 - In line 23-24 of page 9, is the discrepancy small as mentioned? Could it be that the SM2R-CCI was not properly corrected? Please explain into details.**

The difference between annual long-term rainfall value of 1900 mm and 2049 mm, given by different data sources, can be considered small. Besides, if one considers the uncertainty pertinent to each data sources and estimation method, s/he should conclude that the difference is acceptable.

**19 - The legend for Fig 3 needs to be well placed and elaborated.**

We will do it.

**20- In section 5.1.1 of page 11, there is need for technical justification by the authors as this is a very strong section of hydrology. (i) If GLEAM has had validation in other areas, with a good match with observations, then I it would be ok to use it for plausibility checks. However, as it stands, the New Age simulation of ET highly over- or under-simulate the ET fluxes. Should the results thus be fully trusted with these graphs?**

The detail information about the GLEAM is provided in the methodological section (page 11 line 17 to 27), and obviously had several checks. The check of the product was not for a given area and not based on accurate hydrological modeling. Hence

we would not say that NewAGE over or under estimates the budgets. This assumes that GLEAM is the truth. As strictly mentioned in the methodological section, both of them are estimates, which differ but are somewhat coherent. NewAGE, in any case, forces mass to be conserved that brings into the game the whole set of hydrological measurements, and, in our opinion, can be trusted more.

**21 - The author can elect to present one or two of the Graphs/Figures but well elaborated and discussed into details. As it is, figure 4(b) is of limited value and would rather be discussed in the text or annexed.**

Figure 4b will be discussed more in detail in the text of the revised manuscript.

**22- Considering the model/data uncertainties, a KGE of 93% may be theoretically high if not good enough. There is hence a need for a strong justification of how the errors cancelled out during calibration and validation.**

We believe that KGE is high because our model is good, and, bedsides, based on available measure, the components were tested separately from the whole, when possibile. So rainfall estimation was estimated with rainfall measurements (we dedicated a paper to this). Storage was estimated against GRACE data, and so on.

**23 - Fig 5 is not well represented. This can be avoided or the authors can choose the sub- catchments to illustrate 'a prior in the methodology section' as mentioned already. The challenge here is that with the many sub catchments, the author does not seem to know how to cluster them in a consistent manner throughout the paper.**

We agree that we need to explain better what is shown in Figure 5. It seems that we did not clearly shows what we wanted. We modeled daily discharge at all river links of the basin for 16 years. The results were presented in two ways: (1) Time series simulations at few links of the river network where we have observed discharge to compare with.. These comparisons are connected to the basin river network map to show the locations

of these links within the basin (i.e. figure 5). The names of these locations is given in the caption, and information about them is also given in Table 2. (2) In figure 6, we presented a snapshot of discharge estimates for any river links of the basin. To this figure actually correspond a table which will be added in the complementary material of the revised manuscript. We think that these summarises and is the best ways to communicate our results effectively but we agree that the caption and the text can be very much improve to promote the reader understanding

**24 - The results on page 14 can be summarised and well written. On table 2, is the final outlet of Upper Blue Nile located at El Diem with an area of 174 000km2? No idea!**

We will revise the section. Yes, it is the outlet of the basin. Probably we will add a column to the table to clarify further these results.

**25 - Fig 6 on page 15 needs to be elaborated and well explained or else moved to the annex.**

Please see specific comment 23. We will add appropriate comments.

**26 - On page 16, it would be good to justify how the discharge in the entire basin was modelled. I.e. did you add/route all the upstream discharges and accumulated downwards? This as a technical consideration for the paper.**

Thank you for this, and we will add an explanation on how we modeled the discharge routing in the methodology section.

**27 - All the results needs to be discussed from a hydrological standpoint. This section is important for the authors to justify the publication, and provide key element of study that improves the knowledge in hydrology in such areas generally.**

Thank you for the suggestions you gave all through the paper.

**7. CONCLUSIONS**

**28 - The paper needs to be summarised in the context of the study. Considering the uncer- tainties, the results need to be reported with this recognition i.e. ET values between 650-750mm were estimated for various sections of the basin etc**

**There is need for more conclusions about the challenges of the study and the methods generally. This will form a basis for recommending future studies in areas with similar data limitation.**

**As it is, the section is completely lacking and does not provide future research directions in hydrology.**

We will try to improve our conclusions being more specific on uncertainties, and re-marking the challenges we met in our studies. However, we will not take responsibility to indicate future research directions. In our opinion we already show something that is a little beyond the state of art of the discipline. These improvements include the use of various satellite sources for verifying and/or assessing all the water budget terms, and the production of the same water budget at various time scale, verifying mass conservation through the cycle. Besides, we produced the software to obtain it, we made it available, and everybody can replicate our results.

**8. REFERENCE**

**29- The references are not formatted to the Journal requirements as required by HESS. Check and realign all of them.**

References formatting have corrected accordingly.

---

## Author Response (AR1)

**Rebuttal Letter to manuscript hess-2016-290:**

"Water budget modeling of the Upper Blue Nile basin using the JGrass-NewAge model system and satellite data"

Wuletawu Abera; Giuseppe Formetta; Luca Brocca and Riccardo Rigon

Dear Editor Professor Dominic Mazvimavi, and dear reviewers,

We would like to thank you for your comments and suggestions, which gave us the opportunity to improve the paper. In the revised manuscript (MS), we hope to solve all the issues raised. In this document we answer to all the reviewers questions . Comments are shown in bold font, followed by our answer/comment in normal font. The major corrections/changes in the manuscript are displayed between " ".

**Editor's comment:**

**The Reviewers have submitted very detailed and important comments about this manuscript. The authors are encouraged to submit a revised paper that ADEQUATELY addresses the comments of the reviewers.**

**The revised paper will be referred to the referees to establish if all the comments have been adequately addressed.**

Dear Editor,

We thank you for the comment given to our MS which obviously further improves the quality of our paper. In the revised manuscript (MS), we tried our best to address adequately the issues raised by the two reviewers.
* * *
**Reviewers' comment:**

**Anonymous Referee #1:**

**This manuscript proposes a method to improve water budget modelling by using the available, but sparse, hydrometerological data and satellite products. The current manuscript provides a good try to predict hydrological process in data scarce regions or ungauged basins. Although there are publications related to such topic in ungauged basins, the intent of the manuscript is worthy and significant, and is of**

**interest to readers of HESS. Seeing the potential of this study, I am in general supportive of publication if the following comments are addressed in the resubmission.**

Dear reviewer #1, we thank you for the general appreciation of our work, the comments and suggestions you give that helps to further improve our MS. In the following, your comments are answered one by one:

**Major concerns:**
**1. I would encourage the authors to rewrite the methodology section. Give a clear message to the reader what you did and how you did. For example, the manuscript entitled as 'JGrass-NewAge model system'. However, I could not find detail or key information about the method. What's the theory of the method based on? What's the advantage of the method? The headings in method section are the same as those in section 5.**

Regards to the JGrass-NewAge system, it is built on the object modeling system v3 (OMS3) informatics, which aims to deploy modern modeling solutions, with the philosophy of promoting reproducible research. The best way to have general information about it is the paper Formetta et al., 2014. JGrass-NewAGE is a collection of various modeling solutions for all hydrological compartments or fluxes. The detail of each component are presented and validated in various papers: rainfall-runoff modeling (Formetta et al. 2011), shortwave solar radiation modeling (Formetta et al. 2013), longwave solar radiation modeling (Formetta et al. 2016), and digital watershed modeling (Formetta et al. 2014b; Abera et al. 2014). We believe the level of details about JGrass-NewAge in page 4 and 5 are enough, but we revised the section for clarity. Here is the new paragraph about JGrass-NewAGE:

"UBN water budget is estimated using the JGrass-NewAGE hydrological system. It is a set of modelling components, reported in table 1, that can be connected at runtime to create various modelling solutions. Each component is presented in details and tested against measured data in the corresponding papers cited in the table 1. Similar study using JGrass-NewAge system, but using mostly in-situ observations, has been conducted in Posina river basin (northeast Italy), and the model performance is assessed positively (Abera et al., submitted). Brief descriptions on the components used in this study are provided in the following sections. In this study, the shortwave solar radiation budget component (section 3.3), the evapotranspiration component (Priestley and Taylor, section 3.3), the Adige rainfall-runoff model (section 3.4), and all the components illustrated in figure 2 are used to estimate the various hydrological flows."

**Table 1.** JGrass-NewAge system components and respective references. The components in bold are the ones used in this study.

| Role | Component Name | Description |
|---|---|---|
| Basin partitioning | **GIS spatial toolbox and Horton Machine** | A GIS spatial toolbox that uses DEM to extract basin, hillslopes, and channel links for NewAge-JGrass set-up (Formetta et al., 2014a; Abera et al., 2014) |
| Data interpolation | Kriging, Inverse Distance Weighting, and JAMI | Interpolates meteorological data from meteorological stations to points of interest according to a variety of kriging algorithms (Goovaerts, 2000; Haberlandt, 2007; Goovaerts, 1999; Schiemann et al., 2011), Inverse Distance Weighting (Goovaerts, 1997) |
| Energy balance | **Shortwave radiation**, Longwave radiation | Calculate shortwave and longwave radiation, respectively, from topographic and atmospheric data (Formetta et al., 2013, 2016). |
| Evapotranspiration | Penman-Monteith, **Priestly-Taylor**, Fao-Evapotranspiration | Estimates evapotranspiration using Penman-Monteith (Monteith et al., 1965), Priestly-Taylor (Priestley and Taylor, 1972), and Fao-Evapotranspiration (Allen et al., 1998) options |
| Runoff | ADIGE (**Hymod**) | Estimates runoff based on Hymod (Moore, 1985) algorithm (Formetta et al., 2011) |
| Snow melting | Snow melt | Modelling snow melting using three temperature and radiation based snow algorithms (Formetta et al., 2014b) |
| Optimization | **Particle Swarm Optimization**, DREAM, LUCA | Calibrate model parameters according to Particle Swarm Optimization (Kennedy and Eberhart, 1995), DREAM (Vrugt et al., 2009), LUCA (Hay et al., 2006) algorithms respectively. |

Regarding to the titles of the subsections in Methodology and in Results, the titles are the same because they refer to same water budget term (precipitation, evaporation, discharge, and storage, sequentially), and is given in both sections because we think the correspondence helps the understanding. We apologise, instead for the mistake we did in subsections' hierarchy, which is now corrected (please, see answer #12 to revier #2).

**1.a. Some parts in the results analysis and discussion section are more suitable to be in the methodology section. For instance, it would be better to introduce the indices (i.e., KGE, PBIAS, r) in section 4.**

It is true that goodness-of-fitness (GOF) indices can be in methodology section. However, since those indices are common in literature, maintaining their details in the main text is, in our opinion, distractive. That is the reason we decided to move description of the indices in the appendix section. However, we added a phrase that refers to the appendix also in the methodology section. This sentences in section 3.4 i.e. "The objective function used to estimate the optimal value of the parameter is the Kling-Gupta efficiency (KGE, Kling et al. (2012)). The KGE is preferred to the commonly used Nash-Sutcliffe efficiency (NSE, Nash and Sutcliffe (1970)) because the NSE has been criticized for its overestimation of model skill for highly seasonal variables by underestimating flow variability (Schaefli and Gupta, 2007; Gupta et al., 2009). For evaluation of the model performances, in addition to the KGE, two other goodness-of-fit (GOF) methods (percentage bias (PBIAS) and correlation coefficient) used in this study are described in Appendix A."

For validation statistics, the following sentence is added in section 3.6:

"and three goodness-of-fit (KGE, PBIAS, r) are used as comparative indices (for detailed information please see Appendix A)"

**In addition, what's the spatial resolution of the HRU? When performing simulation, what are the time step and the spatial resolution of output?**

The mean spatial resolution of the HRU is about 430 km$^2$ and we use daily time steps. This size is a trade-off between the resolution of the satellite data and the need to group some of them to have some statistical significance. The simulation results are therefore one for each HRU and at each time step of one day. The HRU estimates should be considered as a spatial average. Discharges however, are simulated at the nodes of the river networks. In the introduction section, the following phrases are added to better describe both the spatial and temporal resolution of the simulation:

"It obtains, at relatively small spatial scales and at daily time step, all the water budget components."

In addition, we have mentioned the number of subbasins used, and the mean ± standard deviation, as follows:
"In this study, the UBN basin is divided into 402 subbasins (HRUs of mean area of 430± 339 km2) and channel links, as shown in figure 1b."

"The index k = 1,2,3… is the control volume where the water budget is solved."

"The water budget components are estimated for each HRU and, subsequently, a routing scheme is applied to move the discharges to the basin outlet through the channel network." (section 3.1)

**1.b. There are different hydrometerological data and satellite products, but it is difficult to readers to obtain their information (e.g., what kind of satellite products). I would suggest the authors providing a table to show all the data and their spatiotemporal resolutions. How did you deal with the different resolutions (especially spatial resolution) of input parameters?**

A table was added as requested by the reviewer. The approach we followed on the description of the satellite products is to use a single 'best' satellite product, based, in the case of precipitation, on Abera et al., 2016. For the other water budget terms we were mostly constrained by products availability. Any product is described in the

methodological section along with the description of the methods used to estimate the component. In summary, we used SM2R-CCI for precipitation, GLEAM for ET (but we have provided appropriate comparison with MODIS in supplementary material), in-situ hydrometer data for discharge (no other choice possible), and GRACE for storage change (no other choice possible). In the revised MS, the following table describing all the satellite products used in the paper and its spatial and temporal resolutions has been added at the end of the methodology section:

**Table 1.** Short summary of the list of remote sensing products used in this study.

| Satellite products | Spatial resolution | Temporal resolution | Reference | used as |
|---|---|---|---|---|
| SM2R-CCI | 0.25 | daily | Brocca et al. (2014, 2013); Abera et al. (2016) | input for Precipitation |
| GLEAM | 0.25 degree | daily | Miralles et al. (2011a); McCabe et al. (2016) | verification for evapo-transpiration |
| MODIS ET (MOD16) | 1-km | 8-days | Mu et al. (2007, 2011) | verification for evapo-transpiration |
| GRACE TWS | 1 degree | 30-days | Landerer and Swenson (2012) | Verification for storage change |
| CM-SAF | 0.25 degree | daily | Schulz et al. (2009) | input for evapotranspi-ration component |

The methods for processing and estimating the data at each HRU level are described in methodology section for each component (section 3). The reference spatial resolution for model inputs and validation is the area of each HRU. So, for each HRU, we estimate the weighted average of the quantity weighted by how much of the pixel area overlaps with the HRU polygon. For precipitation, this comments was already mentioned at page 6 line 11, while, in the revised MS, we have added the following sentence regards to ET:

"For comparison with NewAge ET, we estimated area weighted average GLEAM ET for each HRU polygon."

**2. Discussion should be enhanced. What's the disadvantage of the method when applying in data-scarce regions with large area? For example, results of figure 5 indicated that the simulated runoffs were underestimated. What's the reason? Was it caused by uncertainties/errors in precipitation products? I could not find any quantitative information about errors of SM2R-CCI. Meteorological stations should observe precipitation, radiation, and etc. Why didn't you use them for validation and discussion?**

Unfortunately, the meteorological stations seem not to provide any further information besides precipitation. It is true that the model underestimation is most likely due to the

underestimation of SM2R-CCI, as described on the page 11 line 29. Abera et al., 2016 by comparing with in-situ observations, shows that SM2R-CCI slightly underestimates the total cumulative rainfall in the study area. i.e. "Generally, the model predicts both the high flows and low flows well, with slight underestimation of peak flows (figure 5 a), which is likely due to the underestimation of SM2R-CCI precipitation data for high rainfall intensities (Abera et al., 2016)." Additional source of error can also be caused by model inconsistency due to averaging out input data over large areas.

This sentence is added in the revised MS: "Additional source of error can also be caused by model inconsistency due to averaging out input data over large areas"

**3. The authors claimed that the JGrass-NewAGE system are described in a series of papers and not re-discussed in this manuscript. What's the difference between this study and the previous papers? What's the main contribution of this work?**

The previous papers contain description of the single components that were validating separately on other catchments of small size where there was relatively abundant ground meteorological information. Those papers cover the informatics of the system, DEM treatment and river network schematization, and finally radiation, runoff, and snow modeling.

In this paper those components are linked in a unique modelling solution and work all together cooperatively to solve the water budget closure.

In addition, another important contribution of this paper is the application of the obtained modeling solution in a large basin using various data (satellite and in-situ), which is what NewAge was originally developed for.. In poorly gauged area, modeling in our opinion, working in this way is the only way to obtain spatially distributed water resource information that can be used reliably for management purpose.

**Specific comments:**
**1. 1-21. 'up to 2000 mm per year'. It would be much clearer by adding precipitation.**

The point here is to emphasize that some parts of the Nile basin (i.e. parts in Upper Blue Nile and in the equators) receive 2000 mm per year, while others have insufficient precipitation. We rephrased:

"Most of the countries within the basin, such as Egypt, Sudan, Kenya, and Tanzania, receive insufficient fresh water (Pimentel et al., 2004). Exceptions to this are the small areas at the equators and the Upper Blue Nile basin in the Ethiopian highlands, which receives up to 2000 mm per year (Johnston and McCartney, 2010)."

**2. 3-1. It should have space between 'given' and '('. The authors should proof read the manuscript to avoid such mistakes.**

Space has been added; we removed such errors in the revised manuscript.

**3. 3-6. 'the river enters a deep a canyon' contains grammatical errors.**
Thank you for this, we corrected it. Now it is: "After about 150 km, the river enters to a deep canyon, and slowly changes direction to the south."

**4. 3-18. The elevation values show certain difference compared to those in page 2 line 3.**
Thanks you for spotting this. The one in page 2 line 3 was takes from literature value, and the page 3 line 18 was taken from SRTM digital elevation data. Since different values (small differences) were reported in various literatures, we used our SRTM value in both cases.

**5. 3-30. It may mislead to conclude 'the seasonal variability of the basin is very high' because the authors claimed that the temperature has small seasonal variability.**
We explicitly mentioned that the seasonal variability of precipitation (and evapotranspiration) is high at line 3-27, and that the variability of temperature is small at 3-28. Since it does not provide new information, we decided to remove this sentence.

**6. 4-1. Figure 1. I suggest adding units for axes (also other figures) as well as enlarging the schematic map (at least the text). What does the colour represent in figure 1b?**
We re-draw the figure to improve its clarity. The colors in figure 1b represent the mean elevation of HRU in the basin, which is now illustrated by a legend.

[Figure]

**Figure 1.** The Upper Blue Nile basin digital elevation map, along with the gauge stations (a); and subbasin partitions and meteorological stations used for simulation (b). Numbers inside the circles (figure a) designates the river gauging stations. The name of the basin referring to the numbers are provided in table 3.

**7. 4-15. It seems that the citation appeared in the first time, and 2014b should change to 2014a. The authors should proof read the manuscript to avoid such mistakes.**

The citations are in alphabetic order.

**8. 5-4. What does GIS mean? Please consider defining the abbreviation.**
Thank you for this. GIS refers to geographic information system. We have defined GIS in the revised MS.

**9. 5-9. How did you divide the basin into 402 subbasins? According to what kind of rules? I'm not sure whether figure 1b is your results or not.**

The partition of the basin into 402 subbasins is ontained by means of standard watershed partition techniques, and the specific procedures for JGrass-NewAge which are described in detail in Formetta et al., 2014 and Abera et al 2014. In the manuscript, it is also briefly presented at page 5 line 3 to 5 line. We revised the section as follows for clarity:

"The SRTM 90 m X 90 m elevation data is used to generate the basin Geographic Information System (GIS) representation. The basin topographic representation in GIS, as detailed in (Formetta et al., 2014a; Abera et al., 2014; Formetta et al., 2011), is based on the Pfafstetter enumeration (Formetta et al., 2014a; Abera et al., 2014). The basin is subdivided in Hydrologic Response Units (HRUs), where the model inputs (i.e. meteorological forcing data), and hydrological processes and outputs (i.e.

evapotranspiration, discharge, net radiation) are averaged. A routing scheme is applied to move the discharges from HRUs to the basin outlet through the channel network."

 Yes, figure 1b is the subbasin partition results as mentioned in the caption.

**10. 5-13. Figure 2 is difficult to read. The texts were small and difficult to guess their meaning. I suggest the authors redraw it.**
We have increase the text font and thickness of the lines of the figure. In addition, we revised the text for clarity by removing some technical terms (such as .CSV, G.C, F.C), as follows:

[Figure]

**Figure 2.** Workflow with a list of NewAge components (in white), and remote sensing data processing parts (shaded in grey, not yet included in JGrass-NewAGE and currently performed with R tools) used to derive the water budget of the UBN. It does not include the components used for the validation and verification processes.

**11. 6-23. Works cited in a manuscript should be accepted for publication or published already. There are many publications describing psychometric constant.**
We have replaced with appropriate citation (i.e. Brutsaert, 2005).

**12. 6-27. What's the relation between S(t) and T$_B$ in equation 3? Can you explain more?**

There is no relation between S(t) and T$_B$, at least for what related to equation (3). S(t) is the water (storage) present in a HRU. Instead, T$_B$, the Budyko time, affects the alpha in equation (3), because the value of alpha is obtained for balancing the water budget (i.e equation (1)) in such a way that after T$_B$ years the storage equals the initial one, i.e. S(TB) = S(0). This implies the use of an optimisation procedure, and such alpha is obtained together with the other parameters of the overall modelling solution (including runoff production, evapotranspiration, etc.) within the calibration procedure. Detail note on this is available to our under reviewer paper i.e. Abera et al. submitted (Advanced in Water Resources). To explicitly put some notes on relationship between s(t) and T$_B$, and description of the concept, we have added the following sentence and cited the paper under review as follows:

"In this procedure, given that S(t) is not measured, the assumption that there is null water storage difference after a long time, named Budyko's time, T$_B$ , (Budyko, 1978), is required. So, here, what is searched is a time duration (T$_B$) such that the water storage assumes again the initial value (Abera et al., submitted). Once T$_B$ is fixed, automatic calibration can be set to produce the set of parameters, including $\alpha$ $_{PT}$ and Smax, for which, besides discharge is well reproduced, is also S(T$_B$) = S(0) . In this study, T$_B$ = 6 years.."

**13. 7-26. Semicolon should be replaced with 'and'.**

Semicolon is replaced with 'and'.

**14. 8-4. What does KGE mean? Please consider defining the abbreviation.**

Thank you; in the revised MS we have introduced the KGE in the methodological section, as follows:

"The objective function used to estimate the optimal value of the parameter is the Kling-Gupta Efficiency (KGE, Kling et al., 2012)."

**15. 8-8. What does 'described in A' mean? Does 'A' represent 'Appendix'?**

Thank you, we have added Appendix before 'A'.

**16. 9-18. It is curious to use J representing precipitation. In addition, precipitation, evapotranspiration, and discharge are components of water budget. Why did you use different section headings (i.e., 5.1, 5.1.1, 5.1.2, . . .)?**

We adopted J for precipitation to be consistent with other papers of our research group (for instance, Rigon et al. 2016). Yes, there is error in the heading sections, and we revised to use the same level of heading for all the components.

**17. 9-21. I would suggest the authors adding 'the Oromia region (or other mentioned places)' into Fig.1.**

Thank you for this. However, we argue that the important idea here is to show the spatial pattern within the natural basin. We already verified that adding regional boundaries (information) makes figure 1 very crowded. It seemed better to us to delete the Oromia name from the text, as it is the only one mentioned.

**18. 10-1. Figure 3a indicates precipitation is highest in southern region. However, figure 3b showed a different pattern (i.e., east shared highest precipitation), especially in JJA.**

The two figures are different. Figure 3a shows the long-term mean precipitation as perceived by reviewer 1. Figure 3b, however, shows the level of percentage share of precipitation falls by seasons. In the east part of the basin, the highest percentage share (of its lower annual precipitation) falls in summer (JJA) in comparison to the other parts.

**19. 11-4. How and why did you select only some subbasins? Did you consider r and PBIAS (figure 4, e.g., high r and low PBIAS, and low r but high PBIAS)?**

We didn't consider r or PBIAS to select the subbasins. We select the three sub basins systematically to cover the basin spatial distribution: one from eastern, center, and western part of the basin. The following sentences has been added to clarify this:

"Figure 4 a shows the comparisons of the ET time series from 1994-2002 (aggregated at daily, weekly, and monthly, from top to bottom) between NewAge and GLEAM. The Figure specifically refers to three selected subbasins representing different ranges of elevations and spatial locations."

**20. 11-10. 'while the it tends to' contains grammatical errors.**

We removed 'the' from this sentence.

**21. 11-23. 'within the basin at the internal channels (2)'. What does '(2)' mean?**

It is changed to "(Table 2)" in the revised manuscript.

**22. 11-27. I do not think r2=0.92 is lower than r=0.93 or r=0.94. I suggest the authors to unify the index.**

It is very difficult to find similar index across all the papers. But, having PBIAS and r are relatively common, we decided to use r and PBIAS for comparison, in addition to KGE which is our primary index of model evaluation. Thank you for the comment, and here we convert the r2 index values report in literature in to r for unifying the indexes. We are also prudent to do comparison with other studies. So in this section, we just indicate the comparative performances:

"At the outlet, even during the validation period, the model is able to capture the dynamics of the basin response very well (KGE=0.92, PBIAS = 2.4, r = 0.93). The results show that the performances of the NewAge simulation are similar to the performances reported by Mengistu and Sorteberg (2012), with slightly lower PBIAS value (PBIAS=8.2, r =0.95)".

**23. 13-1. Are all the parameters unitless? Why are two [−]? Furthermore, I could not find table 1 in the context.**

The three parameters (with [-]) are unitless and for others it is length and time, which is given by [L] and [T] respectively in the table. Thanks for indicating the confusion between the two $\alpha$[-]. In the revised manuscript the first and second $\alpha$[-] has been changed to $\alpha_{hymod}$[-] and $\alpha_{PT}$ [-] respectively. The following sentence has been added in the MS to refer to the table:

"The optimized parameters of the Adige model, obtained using automatic calibration procedure of NewAge, are given at table 3."

**24. 13-2. Can you number the hydrometer stations and then add these IDs into figures 1b and 5?**

Thank you we have labeled ID both in the figure 1, table 3 and figure 5 (please see the answer to specific comment 6).

**25. 14-8. Are Wase-Tana and FlexB commonly used models? Please consider defining the abbreviation.**

It is true the two models are not common. We cited the papers where the models are described.

"Similarly, without calibration for the Gilgel Abay river, the NewAge simulation performance is better than  the results of Wase-Tana (Wosenie et al., 2014, PBIAS=34)) and FlexB  (Fenicia et al., 2008, PBIAS=77.6) or comparable to  SWAT (PBIAS=5)."

**26. 18-5. Can you provide some radiation, cloud, and wind observations? This may be better to draw the conclusion.**

We don't have observations of radiation, cloud and wind. We used JGrass-NewAge shortwave component to estimate the radiation data, together with the information of cloud fractional cover (CFC) from EUMETSAT Climate Monitoring Satellite Application Facility (CM SAF) project (Schulz et al., 2009). Wind data is not used at all in this study. It is true that including the radiation estimates and cloud data provides more insight to understand the conclusion given at this particular line. Providing spatial maps of these data in the manuscript, however, reduce its readability. Here are some samples (monthly mean for the year 1994) of the cloud cover map for the basin:

[Figure]

But also available at blog: http://ecohydrogeomorpho-metry.blogspot.it/2016/04/cloud-coveron-surface-net-radiation.html

**27. 19-9. What does S mean?**

We changed this into ds/dt.

**28. 19-11. The number of decimal places was set to 3 for precipitation. Is it necessary? I suggest the authors unify the number of decimal places.**
Of course it is not important. We removed all the decimal number throughout the paper.

**29. 21-12. 'figure' should be 'figures'.**
We changed it to 'figures'.

**30. 26-6. 'et al.'. The authors should list all the authors of a citation and unify the citation style. The authors should proof read the manuscript to avoid such mistakes.**
We corrected this and other citation errors.

**31. Texts of most of the figures are unclear. I would suggest the authors redraw the figures.**
In the new manuscript, we improved the figures for clarity.
* * *
**Anonymous Referee #2:**

**GENERAL AND IMPORTANT COMMENTS ABOUT THE MANUSCRIPT**

**The Manuscript (MS) is an attempt to integrate various sources of satellite remote sensing data towards macro-scale hydrologic modelling in a region in Africa. Such a concept is novel considering the eminent data limitations pertaining to lack or limited observed in-situ hydro-meteorological data important for model calibration and validation purposes. In this study, the authors seem to be interested in historical cases of the water budget, and hence may elect to put this is the title, or justify why they are not interest in forecasting. From the present standpoint, however, the paper can be considered for publication in the near future, but only after addressing some serious technical issues that degrade the novel concept proposed and applied by the authors. In this respect, and to improve and make the MS much better, I wish to recommend major revisions before further consideration. The following are some of the major comments that need readdress:**

We thank reviewer #2 for the appreciation of our work. When performing our studies we analyzed historical data, as any other hydrological study. We are, obviously, interested in forecasting the hydrological cycle components, but this necessarily relies on the availability of the meteorological forcings. It is possible to forecast (in the sense of meteorology) discharges (for instance) if we have rainfall (and other meteorological) data. This assumes that we have access to real time data in the basin, which we do not have. More relaxed forecast, or better, projection, could be made after acquiring

appropriate climate projections. But for this, to have a model system which is validated for a given basin is the first step. This is actually one of the goals of the present paper. However, we used as much as possible the suggestions given by the reviewer to improve our new manuscript.

**Major concerns**
**(a). Language Limitation: the MS is poorly written and generally very difficult to read right from the abstract to the conclusions. This may be due to language limitation/culture of the authors, but considering that the MS will have a bigger readership; it would be nice to English edit the MS so that the actual intentions-technical and linguistic-can come out clear. The way the results, especially the statistics and maps, are presented makes one question the objective of the work. In some cases, it is difficult to understand it the authors intend a comparative assessment at various spatial scales of the regions in the basin? There is also the random use of difficult expressions appearing from nowhere without prior definition, i.e. in defining the table in page 15, he used Figure 5, Table 2 which is difficult to understand.**

We used all the suggestions of the two reviewers, and revise the manuscript accordingly. In page 15 there are not Tables. There are Tables in page 13, and we assume the reviewer refers to them. In the revised MS, we modified the introduction to emphasize the objective and novelty of the study, and the figures are revised for clarity.

**(b). the author claim that his research is motivated by data limitation. However, he seems to have some stations with streamflow data as by the hydromet stations in the study area map or otherwise, the hydrographs used in the validation exercise. This begs the question: So where is the boundary of this data limitation he is claiming?**

Data limitation does not mean total absence of data. Certainly we have some precipitations and discharge data. However these data are in 35 locations for precipitation data in an area of 175 thousand square kilometers. Meaning, just a station every 5000 square kilometers or areas of around seventy by seventy square kilometers of side (on average). Convective processes generating precipitation can be as small as 10 kilometers square, so the optimal gauge network distribution should be as small as that, to capture all the relevant phenomena. Considering this fact, almost any region in the world is data-scarce, but some regions such as the Upper Blue Nile basin are even more data-scarce regions than others. For discharge analysis, the numbers of hydrometer stations are very few (16 hydrometers) with a data set having lots of missing data and gaps. So for the objective outlined in the study, the estimation of spatially and temporally hydrological information of the basin, UBN surely can be characterized as data limited basin.

**Could it be possible to use the available data to parameterize the model and later regionalize the model? Or is it possible to develop criteria to extrapolate the results after calibration and validation of the satellite estimates with the limited but available observed data-sets?**

Yes, this is actually what it was done. We use all the data available in a period to calibrate the model and we modeled all the data (hydrological information) by means of NewAGE in the inner points. Actually, if with regionalisation the reviewer means statistical techniques, we did not use any of them. If the reviewer asks for the transferability of our approach, we can confirm that it can be extrapolated to any basin with similar or larger size.

**The authors may also need to justify why 402 sub watershed were delineated considering the limited river gauging stations shown in the study area map.**

Even if hydrometeorological data are available in fewer stations, satellites allow us to have rainfall forcing at a much finer scale. Partition of the basins in 402 parts is functional to use all the rainfall spatial information we have, in a trade-off with a reasonable computational demand. It also serves to accounts for the morphological structure of the river network, which, obviously counts very much in forming the hydrologic response. On the latter topic, the last author co-authored some papers that can support this fact.

**If he wants to retains them, then he should define use a criteria to choose at least 10-15 sub-catchments and provide their morphometry together with the simulated values of the water balance components in the results section, for consistency and clarity. A table (and not maps) in this respect would quickly help things out here.**

If we did not clearly communicate the objective of the paper, obviously, it is our fault. However, the objective of the paper is to estimate spatio-temporally distributed water budget of the UBN basin. Hence, the methodology followed and the results presented are for the whole basin, not for only some specific sub-catchments. When in-situ data is available, that specific sub-catchment is used to verify the performance of the model estimations. In other words, to assess the discharge predictive capacity of the model, those subbasins with observed discharge data are selected (about 16 subbasins), and GOF indexes are presented a table (Table 3). But for the rest of the analysis, we wanted to do water budget closure for each subbasin in the whole basin.

**c. Considering data uncertainties, would it be wise to believe the higher model reliability and hence results?**

We considered ground measures as true. The data provided by the model solution we used show that there is consistency between discharge gauges and rainfall estimates, and the model works satisfactorily also for the validation periods. Model and data are consistent (once the model is calibrated). Abera et al (2016) tried to answer the question of the reliability of the satellite rainfall data comparing with in-situ data. We agree with the reviewer suggestion, and added the following sentences in the conclusion section:

"Despite the good results obtained, it is important to note that this study is limited by the lack of in-situ ET observation and low resolution GRACE data for confirmation of

storage. To these regards, the results of this study would benefit from basin specific assessments of ET and ds/dt RS products based on ground measurements, as done in Abera et al. (2016) for precipitation."

**The authors need a good and elaborate justification of how the errors cancelled out during the simulation.**

Errors do not cancel. When possible, any of the modelling components used was validated separately. We have checked the functioning of each of them in many other cases, as testify by our own literature (as already detailed for the reviewer #1), even if in those cases data were less scarce. In this specific case, precipitation from satellites is verified and corrected using the available few in-situ observations, storage (at least at the whole basin scale) is verified using GRACE data, discharge is verified at about 16 hydrometer stations. So we know that each component, besides implementing sound science, works fine with the appropriate data. That is what we can trust. When we calibrate hydrological model just on discharge data, parameters' values become a collector of uncertainties (a garbage collector, as some colleague calls it), but we assume that this is well understood and does not require a further disclaimer.

**Furthermore, the author seems to be using some part of the available data for calibration, and the same half plus the rest within the time frame for validation.**

We don't. We used some part of the available data to calibrate the model at the main outlet, and used the other part for validation. In addition, the other data sets available in the interior hydrometer stations are used for validation the model capability to estimate discharge at each links of the river network of the basin. This is clarified in section 3.6, as follows:

"At the basin outlet (Ethiopia-Sudan Border), the ADIGE rainfall-runoff component (i.e. HYMOD model) is calibrated to fit the observed discharge during the six years of calibration period (1994-1999) at daily time steps."

"Discharge simulation is validated for separate time-series data at the outlet at Ethiopia-Sudan Border, where the model is calibrated. In addition, the simulation of NewAge at the internal links is validated where in situ data are available. The evaluations at the internal links provide an assessment of model estimation capacity at ungauged locations."

**In my opinion, the conventional way would be to divide the data-sets into two, one for calibration and the other for calibration.**
Correct! That is what we did.

**Could this be the reason for the good efficiency realised? The authors need to justify this methodology very strongly.**
As we said, we did not use the same data for both validation and calibration. Hence, we believe that the reason for good model performance is due to the explicit characterization of inputs component and the goodness of the modeling solutions adopted.

**(1) TITLE**

**1 - The title is okay and acceptable, but may sound better if the authors consider the conventional way of staring a sentence with a verb i.e. Modeling/Estimation/Assessing of the Water Balance etc. This is however trivial at this moment.**
We agree with the reviewer. We changed the title to: "Modelling the water budget of the Upper Blue Nile basin using the JGrass-NewAge model system and satellite data"

**(2) ABSTRACT**

**2 - In my opinion, the first sentence can be made simple and realistic i.e. . . .by saying the region is one of the data scarce regions is the developing regions (but not in the world as this raise a lot of questions and may temp one to ask for proof of review in the introduction. Are there basins in the UNRB that have data? Is the justification of one of the data scarce regions in the world thus still valid?**

Yes, we have changed it: "The Upper Blue Nile basin is one of the most data-scarce regions in developing regions."

**In my opinion, the water budget components of study can be explicitly mentioned in the sentence without the brackets, and the tools used well captured and summarized. This makes the section clear and easy to read. Considering that modeling procedure employed, and the possible uncertainties involved, the results need to be rounded off i.e. by saying that precipitation values between 1000-1600mm were estimated depending on seasonality etc. Generally, the abstract can be well written and summarized in good English language, and only important content.**
We revised the sentence as follows:
"In this study we develop a methodology that can improve the state-of-art by using the available, but sparse, hydrometerological data and satellite products to obtain the estimates of all the components of the hydrological cycle (precipitation, evapotranspiration, discharge, and storage)."

We presented the uncertainty by mean plus/minus (i.e. for precipitation we used 1360 ± 230), and we prefer our to represent the uncertainties and long-term annual mean value.

**(3) INTRODUCTION**

**3 -This section can be language edited and the phrases backed with the latest references. The references also need to be ordered either from the latest to the oldest or vice versa as required by the journal.**

The following sentence is taken from the journal authors' guidelines, and states that

citation can be ordered based on relevance, and that is what we followed:

"In terms of in-text citations, the order can be based on relevance, as well as chronological or alphabetical listing, depending on the author's preference."

**4 - In my opinion, the text in lines 4-10 can be summarised and well captured within the text without using bullets or points.**
In the revised manuscript, we tried to synchronize them in shorter sentences.

**5 - Lines 27-28: the sentence beginning with [The use of RS precipitation products...] can be well written, more content added and justified. Here the authors can introduce and justify the use of other products such as GLEAM, MODIS data products etc for simulation. The author seems to neglect this section/paragraph and YET it forms the basis of their novel idea of using RS for data scarce regions. In my opinion, 'at least two paragraphs' on this section should be added to improve and justify his methodology where he has introduced a lot of RS products from nowhere. For instance, how have these RS tools and methods been applied in other regions of data scarcity? What were the results achieved? Can the methods be replicated in the current study basin? Has the JGrass NewAge (JGNA) model been applied elsewhere and what were results and strengths etc? This section should a major part of the MS and if not well captured then it can be concluded that the MS contributes very little value to hydrological science.**

**We wanted** to avoid the description of various remote sensing (RS) products, and instead suggest that the readers should look for this information in the appropriate papers about the use of RS for hydrology that we cited better in the revised manuscript. However, a review of the overwhelming number of applications of various RS products for hydrology is not the subject of the paper. The justification of the particular remote sensing data for a particular component is explained it the respective section. For instance, the justification as to why we used SM2R-CCI for precipitation is given in detail at section 3.2; the GLEAM for evapotranspiration is given at section 3.3 etc. But, we accept that the general comment on the use of RS for water budget modelling and its prospect can be commented at this section. Hence, in the revised MS, the following paragraph is added:

"To overcome data scarcity, large scale hydrological modelling can be supported by remote sensing (RS) products, which fill the data gaps in water balance dynamics estimation (Sheffield et al., 2012). For instance, a considerable number of researches has been carried out in the last two decades in developing satellite rainfall estimations procedures (Hong et al., 2006; Bellerby, 2007; Huffman et al., 2007; Kummerow et al., 1998; Joyce et al., 2004; Sorooshian et al., 2000; Brocca et al., 2014).
RS is also a viable option to fill the gaps for basin scale evapotranspiration estimation. Global satellite evapotranspiration products have been available by applying energy balance and empirical models to satellite derived surface radiation, meteorology and vegetation characteristics, and they are recognised to have a certain degree of reliability (e.g. Fisher et al., 2008; Mu et al., 2007; Sheffield et al., 2010).

Basin scale storage estimation is the most difficult task. Fortunately, the Gravity Recovery and Climate Experiment (GRACE) (Landerer and Swenson, 2012) came to fill this gap (e.g. Han et al., 2009; Muskett and Romanovsky, 2009; Rodell et al., 2007; Syed et al., 2008; Rodell et al., 2004). Guntner (2008), Ramillien et al. (2008) and Jiang et al. (2014) reviewed the use of GRACE data and positively recommended it for large scale water budget modeling. At the moment, satellite based retrievals of discharge are not available as operational or research products, but, potentially it can be retrieved from satellite altimetry and multispectral sensors (e.g. Tarpanelli et al., 2015; Van Dijk et al., 2016). Moreover, the Surface Water Ocean Topography (SWOT, Durand et al. (2010)) mission, which is expected to be launched in 2020, will provide river elevation (with an accuracy of 10 cm), slope (with an accuracy of 1 cm/1 km) and width that can be used in estimating river discharge (Paiva et al., 2015; Pavelsky et al., 2014).

Notwithstanding the availability of these RS products at various (spatial and temporal) resolutions and accuracy, their use is clearly a new paradigm in water budget closure estimations (Sheffield et al., 2009; Andrew et al., 2014; Sahoo et al., 2011; Gao et al., 2010; Wang et al., 2014)."

In the same mood, we do not want to add much information about JGrass-NewAGE that can be better accessed in previous papers by the same authors. The details provided in section 3.1 seem long enough to describe the model system. Regarding to previous applications of JGrass-NewAge, the following sentence has been added in the revised MS, at section 3.1:

"Similar study using JGrass-NewAge system, but using mostly in-situ observations, has been conducted in Posina river basin (northeast Italy), and the model performance is assessed positively (Abera et al., submitted)."

**(4) THE STUDY AREA**

**6 - There are loose statements here and there that can be tightened and generalized. For instance, in line 5, one would ask: where is Bahir Dar where the river originates? Such loose statements assume and make the MS only fit for regional publication. In my opinion, one elaborate map of topography (DEM), river network and stream gauges can be sufficient here. I am also sure with good GIS skill, and added topological data, the rainfall stations can still be added without making the map look untidy and congested. Or else, he may also elect to take a map of the catchment delineations and the rainfall stations in the methodology, and use that chance to highlight the subcatchments…**

Thank you, we improved figure 1. As suggested by the reviewer, we dedicated one map describing the DEM, river network, and stream gauges, with stream gauge stations labeled by ID number. Since the sub basins are the scale at which the water budget is estimated, we maintain this map along the former.

**7 - (better more than 10) where he wants to focus his results using a table as**

**mentioned above already.**

We do not think that adding more catchments' details is useful for the readability of the paper. However, DEM, important shape files to be used in GIS, and the list of catchments details is provided as supplementary material.

**(5) METHODOLOGY**
**8 - On page 4 lines 12-15, the authors may want to choose one or two more applicable references of the co-author.**

The lists of papers cited are describing different modeling solutions, each for one component of the JGrass-NewAge system. Since all components are used, it is important that we cited all of them. However, the sentence has been revised (see major comment #1 of reviewer #1).

**9 - In page 5, Figure 2 needs simplifications and better explanations. The color coding shades used will not appear if the paper is printed in black and white.**

Thank you, we improved the text and shadings.

**10 - Some parts in section 3.2.1 ideally belong to the introduction. Let the authors focus on the data-sets used and why they were used.**

Actually what has been written in the first and second paragraph was the explanation why and how we used SM2R-CCI precipitation data. Please see the answer for comment 5.

**11 - The reference Abera et al., submitted is completely out placed and may not be necessary at this stage of the journal.**

Since it contains similar efforts, with more details on the foundations of water budget closure studies using hydrological model, but using in-situ observations, it is helpful to cite this paper. In addition, the paper is revised and resubmitted.

**12 - There are many good ways of structuring this section in hydrology. Let the authors develop a simple and flowing structure from section 3.1. For example, section 3.1 can be titled 'Data and Methods'. Section 3.1.1 can be on 'Water Balance Modeling'. Section 3.1.2 can be on 'The Modeling System'. Section 3.1.3 can be on 'Data and Modeling Procedure' etc. The authors are free to choose what structure they want to adopt. As it is at the moment, there is too much information everywhere, a majority of which is not well captured and explained.**

We realized that sub-sectioning of section 3 and 4 went wrong. New subsections are:
3 Methodology
3.1 JGrass-NewAGE System setup
3.2 Precipitation
3.3 Evapotranspiration

3.4 Discharge
3.5 Water storage
3.6 Calibration and validation

4. Results and discussion
4.1 Precipitation
4.2 Evapotranspiration
4.3 Discharge
4.4 Water storage
4.5 Water budget closure

5. Conclusions

We think that in this way there is a clear relation between the topics of the two sections (section 3 and 4).

**13 - Some content in section 3.2.3 on page 7 are not necessary and can be avoided generally.**

Section 3.2.3 contains totally twelve lines. It is very difficult for us to understand what we can avoid to say. We give information about the algorithm we use for reproducing discharges, and the validation method. We believe that this information is necessary.

**14 - Section 4 on calibration and validation can be renamed as section 3.2 and well elaborated as explained before. In this section, the authors need to JUSTIFY WHY the same data period used for calibration is also available for Validation? This may infer a technical limitation that can affect the model results purported by the authors.**

Regarding about section renaming, please see specific comment 12. We did not use the same data for calibration and validation, as described in major comment C.

**6. RESULTS AND DISCUSSION**

**15 - Generally, the results are not balanced and well presented. The spatial maps dominate all the results. Well structured tables may provide more information considering the many catchments of study.**

Depends on the objective of the paper, the deliverability of the results need to be based on the maps. We think that one figures convey more than thousands words if well understood. Evidently we were not able to convey clearly their meaning. We have worked to improve figure captions and comments.

**16 - The first paragraph in the results section may not be necessary, or better be summarized.**

Thank you, we summarized it as follows:

 "The results of the study are organized as follows: firstly, we present the results for 1) precipitation, 2) evapotranspiration, 3) discharge and 4) total water storage; secondly, the JGrass-NewAGE system is used to resolve the water budget closure at each subbasin, and the contribution of each term water budget term  is further is analyzed."

**17 - The authors should find a way of presenting the maps in a nice, simple and clear manner. As they are at the moment, the polygons dominate the results. An elaborated table with selected catchment justified in the methodology can be good enough. Only one or two maps can be used here for visualization and overall balance of presentation of the results.**

Given our objective, the presentation of our results without maps is impossible. We limited one, if not two, figure (plot) for each component. Data are averaged over a subbasin and there is not internal spatial variability in the output. So it is clear that "polygons" stand out.

**18 - In line 23-24 of page 9, is the discrepancy small as mentioned? Could it be that the SM2R-CCI was not properly corrected? Please explain into details.**

The difference between annual long-term rainfall value of 1900 mm and 2049 mm, given by different data sources, can be considered small. Besides, if one considers the uncertainty pertinent to each data sources and estimation method, s/he should conclude that the difference is acceptable.

**19 - The legend for Fig 3 needs to be well placed and elaborated.**

We revised the legend and the caption were improved.

**20- In section 5.1.1 of page 11, there is need for technical justification by the authors as this is a very strong section of hydrology. (i) If GLEAM has had validation in other areas, with a good match with observations, then I it would be ok to use it for plausibility checks. However, as it stands, the New Age simulation of ET highly over- or under-simulate the ET fluxes. Should the results thus be fully trusted with these graphs?**

The detail information about the GLEAM is provided in the methodological section (page 11 line 17 to 27). GLEAM had several checks: "The performance of GLEAM is assessed positively in different studies (McCabe et al., 2016; Miralles et al.,2011b).

The literature checks of the product was not for a given area and were not based on hydrological modeling accurate as our. Hence we would not say that NewAGE over or under estimates the budgets. This would assume that GLEAM is the truth. As mentioned in the methodological section, both of them are estimates, which differ but are somewhat coherent. NewAge results also depend on various other inputs. However we: assessed

rainfall inputs (in another paper), check the consistency of the water budget components (such that mass is conserved) , check the consistency of data and model outcomes. Therefore we are sure that our results are quite robust in comparison with previous ones, including GLEAM's.

**21 - The author can elect to present one or two of the Graphs/Figures but well elaborated and discussed into details. As it is, figure 4(b) is of limited value and would rather be discussed in the text or annexed.**

The whole paragraph (i.e second paragraph of section 4.2) is all about figure 4b, and we believe that it constitutes a sufficient comment.  However, we revised the text as follows:

"The agreement/disagreement between the two estimations varies from subbasin to subbasin (figure 4). The spatial distribution correlation and PBIAS between the NewAge and GLEAM ET is presented in figure 4 b. Spatially, the correlation between JGrass-NewAGE and GLEAM is higher in the eastern and central parts of the basin, while it tends to decrease systematically towards the west (i.e. to the lowlands, see figure 4 b). The correlation between the two ET estimations increases when passing from daily to monthly time steps. The PBIAS between the two estimates ranges from -10% to 10%, with large numbers of subbasin being from -3% to 3%. Spatially, the comparison shows that GLEAM overestimates ET in the western parts of the basin (border to the Sudan) and underestimates ET in the northern parts of the basin (figure 4b). The overall basin correlation is 0.34±0.07 (daily time step), 0.51±0.08 (weekly time step), and 0.57±0.10 (monthly time steps). Generally, except at daily time step, the two estimates have acceptable agreements (very low bias, and acceptable correlation). However, in comparison with the correlation (0.48±0.15) and PBIAS (14.5±18.9%) obtained between NewAge ET and MODIS ET Product (MODET16), as shown in the supplementary material, the correlation and PBIAS between NewAge ET and GLEAM ET is much better."

**22- Considering the model/data uncertainties, a KGE of 93% may be theoretically high if not good enough. There is hence a need for a strong justification of how the errors cancelled out during calibration and validation.**

The modeling components were tested separately from the whole, when possible. So rainfall estimation was estimated with rainfall measurements (we dedicated a paper to this). Storage was estimated against GRACE data, and so on. We do not believe that model/data uncertainties cancel each other. A better hypothesis is that the calibration procedure is able to mask systematic measurement errors.

**23 - Fig 5 is not well represented. This can be avoided or the authors can choose the sub- catchments to illustrate 'a prior in the methodology section' as mentioned already. The challenge here is that with the many sub catchments, the author does not seem to know how to cluster them in a consistent manner throughout the paper.**

We agree that we need to explain better what is shown in Figure 5. It seems that we did not clearly show what we wanted. We modeled daily discharge at all river links of the basin for 16 years. The results were presented in two ways: (1) Time series simulations at few links of the river network where we have observed discharge to compare with.

These comparisons are shown in the river network map to visualize the locations of these links within the basin (i.e. figure 5). The names of these locations are given in the caption, and information about them is also given in Table 2. (2) Figure 6, now moved to the supplementary material, presents a snapshot of discharge estimates for any river links of the basin. We tried to improve the Figure caption to help better the reader understanding

**24 - The results on page 14 can be summarised and well written. On table 2, is the final outlet of Upper Blue Nile located at El Diem with an area of 174 000km2?**
No idea!
We revised the section. Yes, it is the outlet of the basin. We have added a column to the table that connects the table with the spatial location in figure 1.

**25 - Fig 6 on page 15 needs to be elaborated and well explained or else moved to the annex.**
We moved figure 6 to the supplementary materials, as it does not provide any comparison or statistics with observations. However, it shows how we can obtain the discharge at each links.

**26 - On page 16, it would be good to justify how the discharge in the entire basin was modelled. I.e. did you add/route all the upstream discharges and accumulated downwards? This as a technical consideration for the paper.**

Thank you for this. In the methodological section (section 3.4) and the following sentence has been added to explain how the discharge routing is modeled:

"The NewAge Hymod component is applied to any HRU, in which the basin is subdivided and the total watershed discharge is the sum of the contribution of each HRU routed to the outlet."

**27 - All the results needs to be discussed from a hydrological standpoint. This section is important for the authors to justify the publication, and provide key element of study that improves the knowledge in hydrology in such areas generally.**

Thank you for the suggestions you gave all through the paper. We used all of them to improve the paper.

**7. CONCLUSIONS**

**28 - The paper needs to be summarised in the context of the study. Considering the uncertainties, the results need to be reported with this recognition i.e. ET values between 650-750mm were estimated for various sections of the basin etc**

**There is need for more conclusions about the challenges of the study and the methods generally. This will form a basis for recommending future studies in areas with similar data limitation.**
**As it is, the section is completely lacking and does not provide future research directions in hydrology.**

We revised the conclusion section being more specific on our results and uncertainties, and remarking the challenges we met in our studies (see the marked-up MS). However, we do not take responsibility to indicate future research directions. In our opinion we already show something that is a little beyond the state of art of the discipline. These improvements include the use of various satellite sources for verifying and/or assessing all the water budget terms, and the production of the same water budget at various time scale, verifying mass conservation through the cycle. Besides, we produced the software to obtain it, we made it available, and everybody can replicate our results.

[revised manuscript text omitted]

---

## Referee Report (RR1)

The authors have addressed some comments/suggestions and made changes to the paper, which has improved its quality considerably. However, I still have some major concerns:

1.  Based on the following response and the revised manuscript, I cannot agree with the authors that their contribution was linked all the components in one model. Because there's no field observation to validate the modeled water components (except runoff) in this manuscript. Evapotranspiration (ET) was estimated using the Priestley and Taylor (PT) Formula. But GLEAM, which was also estimated based on the PT method, was adopted to validate the estimation. If estimates of each component have large uncertainty, who believe the final output?

**3. The authors claimed that the JGrass-NewAGE system are described in a series of papers and not re-discussed in this manuscript. What's the difference between this study and the previous papers? What's the main contribution of this work?**

The previous papers contain description of the single components that were validating separately on other catchments of small size where there was relatively abundant ground meteorological information. Those papers cover the informatics of the system, DEM treatment and river network schematization, and finally radiation, runoff, and snow modeling.

In this paper those components are linked in a unique modelling solution and work all together cooperatively to solve the water budget closure.

In addition, another important contribution of this paper is the application of the obtained modeling solution in a large basin using various data (satellite and in-situ), which is what NewAge was originally developed for.. In poorly gauged area, modeling in our opinion, working in this way is the only way to obtain spatially distributed water resource information that can be used reliably for management purpose.

2.  As mentioned in my previous comments, the discussion section should be enhanced. However, the authors revised limited content in section 4 (Results and Discussion) in this revision (track change in 'hess-2016-290-author_response-version1.pdf'). For example, it seems that the authors kept discussing some water components in other submitted manuscript. Since the authors think combine modeling all the water components is important, why not focus on this study and discuss the possible limits of such a solution? I know the study area is data-scarce and validation using field observation is impossible. However, reasonable discussion on the comparison and the possible errors may persuade the readers (e.g., see specific comments 5, 9, 11, 14, 15, and 18 for detail). Otherwise, the readers may question about the modeled water components and finally your method.

3.  Both reviewers mentioned figure quality. However, the authors may have paid little attention. For example, although units were added to axes in Fig. 1, units for axes were still missing in the rest of figures (e.g., Figs. 3, 4, 7, 8).

Following specific comments may help the authors understanding my concerns and improving the manuscript.

The numbers in front of the comments indicate page and line number.

1. 1-6. 'to obtain the estimates of all the components of the hydrological cycle (precipitation, evapotranspiration, discharge, and storage)'. It would be better to revise the claim, because precipitation, evapotranspiration, discharge, and storage are main components. For example, interception and infiltration are also components of the hydrological cycle, which you did not address.

2. 4-3. 'Specifically studies' contains grammatical errors. I encourage the authors to check the entire manuscript carefully to avoid such mistakes.

3. 4-8. Please correct the unit for temperature.

4. 5-3. Table 1. I cannot found any description on 'JAMI' or 'three temperature' in the manuscript. If you decide putting such information in the manuscript, please be sure the readers can understand it or find relate context.

5. 8-5. Eq. (3). In table 1, the author said PT method was used as one method to estimate ET. However, 'S(t) and Smax' were added to Eq. (3). Is Eq. (3) valid for this study or is it used for all JGrass-NewAGE application? Furthermore, what's the advantage of using water storage information when estimating ET, especially in data-scarce regions?

   In addition, GLEAM estimates are based on the PT method. Can it be used to 'validate' Eq. (3)? Which version of GLEAM did you use?

6. 9-9. Are 'the ADIGE model', 'the well-known HYMOD model', and 'The NewAge Hymod' the same? If yes, please consider unify the description.

7. 9-12. I'm confused by the description that 'The main inputs for the ADIGE model are J(t) and ET(t)', because 'Q is modelled as functions of basin water storage'. How did you get the water storage? What are the five calibration parameters?

8. 10-17. ET validation is questionable. See specific comment 5.

9. 10-21. ds/dt validation. Similar to ET validation, lacking of discussion on GRACE product and the modeled ds/dt. Did you use the GRACE product directly or

perform any correction? As reported by studies (e.g., Long et al., 2015, Water Resour. Res., 51, 2574–2594), GRACE data are noisy in smaller basins less than the GRACE footprint (~200,000 km$^2$), as well as in areas with intensive irrigation. Considering the UBN is approximately 176,000 km$^2$ and the highlands have high water demands for irrigation, the product may include typical errors. The author should discuss such uncertainty and the possible impact on the modeled ds/dt.

10. 10-26. The headings in section 4 are the same as those in section 3, and the authors claimed that in this way there is a clear relation between the topics of the two sections. I cannot agree with them. For example, in rainfall section, the spatial distribution was described and then compared with some published results. But for ET section, both spatial and temporal distributions were presented, as well as 'validation'. I don't think the headings in results section reflecting any useful information.

11. 11-1. Table 2. The unit for SM2R-CCI's spatial resolution is missing. Could you provide time periods for these data used? Little information about the used data was presented.

In section 4.1, can you discuss what's the difference between corrected and uncorrected SM2R-CCI products? That is how systematic error (bias) of SM2RCCI affects rainfall amount, as some ungagged basin has no in-situ observation to perform the correction.

12. 14-6. The section is to 'validate' NewAge ET. I'm not sure why the authors talk about GLEAM estimation.

13. 14-10. A better correlation between NewAge ET and GLEAM is because they both estimated using the PT method.

14. 14-25. Table 3. It's difficult for readers to know what these parameters represent. Furthermore, $\alpha_{PT}$ has a value of 2.9. It's relatively high compared with the commonly used value (1.26) in the PT method, or the value (1.5-1.8) recommended for estimating ET in more arid regions (ASCE. 1990. Evapotranspiration and irrigation water requirements. ASCE. Manuals and reports on engineering practice. No. 70. New York, NY, USA). In this case, ET may be

overestimated. It can be seen from Fig. 4 and Fig. S3 that the NewAge ET higher than GLEAM and MODIS ET, especially the peak value. If ET is overestimated, runoff should be underestimated when precipitation unchanged. Fig. 5 did show that in most cases, the modeled runoff is smaller than the observation, and obvious difference occurred also at peak values. The authors should discuss such physical processes that may cause model uncertainty. Only insightful analyses and discussion on the mechanism behind can highlight the scientific merit of the manuscript.

15. 15-1. Table 4. There may be something interesting, i.e., KGE varied with basin area and may have a poor correlation with area. It's often taken for granted that a hydrological model will perform much better in relative smaller basins than in larger ones. Can you discuss why some times the JGrass-NewAGE System performed good or bad in sub basins with similar area?

16. 16-1. Fig. 5. Is there any observation used in Fig. d?

17. 18-13. What does S mean? Please consider defining the abbreviation.

18. 21-1. Fig. 9. It would be better to change the water components to percentage.

19. 22-1. Fig. 10. I'm curious why ET was so low in the hot season. Supposing most of the rainfall infiltrated into the deep soil (high ds/dt values) in the hot and wet seasons, can they evaporate easily in the dry season? Again, more discussion may be required to persuade the readers about the modeling results.

---

## Author Response (AR2)

**Rebuttal Letter related to manuscript hess-2016-290:**
**"Water budget modeling of the Upper Blue Nile basin using the JGrass-NewAge model system and satellite data"**

By Wuletawu Abera; Giuseppe Formetta; Luca Brocca and Riccardo Rigon

Dear Editor Professor Dominic Mazvimavi,

We would like to thank you for your comments and suggestions, which gave us the opportunity to further improve the paper. In the revised manuscript (MS), we hope to solve all the issues raised. In this document we answer to all the reviewers questions. Comments are shown in bold font, followed by our answer/comment in normal font.

**Editor's comment:**

**Comments to the Author:**
**The authors have not adequately addressed the comments of the reviewers. I agree with the comments provided in the second review that some important issues have not been addressed. The manuscript still contains a lot of unclear statements or phrases. Some specific comments are given below.**

Dear Editor,

We thank you for the comment given to our manuscript (MS) which obviously further improves the quality of our paper. In the revised MS, we tried our best to address adequately the issues raised by the reviewers.
* * *
**1. The authors should avoid paragraphs made up of one sentence.**

This was done. There are no more one-sentence paragraphs in the new manuscript. They were joined as necessary.

**2. The authors need to check the correct usage of the colon and semi-colon.**

Usage of the column and semicolon was revised.

**3. The authors still maintain the phrase that their model estimates "all components" of the water budget, when they are only dealing with precipitation, evapotranspiration, runoff, and subsurface storage of water. Other components of the water budget such as interception, groundwater recharge, groundwater**

**discharge, etc. are not covered in the model. One of the reviewers pointed out this, but the revised manuscript still uses "all components".**

We are sorry for having neglected this. We modified the manuscript by specifying which components of the budget we were talking about. However, we believe that what the water budget is composed depends on the control volume used to estimate it. At the scale we are working the fluxes are those we estimate with the exception of the groundwater flow at the outlet, which is assumed negligible. In our case, interception is an internal flux and does not appears in the budget, as well as groundwater recharge.

**5. What does "To this scope" in Line 7 of the Abstract mean?**

The scope we are talking about in the old manuscript is to develop a new methodology about improving the estimation of the water budget components that appears in our equation (1). We changed the wording as: "To obtain the water budget closure"

**6. Line 12 of the Abstract, what does "long term mean budget" mean?**

The words "long term" are unessential, and we cut them out.

**7. Line 13 of the Abstract, what does "Evapotranspiration covers 56% of the yearly budget" mean? Same applies to the % given in this line**

Possibly we do not properly interpret this observation. The symbol "%" means "per cent or percentage" (https://en.wikipedia.org/wiki/Percentage). This was specified at the first appearance and the symbol was replaced everywhere with this phrase. In place of "budget" we wrote "water budget" to mean percentage of what. In case if the word "cover" is the problem, we replace with "account for". So now the sentence is "Evapotranspiration accounts for 56% (per cent) of the annual water budget, runoff is 33%, storage varies from minus 10% to plus 17% of the water budget."

**8. Page 1, line 21, what does "2000 mm per year" refer to? Presumably precipitation. It is also possible to receive inflows equivalent to 2000 mm/yr**

Yes, the reviewer is correct, it refers to precipitation. It was added to the text: "of precipitation".

**9. Page 1 line 22, the claim that the basin is one of the most complex in the world has no basis.**
The six points listed in the old manuscript are the base for saying UBN basin is probably one of the most complex. But we removed the phrase and revised the paragraph as suggested. (See specific comment 11)

**10. Page 1, last line, 85% of what?**

It refers to discharges, and now this is specified in the text.

**11 Page 2, lines 1 – 8 are not well written. What do you mean by "diplomatic discussions" when referring to management of transboundary water resources?**

We changed the whole paragraph as follows:

"In Ethiopia, UBN is inhabited by 20 million people whose main livelihood is subsistence agriculture (Population Census Commission 2008). The Ethiopian government, therefore, has started many water resource development projects, such as irrigation schemes and dams, among which the Grand Ethiopia Renaissance Dam (GERD), which, upon completion, will be one of the largest in Africa. However, as the principal contributor (i.e 51% of discharge) to the main Nile basin, UBN also supports hundreds of millions of people living downstream, and it is referred to as the "Water Tower" of northeast Africa. Therefore UBN is a part of trans-boundary river, and its development and management require obtaining agreements between many national governments and also non-governmental organizations, each involving different policies, legal regimes, and contrasting interests."

**12. Page 2, line 9, what are you referring to by "all these facts".**

We mean the facts and challenges enumerated in the previous paragraph (in the old version). However, in the new manuscript, the entire paragraph was rewritten, and this particular sentence is revised as:

"Tackling all these complexities and developing better water resource development strategies is only possible by gathering quantitative information (Hall et al., 2014)."

**13. Page 3, line 9, what are you referring to by "aforementioned problems"?**

We specified as: "aforementioned management problems by resolving"

**14. Page 3, line 10, "resolve the water budget" is not a very clear phrase. If you intend to estimate the components of the water budget, then clearly state this.**

This was accomplished, as can be seen, in the new version of the manuscript. It is re-written: "It obtains, at relatively small spatial scales and at daily time step, groundwater storage, evapotranspiration, discharges in such a way to satisfy the water budget equation."

**15. Page 3, line 11, which previous studies do you wish to improve?**

Clearly the studies are those cited in the paper (page 2, line 9-23), where we collect all what is available, to our knowledge, for the study area. And the limitation of those papers are mentioned on page 2, line 12-23. However, mentioning "previous studies" is unessential here and was eliminated. These improvements were already highlighted in Conclusions.

**16. Page 3, line 11-12, be explicit about the components of the water budget that you are attempting to estimate.**

In the revised manuscript we are more explicit (see the specific comment 14).

**17. Page 3, line 24, you state that the basin covers "17% of the total area of the country", which country are you referring to since you are dealing with a transboundary river?**

Country is clearly Ethiopia, since the Blue Nile area studied is entirely in Ethiopia, even if the Nile is a transboundary basin. We correct the text accordingly.

**18. Page 3, line 26, you state that "the UBN basin has the lion's share of the total Nile flow". Do you mean that this basin receives or gets the largest amount, because this is what this phrase means?**

We change the phrase as: "Since the UBN basin gives the largest contribution to the total Nile flow, it is the economic mainstay of downstream countries (i.e. Sudan and Egypt)."

**19. Page 3, line 29, what is a topographic distribution?**

We mean the topography, so we changed the phrase into:
"the maps of elevation"

**20. Page 4, is the information given in line 1 - 8 relevant to this paper?**

The section was describing what controls regional precipitation. In the revised manuscript, we removed it accordingly.

**21. Page 4, Equation (1), while authors have the liberty to use any letters or symbols to represent variables, it helps readers if authors use commonly used letters and symbols. It is common practice to represent precipitation by P. the use of J is rather unusual, although there is nothing wrong with this. This just gives readers a hard time remembering what J represents in your paper.**

We understand this. However, as stated before, we adopted J for precipitation to be consistent with other papers of our research group, and some of the papers cited in the manuscript (for instance, Rigon et al. 2016, Abera et al. 2017).

**22. Page 5 line 1, it would have been much easier to refer to Qki(t) as inflows from upstream HRUs.**

We have changed the text from "Qki(t) is the discharge from the contributing streams." to "Qki( t) are inflows from upstream HRUs."

**23. Page 5, reviewers highlighted that it is not clear whether the "JGrass-NewAge system" is a new model the authors developed or are utilising. The revised manuscript does not enable readers to understand whether this is a modelling system that the authors have developed or not.**

List of papers are cited showing the development of JGrass-NewAge system. Some of the authors developed the model system, but in this manuscript an application of the model system is described. We believe that in this paper we did not mention anything that indicates the development of the model system. The focus of the paper is better described also in the general answer to Reviewer #1.

**24. Page 6, line 5, what were the criteria used to define a Hydrological Response Unit. The definition of an HRU will differ depending on the type of model used, and the problem to be solved. As this is the fundamental modelling unit, the manuscript should have provided this information. It seems that the authors subdivided the basin into sub-basins using a DEM. These sub-basins cannot be said to be HRUs. An HRU has a very specific meaning and not necessarily the same as a sub-basin.**

While the term "subbasin" and "HRU" are defined in various models in different ways, here we mean that the subbasin partition used for modeling. In our paper it is the computational unit assumed to be homogeneous in estimating the terms of the water budget we deal with.
The level of details of our HRUs is based on the problem under investigation and the data availability. For instance, if forcing input is available at few meter resolutions or enough in-situ observation density is available to generate spatially interpolated input data, then the HRU can be as small as few kilometers.

The general procedure to obtain sub basins (HRUs) is more or less the same in all GIS environments. The partitioning of the basin into subbasin particular to specific procedures for JGrass-NewAge is described in detail in other studies (Formetta et al., 2014 and Abera et al., 2014). So we thought that adding description of each step of subbasin extraction is not necessary for this paper.

**25. Page 6, line 8, what was the rationale for dividing the basin into 402 sub-basins? Reviewers made the same comment which has not been addressed. Is a sub-basin**

**similar to an HRU? In some models, a sub-basin is made up of possibly more than one HRU.**

The rationale for dividing the basin into 402 subbasin is to capture the spatial heterogeneity of precipitation data provided by the satellite product used. The following sentence was already available (page 6: line 9-10) to describe this rationale:

"This spatial partitioning may not be the finest scale possible, however, considering the size of the basin and model input data resolutions, and the resolution of satellite products, it can be considered an acceptable scale to capture the spatial variability of the water budget."

However, to make our thinking more clear, we wrote in the revised manuscript:

"This spatial partitioning may not be the finest scale possible, however, considering the size of the basin, it is consistent with input data resolution, including satellite products, meaning that a finer subdivision would imply uniform inputs for adjacent HRUs, and a coarser one would average out inputs variability."

Yes, in SWAT HRUs are constituted of a subbasin. In NewAge, HRU and subbasin are the same and are interchangeably used in this manuscript. For clarity, the following sentence has been added in section 3.1:

"In this paper, the term HRU actually identifies subbasin."

**26. Page 3 Section 3.2 Precipitation. The authors should have provided information about how accuracy of the precipitation product they used. Provide the error level established in similar environment by previous studies.**

The precipitation product used (SM2R-CCI) is assessed in various regions. SM2R-CCI for the particular study area is already the subject of our previous paper (Abera et al 2016).

In the original MS this information was given as:
"Recently Abera et al. (2016) compared five of them with high spatial and temporal resolutions over the same basin. It was shown that SM2R-CCI (Brocca et al., 2013, 2014) is one of the best products, particularly in capturing the total rainfall volume."

Based on the suggestion given to add more information on quality of SMR-CCI precipitation product, more citations are added as follows:

"Currently there are several satellite rainfall estimates (SREs) available for free, and Abera et al. (2016) compared five of them with high spatial and temporal resolutions over the same basin. It was shown that SM2R-CCI (Brocca et al., 2013, 2014) is one of the best products, particularly in capturing the total rainfall volume. Regards to the quality of

SM2RAIN-based products, recent studies positively assessed their accuracy on a regional (Brocca et al. 2016; Ciabatta et al. 2017) and a global (Koster et al. 2016) scale."

**27. Page 6 and 8, does ET in Equation (2) refer to reference evapotranspiration? In Equation (3) ET(t) seems to represent actual evapotranspiration. You need to clearly state this as you are using almost the same letters to represent different variables.**

Thank you for this, we have changed *ET* in equation 2 into *PET* this to represent the Priestly and Taylor potential evapotranspiration, and it is given in the description of the equation terms.

**28. Page 8, line 6, S(t) is not groundwater but subsurface storage of water which includes soil water. Smax should be maximum subsurface storage capacity, not just maximum storage capacity.**

We changed the text accordingly to the Editor suggestions. However, in our case, S(t), by construction, includes also the groundwater storage.

**29. Page 8, line 11, how was that automatic calibration done? What was the objective function used?**

Particle swarm (PS) optimization is used to optimize both the rainfall-runoff and ET alpha parameters. It was used to fit the discharge observations and optimize ET to close the water budget after a certain time, named Budyko time (and indicated by $T_B$). Essentially, two objective functions are used, KGE for optimizing discharge, $S(t) = 0$, for closing the water budget after $T_B$. The description for discharge calibration was given at page 9, lines 14-19. In this section, the calibration procedures for two parameters of ET, $\alpha_{PT}$ and Smax , were described instead. It was:

"Once $T_B$ is fixed, particle swarm (PS) automatic calibration can be set to produce the set of parameters, including $\alpha_{PT}$ and Smax, for which, besides discharge is well reproduced, is also $S(TB) = S(0)$ ."

Now we have revised the sentence
"Once $T_B$ is fixed, automatic calibration set to produce the set of parameters, including $\alpha_{PT}$ and Smax, for which, besides discharge is well reproduced, is also $S(TB) = S(0)$."

**30. Page 9, 3.4 Discharge. The authors should have given a basic description of the ADIGE model used. It does not help readers to refer to a paper that has not been published. The reviewers highlighted this problem, and this has not been addressed.**

ADIGE model is based on the well known Hymod model, and we thought that detailed description of the model was not useful in this paper. The paper Abera et al. 2017 is now published on-line in Advances in Water Resources (and scheduled for the June printed version of the Journal). However, based on the reviewer suggestion, we have added some

level of details for Hymod description in the Appendix A, and cited the Appendix in the main text as follows:

At the main text, it was given as follows:

"Detailed descriptions of HYMOD implementations in the NewAge model system are given at Formetta et al. (2011) and Abera et al. (submitted)."

And now it is revised as follows:
"Detailed descriptions of HYMOD implementations in the NewAge model system are given in Formetta et al. (2011), Abera et al. (2017) and summarized in Appendix A"

Then, in appendix A, the following details about Hymod has been given:

"The NewAge system executes one Hymod model for each HRU, and routes water downslope. A detailed description of Hymod model is provided in many previous studies (Moore, 1985; Van Delft et al., 2009; Boyle et al., 2001; Formetta et al., 2011). In Hymod, each HRU, is supposed to be a composition of storages of capability *C[L]* according to distribution (Moore, 1985):

$$F(C < c) = 1 - (1 - \frac{c}{C_{max}})^{B_{exp}} \tag{A1}$$

Where *F(C)* represents the cumulative probability of a certain water storage capacity (*C*); $C_{max}$ is the largest water storage capacity within each HRU and $B_{exp}$ is the degree of variability in the storage capacity. As shown in the schematic diagram (figure 11), the precipitation exceeding $C_{max}$ is send directly to the volume generating surface runoff. If we call the precipitation volume in a time interval $\Delta t$ , J(t) := P(t) $\Delta t$ , then this "direct" runoff can be estimated according to:

$$R_H(t) = \max(0, J(t) + C(t) - C_{max}) \tag{A2}$$

Where *C(t)* defines the fraction of storages already filled at time t . The latter equation is true for any precipitation and storage level, even when the maximum storage $C_{max}$ is not exceeded. When precipitation does not exceeds $C_{max}$ runoff volume can be produced by filling some of the smaller storages. To which extent this happens, can be derived by the knowledge of the storage distribution, eq. (A1), the initial storage *C(t)* and the precipitation *J(t)*. This residual runoff is, in fact, given by:

$$R(t) = \int_{C(t)}^{\min(C(t)+J(t),c_{max})} F(c) \, dc \tag{A3}$$

An analytic expression for the integral in eq. (A3) is available, which makes the computation easier. Water in storage is made available to evapotranspiration. Water going into the runoff volume, i.e. *R(t)* and *RH(t)*, is further subdivided into a surface runoff volume and subsurface storm runoff. Surface runoff, in turn, is composed by the whole of $R_H(t)$ and part of *R(t)* , and *R(t)* is split according to a partition coefficient α

such that the part $\alpha R(t)$ goes into surface runoff volume and $(1-\alpha)$ into the subsurface runoff volume. In Hymod, $\alpha$ is a calibration coefficient.

Finally, surface runoff volumes are routed through three linear reservoirs, and, subsurface storm runoff volume is routed through a single linear reservoir. A summary of equations for the surface runoff is therefore:

$$\frac{dS_1(t)}{dt} = \alpha R(t) + R_H(t) - kS_1(t) \qquad\qquad Q_1(t) = \frac{S_1(t)}{k} \qquad\qquad \text{(A4)}$$

where $S_1[L^3]$ is the storage in the first of the linear reservoirs, and $k[T]$ is the mean residence time in each of the reservoirs. Then:

$$\frac{dS_i(t)}{dt} = Q_{i-1}(t) - kS_i(t) \quad Q_i(t) = \frac{S_i(t)}{k} \qquad\qquad \text{(A5)}$$

for the other two reservoirs, where $S_i$ [L] with i = 2;3 is the storage in the two remaining surface reservoirs. Subsurface storm runoff is then modeled by:

$$\frac{dS_{sub}}{dt} = (1-\alpha)R(t) - k_{sub}S_{sub}(t) \qquad\qquad \text{(A6)}$$

where $S_{sub}$ $[L^3]$ is the storage in the subsurface storm-flow system and $k_{sub}[T]$ is its mean residence time. A budget equation can be written for the groundwater system as:

$$\frac{dS_g(t)}{dt} = (J(t) - R(t) - R_H(t)) - ET(t) - Q_g(t) \qquad\qquad \text{(A7)}$$

where $S_g(t)[L^3]$ is the groundwater storage, and $Q_g(t)$ the groundwater flow which becomes surface flow at the closure of the HRU.

Summarizing, Hymod subdivides each HRU into three reservoirs: a groundwater reservoir, from where evapotranspiration and groundwater flow is allowed, a subsurface storm-water reservoir, and a surface runoff reservoirs set. Partition of precipitation into the three reservoirs is obtained by a calibration coefficient $(\alpha)$ and the use of a probability distribution function of storages' capacity, $F(c)$.

[Figure]

**Figure 11.** Schematic diagram of hymod model (adapted from Van Delft et al. (2009))

"

**31. Page 9, line 18, explain the exact objective function used. At this stage, readers do not have knowledge about which parameters were being calibrated.**

In the revised MS, we have added brief description of Hymod model and its parameters in the appendix section, and this is cited on page 9 line 13.

It was
"The ADIGE rainfall-runoff has five calibration parameters, and the calibration is performed using the particle swarm (PS) optimization."

It is revised as follows:

"The ADIGE rainfall-runoff has five calibration parameters ($C_{max}$, $B_{exp}$, $\alpha_{Hymod}$, $R_s$, $R_q$, see the details in Appendix A), and the calibration is performed using the particle swarm (PS) optimization method."

**31. Page 10, line 6,how many rainfall stations were used to correct the precipitation estimates? Was this done in this study?**

In the basin and around, there are about 33 stations, and the procedure of error correction for various satellite precipitations was presented in our previous paper (Abera et al. 2016). Here, the same procedure is applied to SM2R-CCI that is used as model input.

**32. Page 10, line 12 -13, be explicit regarding the data used for validation. Reviewers highlighted this problem. How many river flow stations were used for validation? How were they selected?**

For discharge validation, we used all the available measurement stations, except the one at the basin outlet (Ethiopia-Sudan border), which is used for calibration. They are about 15 stations. They are portrayed at page 4 (figure 1a). Validation results at page 15 (table 4). We believe this could be enough but, we can highlight the number of stations used for validation in this particular sentence as follows:

It was:
"In addition, the simulation of NewAge at the internal links is validated where in situ data are available."

Now its is revised as follows:

"In addition, the simulation of NewAge at the internal links is validated where in situ data are available (15 discharge measurement stations)."

**33. Page 12, Figure 3b. The use of % share to express amount of rainfall received in a season is rather confusing. Is the seasonal contribution important? This is also done on Page 11 when describing precipitation**

Yes, seasonal variability is really important, probably more important than spatial variability. The contribution of each season to the total amount of water received is a topic on which other studies are less explicit about.

**34. Page 13, given that the estimation of actual evapotranspiration involves some uncertainty, which ET rates are acceptable, GLEAM or your modelled values, and why?**

We believe that given (1) improved procedure to chose the best rainfall satellite products (i.e. based on Abera et al. 2016), (2) the bias correction procedures applied, (3) the good model performances obtained at the 15 internal stations where discharge is measured, (4) the procedure of water budget closure to conserve mass, and (5) the independent evaluation using GRACE storage change, our modeled ET is expected to be more accurate than GLEAM estimates.

**35. Page 14, Table 3, since the structure of the model has never been presented, a reader does not have an idea about the meaning of the parameter values given in this table.**

As mentioned above, we have added brief note about Hymod model and its parameters in Appendix A. The following phrase has been added to Figure's caption: "Parameters' physical meaning is explained in Appendix A"

**36. Page 15, the manuscript describes some forecasting done. It is not clear how this was done. Reviewers also raised this point.**

The word "forecasting" is used to mean that given precipitation estimates and the model calibrated at some location (in our case the outlet), we give predictions of the water discharge at internal basin points. However, this comment tells us that it could provide misunderstanding to the reader; hence, we changed "forecasting" with "simulation".

**Reviewer's comment:**

**The authors have addressed some comments/suggestions and made changes to the paper, which has improved its quality considerably. However, I still have some major concerns:**

**1.a Based on the following response and the revised manuscript, I cannot agree with the authors that their contribution was linked all the components in one model.**

Obviously, if the contribution of this paper had been just "linking components" our contribution would have been poor. As we stated better in the revised manuscript, the contribution of this paper is to estimate the water budget flows and storages in a semi-distributed way in a poorly gauged area with the help of sound modeling and remote sensing products (RS). Sound modeling means, in our case, using JGrass-NewAge modeling components, each one tested separately in several other studies, and search for the closure of the water budget. The latter information, in our opinion, adds credibility and consistency to ours results.

Specifically to UBN,

- Such tasks were never accomplished in literature at the spatial and temporal resolution we use, if not for limited parts of the basins.

In order to obtain this, we used all the resources types one can have:

- Various remote sensing precipitation products (eventually using a single one after a comparative study, i.e. Abera et al., 2016), supported by 33 in situ meteorological stations.
- Discharge observation at 16 locations,
- Evapotranspiration from two satellite products (MODIS and GLEAM),
- And storage change at the basin scale from GRACE RS.

We claim that no better can be done for UBN in this moment, and, vice-versa, our study can be the seed to promote more measurements to support more refined studies.

- Besides we claim that the procedure we followed can be easily transported in any other poorly gauged basin.

We added to the Conclusions the following statements:

"The study covered 16 years from 1994-2009 at a finer spatial and temporal resolution than in previous studies. In order to obtain this result, we used, various remote sensing products, rainfall from SM2R-CCI, cloud cover from SAF EUMETSAT CFC,

evapotranspiration from GLEAM and MODIS (used for comparison), and storage change from GRACE. We also used all the ground data currently available, i.e. sixteen discharge time series, and thirty-five ground based meteorological stations."

And also:

"We claim that the procedure we followed can be easily transported in any other poorly gauged basin with benefits for the hydrological knowledge of any region on Earth."

**1.b Because there's no field observation to validate the modeled water components (except runoff) in this manuscript, evapotranspiration (ET) was estimated using the Priestley and Taylor (PT) Formula. But GLEAM, which was also estimated based on the PT method, was adopted to validate the estimation. If estimates of each component have large uncertainty, who believe the final output?**

Saying that GLEAM estimates evaporation with Priestly Taylor (PT) equation is true, but, at the same time, an approximation, because what characterizes any of the cited products is not only the choice of PT equation but the way its terms and parameters are evaluated.

GLEAM uses PT estimates for potential ET (PET), but the alpha coefficient is set to a particular value (0.98) got from literature, and above all, potential ET is additively increased, by intercepted rainfall estimated according to a version of the Gash model, and multiplicatively decreased by a "stress coefficient" made dependent on five soil cover types (bare soil, snow, tall vegetation, two levels of low vegetation). The stress coefficients are evaluated through various remote sensing products (RS), which are described in the paper by Martens et al., 2017, and applied to a "three reservoirs schematization" on any pixel of 0.25 degree in latitude and longitude (~28 km of side or ~800 square kilometers of area). Eventually also the radiation term is estimated by another different RS.

On the contrary, in our case, the alpha coefficient is estimated by closing the water budget, our stress coefficient does not depend on vegetation type (which could be seen as a limitation) but only on the overall water content in the HRU (of average size of ~420 square kilometers, approximately twice the resolution of GLEAM data). Radiation in our case is estimated locally (and averaged over a representative number of points for each HRU) according to various topographic features present in the HRU (slope, aspect, sky view factor, shadows).

In turn, while we have no means to validate PET or AET on UBN sites, GLEAM was validated in various (64) Fluxnet sites around the world with consistent results (e.g. Martens et al., 2017). Many other differences between GLEAM and JGrass-NewAge can be easily deduced from reading our and Martens paper.

Therefore we can conclude that:

- As an educated best guess, GLEAM supposedly behaves properly in Ethiopia with statistics similar to those it reproduces for other parts of the world, and therefore, it can be used as reference
- GLEAM and our model can be thought as quite independent estimates of the same quantity, even if using the same PT formula, and, therefore, the comparison between our results and GLEAM result does not produce agreements just because they use the same algorithm, as the comment of the Reviewer could imply.

Certainly the word "validation" has to be probably substituted by "comparison" . In the revised manuscript we added part of this information.

We added the following statements in the new manuscript:

"GLEAM, as well NewAge, uses the PT scheme for estimating ET. However, all inputs of the formula in GLEAM and NewAge are according to different strategies and RS tools. GLEAM sets $\alpha_{PT} = 0.98$ while in NewAge it has been calibrated. In GLEAM PET is additively increased, by intercepted rainfall estimated according to a version of the Gash model (Gash, 1979), and multiplicatively decreased by a stress coefficient depending on five soil cover types (bare soil, snow, tall vegetation, two levels of low vegetation) and has a different expression for anyone of the storages. Moreover, according to the case, the stress coefficients are evaluated through various RS, according to procedures which are described in the paper by Martens et al, 2016."

"The most recent version of GLEAM was validated globally over sixty-four Fluxnet sites (Martens et al., 2016) with consistent results, letting us guess that it behaves properly also in Ethiopia. The differences between NewAGE estimation and GLEAM's one allow to assume that the our results and their results can be seen as largely independent."

**2. As mentioned in my previous comments, the discussion section should be enhanced. However, the authors revised limited content in section 4 (Results and Discussion) in this revision (track change in 'hess-2016-290-author_response-version1.pdf'). For example, it seems that the authors kept discussing some water components in other submitted manuscript. Since the authors think combine modeling all the water components is important, why not focus on this study and discuss the possible limits of such a solution? I know the study area is data-scarce and validation using field observation is impossible. However, reasonable discussion on the comparison and the possible errors may persuade the readers (e.g., see specific comments 5, 9, 11, 14, 15, and 18 for detail). Otherwise, the readers may question about the modeled water components and finally your method.**

We thank the reviewer for the suggestions given to enhance the discussion section, and we are sorry if we did not adequately treat all the comments given in the first review. In this version of the MS, we used the specific comments mentioned to enhance our discussion. Detailed comments are reported along with points 5,9,11,14,15 and 18.

**3. Both reviewers mentioned figure quality. However, the authors may have paid little attention. For example, although units were added to axes in Fig. 1, units for axes were still missing in the rest of figures (e.g., Figs. 3, 4, 7, 8).**

The label latitude and longitude is enough to tell that the maps are given in geographical coordinate system, representing the axis in degree. But, given the reviewer suggestion, we have added the unit (degree) in all the figures.

**Following specific comments may help the authors understanding my concerns and improving the manuscript.**
**The numbers in front of the comments indicate page and line number.**

**1.1-6. 'to obtain the estimates of all the components of the hydrological cycle (precipitation, evapotranspiration, discharge, and storage)'. It would be better to revise the claim, because precipitation, evapotranspiration, discharge, and storage are main components. For example, interception and infiltration are also components of the hydrological cycle, which you did not address.**

We are sorry for having neglected this. We modified the manuscript by specifying which components of the budget we were talking about. However, we believe that what the terms present in the water budget depend on the control volume used to estimate it. At the scale we are working the fluxes are those the Reviewer list among parentheses with the exception of the groundwater flow at the outlet, which is assumed negligible. In our case, interception is an internal flux and does not appears in the budget, as well as groundwater recharge.

**2. 4-3. 'Specifically studies' contains grammatical errors. I encourage the authors to check the entire manuscript carefully to avoid such mistakes.**

This paragraph is deleted following editor's specific comment 20.

And we have checked the whole manuscript for any grammar errors.

**3. 4-8. Please correct the unit for temperature.**

Thank you, we corrected the temperature unit. We have changed $18.5^{\circ}$ to $18.5^{\circ}C$.

**4. 5-3. Table 1. I cannot found any description on 'JAMI' or 'three temperature' in the manuscript. If you decide putting such information in the manuscript, please be sure the readers can understand it or find relate context.**

Thank you, it is true that there is no literature about JAMI, hence we removed it from the manuscript. The "three temperature" also revised as "Modelling snow melting using three types of temperature and radiation based algorithms."

**5. 8-5. Eq. (3). In table 1, the author said PT method was used as one method to**

estimate ET. However, 'S(t) and Smax' were added to Eq. (3). Is Eq. (3) valid for this study or is it used for all JGrass-NewAGE application? Furthermore, what's the advantage of using water storage information when estimating ET, especially in data-scarce regions?
In addition, GLEAM estimates are based on the PT method. Can it be used to 'validate' Eq. (3)? Which version of GLEAM did you use?

The addition to a "stress term" to Priestly-Taylor equation is, in fact quite a common practice. This has been popularized, for instance in works by Rodriguez-Iturbe and coworkers, since 1999, but has its roots in the work of Feddes and coworkers. GLEAM itself uses a stress term, even if its estimation differs from our.

We added the following phrase to the revised manuscript:
"The ratio *S(t)/Smax* a stress coefficient which became very popular since the work of Feddes and coworkers (Feddes et al., 2001)."

In our case, we already used previously a similar scheme in Abera et al., 2017. PT formula in fact is used in literature for estimating mostly potential evapotranspiration, while when dealing, as in our case with the actual evapotranspiration, the action of resistances has to be accounted for.

In our case, modeling surface water with a customized version of HYMOD [Moore et al, 1985] we need an estimation of the actual evapotranspiration, AET, to be subtracted to the water available in storages to produce runoff.

Regarding the GLEAM ET in the manuscript we used GLEAM_v3a_BETA. As said with more detail at the beginning of these answers, we believe that GLEAM estimates are quite independent from ours. This and other information was added in the main text.

**6. 9-9. Are 'the ADIGE model', 'the well-known HYMOD model', and 'The NewAge Hymod' the same? If yes, please consider unify the description.**

ADIGE model uses HYMOD model for each HRU. Therefore, Adige model is an assembly of HYMODs, each one for HRU. Detail notes about the Adige model is available in the papers cited. So, we revised the text to clearly show this idea.

**7. 9-12. I'm confused by the description that 'The main inputs for the ADIGE model are J(t) and ET(t)', because 'Q is modelled as functions of basin water storage'.**
**How did you get the water storage? What are the five calibration parameters?**

We have added Appendix A to describe better the ADIGE model and the parameters. Water storage here is assumed to be the sum of the water contained in all the HYMOD storages, for all the HRUs.

**8. 10-17. ET validation is questionable. See specific comment 5.**

Thank you. We already expressed our opinion in answering the general comment. However, we have removed the term "validation", and replaced with "comparison". Please see reply for major comment 1, and for specific comment 5.

**9. 10-21. ds/dt validation. Similar to ET validation, lacking of discussion on GRACE product and the modeled ds/dt. Did you use the GRACE product directly or perform any correction? As reported by studies (e.g., Long et al., 2015, Water Resour. Res., 51, 2574–2594), GRACE data are noisy in smaller basins less than the GRACE footprint (~200,000 km2), as well as in areas with intensive irrigation. Considering the UBN is approximately 176,000 km2 and the highlands have high water demands for irrigation, the product may include typical errors. The author should discuss such uncertainty and the possible impact on the modeled ds/dt.**

The description about GRACE available at page 9 (line 24-33) and page 10 (line 1-4) is enough from our perspective. The area of UBN basin is almost the same size of GRACE footprint (176,000 km2), and, in fact, we used the comparison at the whole basin scale not at subbasin scale, as made more explicit in various parts of the new manuscript. The UBN basin has high water demands for irrigation, but actual irrigation is very minimal, and therefore not affecting our estimates.

**10. 10-26. The headings in section 4 are the same as those in section 3, and the authors claimed that in this way there is a clear relation between the topics of the two sections. I cannot agree with them. For example, in rainfall section, the spatial distribution was described and then compared with some published results. But for ET section, both spatial and temporal distributions were presented, as well as 'validation'. I don't think the headings in results section reflecting any useful information.**

We are sorry if we were not clear when we say that the one-to-one correspondence of the title in the method and in the result sections. But, the idea is simple. In the methodological section, we presented how each component is estimated for each HRU level, and in the result section we analyzed and discussed the result of the component use. Of course, the discussion may not follow exactly the same pattern for four water budget components. For instance, since we have some published work, for precipitation, in addition to the spatial and temporal analysis, comparison of our estimates with some published work is to strengthen our discussion. But, when it comes to ET, we do not have studies in the area, and we instead focus on comparison of modeled ET with GLEAM. We are sorry if we still miss the reviewer's idea!

**11. 11-1. Table 2. The unit for SM2R-CCI's spatial resolution is missing. Could you provide time periods for these data used? Little information about the used data was presented.**
**In section 4.1, can you discuss what's the difference between corrected and uncorrected SM2R-CCI products? That is how systematic error (bias) of SM2RCCI affects rainfall amount, as some ungauged basin has no in-situ observation to perform the correction.**

Thank you, we have added unit 'degree' for SM2R-CCI's spatial resolution. The time period used for the products is given in the text, for instance, SM2R-CCI (in the old manuscript page 6, line 21), for CFC (old manuscript page 8, line 24), and for GLEAM (old manuscript page 10, line 19). However, based on the suggestion, and for clarity, in the revised manuscript, we have added a column in Table 2 for providing the time period used.

The impacts of bias correction on rainfall estimation and on discharge estimation are not the subject of this paper, and can be reviewed from other studies (Habib et al 2014; Gumindoga et al. 2016; Bitew et al 2012; Najmaddin et al. 2017; Valdés-Pineda et al, 2016; among many). Regards to the impacts on discharge, for instance, in Koga basin and Gilgel Abbay (subbasins of the UBN basin), Bitew et al. 2014 and Habib et al. 2014 showed that bias correction procedure improved model discharge prediction performances for all satellites used (CMORPH, PERSIANN, TRMM- 3B42RT, TRMM-3B42).

**12. 14-6. The section is to 'validate' NewAge ET. I'm not sure why the authors talk about GLEAM estimation.**

Given that NewAge estimates ET consistently with the water budget closure, a sentence on GLEAM from this perspective does not harm, in fact, it could be a useful suggestion for improvement of GLEAM in the region.

**13. 14-10. A better correlation between NewAge ET and GLEAM is because they both estimated using the PT method.**

Please see reply for general comment 1, specific comment 5

**14. 14-25. Table 3. It's difficult for readers to know what these parameters represent. Furthermore, aPT has a value of 2.9. It's relatively high compared with the commonly used value (1.26) in the PT method, or the value (1.5-1.8) recommended for estimating ET in more arid regions (ASCE. 1990. Evapotranspiration and irrigation water requirements. ASCE. Manuals and reports on engineering practice. No. 70. New York, NY, USA). In this case, ET may be overestimated. It can be seen from Fig. 4 and Fig. S3 that the NewAge ET higher than GLEAM and MODIS ET, especially the peak value. If ET is overestimated, runoff should be underestimated when precipitation unchanged. Fig. 5 did show that in most cases, the modeled runoff is smaller than the observation, and obvious difference occurred also at peak values. The authors should discuss such physical processes that may cause model uncertainty. Only insightful analyses and discussion on the mechanism behind can highlight the scientific merit of the manuscript.**

Regarding the parameters description, we have added them in Appendix A.

It is true that PT alpha is higher than the literature. However, that literature does not

include a stress factor as we do. Besides, as reported by Cristea et al. (2012) literature values of PT alpha varying from 0.6 to 2.4, making literature quite useless in applications in ungauged basins. In fact, the Reviewer should also notice that most of literature regarding PT estimates does not search for water budget closure, as we do, and their estimates, seen the large errors implied in ET measures, can contain very large bias or not close the water budget at all (Examples of studies really closing the water budget are pretty limited, and, summarized, for instance, at this link: https://www.authorea.com/users/24891/articles/142520-a-list-of-papers-that-perform-the-water-budget-estimation/_show_article ).

The reviewer is right when s/he argues about the reciprocal (inverse) dependence of ET and Q, but the interpretation can be simplistic, since the role of the variation in storage is not trivial: the three not the two quantities have to match, and Figure 10 shows exactly this. Besides our discharges are estimated by calibration (at least at selected gauge station), taken ET into account. Therefore the argument of the reviewer can be, in case, reversed, in the sense that discharge underestimation causes ET overestimation and not vice versa. Actually in our procedure we cannot really disentangle ET and Q estimates. Besides, as it can be read in the manuscript underestimation of discharges (as total volume) can vary spatially, thus making weaker his/her arguments.
Finally, in our experience deriving from many simulations, in basins of various size, we observed that our models, and models similar to our, tend systematically to underestimate most of the peak flows. At present, our investigations cannot discriminate if the problem is actually in models or in measures. The latter in fact are deduced as extension of looping rate curves behavior from normal discharges to high discharges, and it is not actually known how reliable this approach is.
Definitely all the questions raised by the reviewer comment deserve more attention in future work and papers.

The following comments has been added in the revised manuscript (in the discharge section 4.3):

"Additional source of error can also be caused by model inconsistency due to averaging out input data over large areas or from some inadequacy in stage-discharge curves used to obtain discharges from water levels. The slight underestimation of runoff could result from the overestimation of evapotranspiration. However, in this case, GLEAM (or MODIS) would cause larger discrepancies."

The following statement has been added in the revised manuscript at the end of section 4.5:

"The same Figure also shows the complex interplay between discharges, (variation of) storages and evapotranspiration. A first look at Figure 4 and 5 could bring to the conclusion that overestimation of ET brings in underestimation of Q. However, Figure 10 shows that the role of ds/dt is not negligible at all."

**15. 15-1. Table 4. There may be something interesting, i.e., KGE varied with basin area and may have a poor correlation with area. It's often taken for granted that a hydrological model will perform much better in relative smaller basins than in larger ones. Can you discuss why some times the JGrass-NewAGE System performed good or bad in sub basins with similar area?**

We do not share the Reviewer opinion that hydrological models perform better in smaller basins. Our experience, in fact, is the opposite (regarding the only discharge, indeed). However, there are many factors that could affect the model performance. These basins are distributed in various climatic zones, with different topography, and different vegetation distribution. All these factors can affect the model performance, not just the basin size. For instance, taking lake Tana station the model performance is relatively weak most likely due to the water regulation by the lake. This is commented on page 17 line 7 (old manuscript). As suggested, we have added the following general sentence to indicate this:

"Model performance varies with basins and a consistent behavior with respect to basin size, climate, vegetation density and topographic complexity is not found. Indeed, there are many factors that affect the model performance, including uncertainties in input observations."

**16. 16-1. Fig. 5. Is there any observation used in Fig. d?**

There is no observation for this site. This is plotted to show the model forecast at any link whether we have observation or not.

**17. 18-13. What does S mean? Please consider defining the abbreviation.**

Sorry for this error, we have changed S into ds/dt, as it should be.

**18. 21-1. Fig. 9. It would be better to change the water components to percentage.**

The percentage has a limitation. It does not show the actual volume, the annual variability between the components and the total water budget. If we change the bar into percentage, all years will have the same bar length. In the present form, we can see the total volume of each component and the total water budget. We already have annual mean percentage for each component reported in the text.

**19. 22-1. Fig. 10. I'm curious why ET was so low in the hot season. Supposing most of the rainfall infiltrated into the deep soil (high ds/dt values) in the hot and wet seasons, can they evaporate easily in the dry season? Again, more discussion may be required to persuade the readers about the modeling results.**

In the first months of hot season i.e. September, October, November, and December, ET is very high for the reasons mentioned by the reviewer, i.e., due to accumulation of

storage in rainy season (June, July, and August) and lack of cloud causes high evapotranspiration. But, during the last months of hot season (March, April and May), lack of storage limits ET despite atmospheric water demand is high due to high net radiation.

It was:

"Figure 10 provides long term monthly mean estimates of water budget fluxes and storage. The basin scale mean budget is highly variable. The highest variability is mainly in J and S. During summer months, J, Q, and ds/dt shows high magnitude. ET is not highest in June, July and August, but in October and December it is. The S accumulated in the summer season feeds the highest ET in autumn, and causes very high drops in ds/dt (figure 10)."

We have added more discussion on this and revised as follows:

"Figure 10 provides long term monthly mean estimates of water budget fluxes and storage. The basin scale mean budget is highly variable. The highest variability is mainly in J and ds/dt. During summer months, J, Q, and ds/dt shows high magnitude. ET is not high in June, July and August, but in October and December. The S(t) accumulated in the summer season feeds high ET in autumn, and causes very high drops in ds/dt (figure 10). The seasonal trend between J and ET is slightly out-of-phase, i.e., the highest energy to evaporate water (March, April, May) occurs during low precipitation months. Due to this slight out-of-phase, ET is minimal and Q and ds/dt is enhanced during wet months (figure 10), thus revealing that ET is water limited more than energy limited."

[revised manuscript text omitted]